# Lions and Muons: Optimization via Stochastic Frank-Wolfe under Heavy-Tailed Noises

**Maria-Eleni Sfyraki** [1]   **Jun-Kun Wang** [1]

## Abstract

Stochastic Frank-Wolfe is a classical optimization method for solving constrained optimization problems. On the other hand, recent optimizers such as Lion and Muon have gained quite significant popularity in deep learning. In this work, building on recent initiatives, we provide a unifying perspective by interpreting these seemingly disparate methods through the lens of Stochastic Frank-Wolfe. Specifically, we show that Lion and Muon with weight decay can be viewed as special instances of a Stochastic Frank-Wolfe, and we establish their convergence guarantees in terms of the Frank-Wolfe gap, a standard stationarity measure in non-convex optimization for Frank-Wolfe methods. We further find that convergence to this gap implies convergence to a KKT point of the original problem under a norm constraint for Lion and Muon. Moreover, motivated by recent empirical findings that stochastic gradients in modern machine learning tasks often exhibit heavy-tailed distributions, we extend Stochastic Frank-Wolfe to settings with heavy-tailed noise by developing two robust variants with strong theoretical guarantees that hold for general compact convex sets without the need for a large batch size, filling the gap in the literature on Stochastic Frank-Wolfe for non-convex optimization. Our contributions in the later part of this work, in turn, yield new variants of Lion and Muon, that better accommodate heavy-tailed gradient noise, thereby enhancing their practical scope.

---

[1]University of California San Diego. Correspondence to: Maria-Eleni Sfyraki <msfyraki@ucsd.edu>, Jun-Kun Wang <jkw005@ucsd.edu>.

*Proceedings of the 43$^{rd}$ International Conference on Machine Learning*, Seoul, South Korea. PMLR 306, 2026. Copyright 2026 by the author(s).

## 1  Introduction

Frank-Wolfe (FW) methods (i.e., Conditional Gradient Methods) are classical algorithms in optimization and machine learning (Frank & Wolfe, 1956; Jaggi, 2013). Over the past decade, there has been sustained interest in extending and analyzing FW variants in a broad range of settings, including refined algorithms and convergence analysis (Lan & Zhou, 2016; Lacoste-Julien, 2016; Li et al., 2021a; Lu & Freund, 2021; Braun et al., 2022; Chen & Mazumdar, 2024; Martínez-Rubio & Pokutta, 2025), acceleration under certain constraint sets for minimizing functions with certain properties (Garber & Hazan, 2015; Abernethy et al., 2018; Kerdreux et al., 2021; Garber, 2023; 2025), projection-free algorithms for online learning (Hazan & Luo, 2016; Wan et al., 2022; Garber & Kretzu, 2023), connections to submodular maximization (Mokhtari et al., 2020), a new interpretation as finding a game-theoretic equilibrium (Abernethy & Wang, 2017; Wang et al., 2024), and stochastic optimization (Hazan & Luo, 2016; Reddi et al., 2016; Yurtsever et al., 2019; Shen et al., 2019; Zhang et al., 2020b; Negiar et al., 2020; Hassani et al., 2020; Beznosikov et al., 2023; Nazykov et al., 2024), among others, to name just a few. FW-type methods, which rely on a linear optimization oracle, can enjoy lower per-iteration cost compared to methods that require a projection for some constrained optimization problems. However, in deep learning, training is typically framed as an unconstrained minimization problem, where practitioners often favor recent optimizers such as AdamW (Loshchilov & Hutter, 2018), Lion (Chen et al., 2024b), and Muon (Jordan et al., 2024). It remains unclear to what extent FW methods are practically effective or competitive as optimizers for training neural networks, despite previous efforts to explore their application in training deep learning models (e.g., Pokutta et al. (2020)).

In this work, we first show that Lion and Muon, two recent state-of-the-art optimization methods in deep learning, are in fact specific instances of a stochastic FW. This unification is grounded in a sequence of recent insights that reveal how solving the linear optimization oracle under various norm constraints naturally gives rise to a few popular updates in training neural networks such as obtaining a signed stochastic gradient (Bernstein et al., 2018) and getting a

preconditioned stochastic gradient (Chen et al. (2024a); Xie & Li (2024); Bernstein & Newhouse (2024); Pethick et al. (2025)). Yet, we note that the similar observations can be traced back to an earlier initiative of investigating the linear optimization oracle in the FW literature, see e.g., Jaggi (2013); Combettes & Pokutta (2021) and the references therein. We advance this perspective by establishing that Lion is exactly a stochastic FW under an $l_\infty$-norm constraint, while Muon with weight decay corresponds precisely to the *same* stochastic FW under a spectral-norm constraint. We further provide a convergence rate guarantee of the proposed stochastic FW in terms of the Frank-Wolfe Gap under the standard smoothness and bounded variance assumption, and hence our result offers a theoretical support for Lion and Muon. We then discuss the interpretation of convergence guarantee for the Frank-Wolfe gap and show that the notion implies converging to a KKT point of the original optimization problem subject to a corresponding norm constraint, for Lion and Muon.

We then delve further into Stochastic FW under the scenario where the stochastic gradients satisfy a $p$-moment noise assumption, where $p \in (1, 2]$. More precisely, we consider solving $\min_{\mathbf{x} \in \mathcal{C}} F(\mathbf{x}) := \mathbb{E}_\xi [f(\mathbf{x}, \xi)]$, using an unbiased stochastic gradient oracle that returns $\nabla f(\mathbf{x}, \xi)$ and satisfies the bounded $p$-th moment noise condition: $\mathbb{E}_\xi [\|\nabla f(\mathbf{x}, \xi) - \nabla F(\mathbf{x})\|^p] \leq \sigma^p$ for some $p \in (1, 2]$. The $p$-th moment noise assumption in stochastic optimization can be traced back to Zhang et al. (2020b), which is motivated from the observations that the finite-variance assumption (i.e., bounded second moment) may not hold when training modern machine learning models such as deep neural networks (Şimşekli et al., 2019; Zhang et al., 2020a). Therefore, Zhang et al. (2020a) proposed considering the bounded $p$-th moment noise condition, where the case of $p < 2$ can capture heavy-tailed noise. They further provide a couple of concrete examples of distributions that has bounded $p$-th moment for some $p < 2$ but unbounded variance. Since then, there has been a flurry of research in stochastic optimization that provide improved theoretical analyses and algorithmic developments under this setting, e.g., Cutkosky & Orabona (2019); Nguyen et al. (2023); Kornilov et al. (2023); Liu et al. (2023); Sadiev et al. (2023); Liu & Zhou (2025); Puchkin et al. (2024); Hübler et al. (2024); Kornilov et al. (2025). However, to the best of our knowledge, little is known about stochastic optimization methods that rely on a *linear optimization oracle* under the bounded $p$-th moment noise condition. Hence, we explore this direction and propose two Stochastic FW type methods. The first method incorporates clipping the magnitude of the stochastic gradients and converges to the FW gap at a rate of $O\left(\log(T/\delta)T^{\frac{1-p}{3p-2}}\right)$, with probability at least $1 - \delta$. The second method integrates both the clipping technique and a variance reduction method and achieves an improved

rate of $O\left(\log(T/\delta)T^{\frac{1-p}{2p-1}}\right)$, under the additional structural assumption that the stochastic gradients are Lipschitz. This result parallels the aforementioned related works on SGD-type methods in terms of finding a point with $\epsilon$ gradient norm with high probability. Furthermore, by building upon the connection between FW, Lion, and Muon established in the first part of our contributions, our result leads to two variants of Lion and Muon with strong guarantees under the heavy-tailed noise.

Moreover, in the $p = 2$ regime where the variance of the stochastic gradient is bounded, we also provide a convergence guarantee in terms of the expected FW gap for our proposed method. Specifically, we show that the total number of stochastic gradient evaluations is $O(1/\epsilon^3)$ to get an expected $\epsilon$ gap, which not only matches the best known complexity results in Yurtsever et al. (2019); Hassani et al. (2020); Zhang et al. (2020b); Nazykov et al. (2024), but also simultaneously avoids the need for a gigantic average batch size across iterations *and* relies additionally only on the smoothness and the averaged Lipshitz gradient assumption. To our knowledge, this is the first result in the literature of Stochastic FW that achieves both desiderata.

## 2 Preliminaries

### 2.1 Notations and assumptions

As this work concerns optimization via Stochastic FW, we begin by recalling the definition of the Frank-Wolfe gap, i.e.,

$$\mathcal{G}(\mathbf{x}) := \max_{\mathbf{v} \in \mathcal{C}} \langle \mathbf{v} - \mathbf{x}, -\nabla F(\mathbf{x}) \rangle. \tag{1}$$

One can show that $\mathcal{G}(\mathbf{x}) = 0$ if and only if $\mathbf{x}$ is a stationary point (Lacoste-Julien, 2016). Furthermore, when the function $F(\cdot)$ is convex, then the Frank-Wolfe Gap $\mathcal{G}(\mathbf{x})$ is an upper bound of the optimality gap $\Delta_x := F(\mathbf{x}) - \min_{\mathbf{x} \in \mathcal{C}} F(\mathbf{x})$, i.e., $\Delta_x \leq \mathcal{G}(\mathbf{x})$, which can be obtained naturally as a by-product of solving the linear optimization problem in FW.

In this paper, we focus on a non-convex stochastic optimization problem of the form: $\min_{\mathbf{x} \in \mathcal{C}} F(\mathbf{x}) := \mathbb{E}_\xi [f(\mathbf{x}, \xi)]$, where the objective $F : \mathcal{H} \to \mathbb{R}$ is a differentiable function defined over a Hilbert space $\mathcal{H}$, and $\xi$ is an independent random variable that represents the randomness of the stochastic gradient (e.g. the randomness of the mini-batch sampling). We let $\|\mathbf{A}\|_2$ and $\|\mathbf{A}\|_{tr}$ represent the spectral and nuclear norm, respectively, when $\mathbf{A}$ is a matrix. Recall that for a norm $\|\cdot\|$, its *dual norm* $\|\cdot\|_*$ is defined as $\|\mathbf{y}\|_* := \sup_{\|\mathbf{x}\| \leq 1} \langle \mathbf{y}, \mathbf{x} \rangle$. For example, the dual norm of an $\ell_p$ norm is an $\ell_q$ norm, where $\frac{1}{p} + \frac{1}{q} = 1, p, q \in [1, \infty]$. We further let $\mathcal{O}_{m \times n} := \{\mathbf{A} \in \mathbb{R}^{m \times n} : \mathbf{A}^\top \mathbf{A} = \mathbf{I}_n \text{ or } \mathbf{A} \mathbf{A}^\top = \mathbf{I}_m\}$ denote the set of semi-orthogonal matrices. For brevity, we

let $a \vee b := \max(a, b)$ and $a \wedge b := \min(a, b)$ denote the maximum and minimum operators, respectively.

In all the assumptions that follow, we clarify that the norms involved are Hilbert-space norms. We assume that the constraint set $\mathcal{C} \subseteq \mathcal{H}$ is a convex and compact set with diameter $D$, i.e. $\|\mathbf{x} - \mathbf{y}\| \leq D$, for all $\mathbf{x}, \mathbf{y} \in \mathcal{C}$. Throughout, we also assume that $F$ is bounded below, i.e. $\inf_{\mathbf{x} \in \mathcal{C}} F(\mathbf{x}) > -\infty$. We will use the following assumptions in this work. However, not all of our results rely on all of the following assumptions. For each theoretical result in this work, we will explicitly state which assumptions in the following are invoked.

**Assumption 2.1.** (L-smoothness of $F(\cdot)$) The function $F(\cdot)$ is L-smooth, i.e. for all $\mathbf{x}, \mathbf{y} \in \mathcal{C}$, we have $F(\mathbf{y}) \leq F(\mathbf{x}) + \langle \nabla F(\mathbf{x}), \mathbf{y} - \mathbf{x} \rangle + \frac{L}{2} \|\mathbf{y} - \mathbf{x}\|^2$.

**Assumption 2.2.** (Averaged L-Lipschitz gradient of $f(\cdot, \xi)$) The function $f(\cdot, \xi)$ has averaged $L$-Lipschitz gradient, i.e. for all $\mathbf{x}, \mathbf{y} \in \mathcal{C}$ we have $\mathbb{E}_\xi \left[ \|\nabla f(\mathbf{x}, \xi) - \nabla f(\mathbf{y}, \xi)\|^2 \right] \leq L^2 \|\mathbf{x} - \mathbf{y}\|^2$.

**Assumption 2.3.** (a) (L-Lipschitz gradient of $F(\cdot)$) $\|\nabla F(\mathbf{x}) - \nabla F(\mathbf{y})\| \leq L\|\mathbf{y} - \mathbf{x}\|, \forall \mathbf{x}, \mathbf{y} \in \mathcal{C}$. (b) (L-Lipschitz gradient of $f(\cdot, \xi)$) $\|\nabla f(\mathbf{x}, \xi) - \nabla f(\mathbf{y}, \xi)\| \leq L\|\mathbf{x} - \mathbf{y}\|, \forall \mathbf{x}, \mathbf{y} \in \mathcal{C}$, w.p. 1.

We note that Assumption 2.3-a implies Assumption 2.1.

**Assumption 2.4.** (Bounded $p_{\text{th}}$ moment noise) The stochastic gradient $\nabla f(\cdot, \xi)$ is an unbiased estimate of the true gradient $\nabla F(\cdot)$, i.e. $\mathbb{E}_\xi[\nabla f(\mathbf{x}, \xi)] = \nabla F(\mathbf{x}), \forall \mathbf{x} \in \mathcal{C}$. Furthermore, for some finite $\sigma \geq 0$, we have $\mathbb{E}_\xi[\|\nabla f(\mathbf{x}, \xi) - \nabla F(\mathbf{x})\|^p] \leq \sigma^p, \forall \mathbf{x} \in \mathcal{C}$, where $p \in (1, 2]$.

It is noted that when $p = 2$, i.e., $\mathbb{E}_\xi[\|\nabla f(\mathbf{x}, \xi) - \nabla F(\mathbf{x})\|^2] \leq \sigma^2$, Assumption 2.4 becomes the assumption that the variance of the stochastic gradients is bounded, which is a common assumption in stochastic optimization (Fang et al., 2018; Tran-Dinh et al., 2019; Cutkosky & Orabona, 2019; Levy et al., 2021; Li et al., 2021b).

**Assumption 2.5.** (Bounded gradient norm over the set $\mathcal{C}$) For all $\mathbf{x} \in \mathcal{C}$, we have $\|\nabla F(\mathbf{x})\| \leq G$.

## 2.2 Lion and Muon

The Lion optimizer, proposed by Chen et al. (2024b), is a recent algorithm discovered via program search that has been experimentally shown to have superior generalization properties on various tasks compared to AdamW, which in turns has drawn further research attention in improving the algorithm and theoretical foundation (Chen et al., 2024a; Liu et al., 2024; Dong et al., 2024; Kosson et al., 2024b;a; Liang et al., 2024; 2025; Zhao et al., 2025). Notably, Chen et al.

(2024a) introduce a general family of Lion-$\mathcal{K}$ algorithms by developing a Lyapunov function for the optimization dynamics and show that Lion solves an $\ell_\infty$-norm constrained optimization problem. Kosson et al. (2024b) analyze the influence of the interplay between weight decay and learning rate in the update dynamics of various optimizers, including Lion. A subsequent work by the same authors proposes two Lion variants that scale the update size and adjust the hyperparameter choices to be more comparable to those of AdamW (Kosson et al., 2024a). Yuan et al. (2025) develop a framework by incorporating variance reduction into adaptive gradient methods, with one variant based on Lion. While their setting offers interesting insights, in contrast to this work, it does not incorporate the weight decay term, and convergence guarantees for the Lion variant are not included. They further assume the use of positive-definite preconditioners, a condition relying on the algorithmic iterates.

Muon (Jordan et al., 2024), on the other hand, is a preconditioned gradient method. The design of Muon was motivated from improving Shampoo (Gupta et al., 2018), which aims to maintain necessary second-order information while being efficient in optimization over tensor spaces. Shampoo regained prominence after winning the external tuning track at the 2024 AlgoPerf: Training Algorithms competition (Dahl et al. (2023); Vyas et al. (2025); Morwani et al. (2025)). Building on this, Jordan et al. (2024) propose Muon, an update rule that can be interpreted as a variant of Shampoo without the use of preconditioner accumulators. They further introduce a more efficient variant of Muon by incorporating an additional Nesterov momentum step. Bernstein & Newhouse (2024) demonstrate that Muon can also be viewed as a steepest descent method under the spectral norm and provide a more efficient Newton-Schulz iteration scheme to perform approximate SVD. A recent work by Pethick et al. (2025) showcased Muon without the additional Nesterov-momentum step as an instance of one of their proposed methods labeled Unconstrained Stochastic Conditional Gradient Method, with a guarantee on the expected gradient norm, and also recovered Muon with weight decay from a constrained variant, with a guarantee on the expected Frank-Wolfe gap. Their work further establishes a connection between the linear minimization oracle in Stochastic Frank-Wolfe and other norm-constrained optimizers, offering a valuable step toward unifying these approaches. While offering important contributions, their constrained variant does not incorporate the momentum extrapolation step, and as a result, does not recover Lion or the Nesterov-momentum variant of Muon. Since then, several works have analyzed the convergence properties of Muon in the absence of a weight-decay term (Li & Hong, 2025; Shen et al., 2025; An et al., 2026; Kovalev, 2025; Riabinin et al., 2025). Notably, Chen et al. (2025) extend this line

---

**Algorithm 1** Lion (Chen et al., 2024b)

---

**Required:** Momentum parameters $\beta_1, \beta_2$, step size $\{\eta_t\}$, weight decay parameter $\lambda$.
**Initialize:** $\mathbf{m}_0 = 0$ and $\mathbf{x}_1 \in \mathcal{C}$.
**for** $t = 1, 2, \ldots$ **do**
  Sample $\Xi_t \sim \mathcal{D}$.
  Compute $\bar{\mathbf{g}}_t = \nabla f(\mathbf{x}_t; \Xi_t)$.
  Update $\mathbf{c}_t = \beta_1 \mathbf{m}_{t-1} + (1 - \beta_1)\bar{\mathbf{g}}_t$.
  Update $\mathbf{x}_{t+1} = \mathbf{x}_t - \eta_t (\text{sign}(\mathbf{c}_t) + \lambda \mathbf{x}_t)$.
  Update $\mathbf{m}_t = \beta_2 \mathbf{m}_{t-1} + (1 - \beta_2)\bar{\mathbf{g}}_t$.
**end for**

---

**Algorithm 2** Muon (Jordan et al., 2024)

---

**Required:** Momentum parameter $\mu$, step size $\{\eta_t\}$, weight decay parameter $\lambda$.
**Initialize:** $\mathbf{B}_0 = 0$ and $\mathbf{X}_1 \in \mathcal{C}$.
**for** $t = 1, 2, \ldots$ **do**
  Sample $\Xi_t \sim \mathcal{D}$.
  Compute $\mathbf{G}_t = \nabla f(\mathbf{X}_t; \Xi_t) \in \mathbb{R}^{m \times n}$.
  Update $\mathbf{B}_t = \mu \mathbf{B}_{t-1} + \mathbf{G}_t$.
  Update $\mathbf{O}_t = \arg\min_{\mathbf{A} \in \mathcal{O}_{m \times n}} \|\mathbf{A} - \mathbf{B}_t\|_F$.
  Update $\mathbf{X}_{t+1} = \mathbf{X}_t - \eta_t (\mathbf{O}_t + \lambda \mathbf{X}_t)$.
**end for**

---

of work by studying the convergence of Muon with weight decay within the Lion-$\mathcal{K}$ framework. Muon with weight decay has been shown experimentally to outperform vanilla Muon in the over-train regime (Liu et al., 2025a). We emphasize that, while convergence guarantees for Muon with weight decay can be recovered from the proposed Stochastic Frank-Wolfe in this paper, our results are established for the general Stochastic Frank-Wolfe and therefore extend to a broad class of algorithms. Moreover, new variants of Muon can be derived within our proposed methods, for which we also recover the corresponding convergence guarantees. Several recent variants of Muon have additionally been proposed, including Ma et al. (2025); Liu et al. (2025b); Lau et al. (2025); Huang et al. (2025); He et al. (2025); Si et al. (2025); Liu et al. (2025a).

## 3 Lion and Muon as a Stochastic FW

The mechanism behind the Frank-Wolfe method and its variants in constrained optimization lies in their projection-free property. In particular, at each iteration, the algorithm solves a linear minimization problem of the form

$$\arg\min_{\mathbf{v} \in \mathcal{C}} \langle \mathbf{v}, \mathbf{g} \rangle,$$

for some input $\mathbf{g} \in \mathcal{H}$, followed by a convex averaging step. Solving the linear optimization problem can be less expensive compared to the projection for some constraint sets $\mathcal{C}$. Of particular interest is the special case where the constraint set is set to be a norm constraint of the form $\|\mathbf{v}\| \leq \frac{1}{\lambda}$, for some norm $\|\cdot\|$ and sharpness $\lambda > 0$. More specifically, for any $\mathbf{g} \in \mathcal{H}$, it is easy to verify that $\arg\min_{\|\mathbf{v}\| \leq \frac{1}{\lambda}} \langle \mathbf{v}, \mathbf{g} \rangle = -\frac{\partial \|\mathbf{g}\|_*}{\lambda}$, where $\partial \|\mathbf{g}\|_*$ denotes the subdifferential of $\|\cdot\|_*$ at point $\mathbf{g}$, defined as $\partial \|\mathbf{g}\|_* := \{\mathbf{x} \in \mathbb{R}^d : \|\mathbf{x}\| = 1, \langle \mathbf{x}, \mathbf{g} \rangle = \|\mathbf{g}\|_*\}$. Therefore, if $\mathbf{g}$ is a gradient, $\mathbf{v} = -\partial \|\mathbf{g}\|_*/\lambda$ can be viewed as a scaled steepest descent direction, i.e., the direction that minimizes the inner product with the gradient.

What we are going to show is that Lion is an instance of a variant of Stochastic FW (Algorithm 3) under an $\ell_\infty$-norm ball constraint.

---

**Algorithm 3** A stochastic FW method

---

**Required:** Momentum parameters $\{\beta_{1,t}\}, \{\gamma_t\}$, step size $\{\eta_t\}$.
**Initialize:** $\mathbf{g}_0 = 0$, and $\mathbf{x}_1 \in \mathcal{C}$.
**for** $t = 1, 2, \ldots$ **do**
  Sample $\Xi_t \sim \mathcal{D}$.
  Compute $\bar{\mathbf{g}}_t = \nabla f(\mathbf{x}_t; \Xi_t)$.
  Update $\mathbf{g}_t = (1 - \gamma_t) \mathbf{g}_{t-1} + \gamma_t \bar{\mathbf{g}}_t$.
  Set $\hat{\mathbf{g}}_t = \frac{\beta_{1,t}}{(1 - \gamma_t)} \mathbf{g}_t + \left(1 - \frac{\beta_{1,t}}{(1 - \gamma_t)}\right) \bar{\mathbf{g}}_t$.
  Compute $\mathbf{u}_t = \arg\min_{\mathbf{v} \in \mathcal{C}} \langle \mathbf{v}, \hat{\mathbf{g}}_t \rangle$.
  Update $\mathbf{x}_{t+1} = (1 - \eta_t)\mathbf{x}_t + \eta_t \mathbf{u}_t$.
**end for**

---

**Theorem 3.1.** *(Lion as Stochastic FW) Lion (Algorithm 1) is an instance of a Stochastic Frank-Wolfe (Algorithm 3) when using parameters $\beta_{1,t} = \beta_1$, $\gamma_t = 1 - \beta_2$, $\eta_t^{\text{Alg 3}} = \lambda \eta_t^{\text{Alg 1}}$, for all $t$, setting $\mathcal{C} = \{\mathbf{v} \in \mathbb{R}^n : \|\mathbf{v}\|_\infty \leq \frac{1}{\lambda}\}$, and letting $\langle \cdot, \cdot \rangle$ denote the Euclidean inner product.*

For the proof of Theorem 3.1, we refer the reader to Appendix B.

The Muon optimizer (Jordan et al., 2024) is designed for optimization problems where the model is structured as blocks of weight matrices (e.g., neural networks). Bernstein & Newhouse (2024) demonstrate that at each iteration, the algorithm performs steepest descent under the spectral norm ball. Similarly, Pethick et al. (2025) establish that at each iteration the Muon update step uses a linear minimization of the form $\arg\min_{\|\mathbf{A}\|_2 \leq \frac{1}{\lambda}} \langle \mathbf{A}, \mathbf{G} \rangle$, for some $\lambda > 0$, where $\langle \cdot, \cdot \rangle$ denotes the Frobenius inner product and $\|\cdot\|_2$ denotes the spectral norm. We show that by adding a weight decay term to the Muon update, as shown in Algorithm 2, the method can be produced from the same variant of Stochastic FW that yields Lion (i.e., Algorithm 3)

**Theorem 3.2.** *(Muon as Stochastic FW) Muon (Algorithm 2) is an instance of a Stochastic Frank-Wolfe (Algorithm 3) when using the parameters $\beta_{1,t} = \mu$, $\gamma_t = 1 - \mu$, $\eta_t^{\text{Alg 3}} = \lambda \eta_t^{\text{Alg 2}}$, for all $t$, setting $\mathcal{C} = \{\mathbf{A} \in \mathbb{R}^{m \times n} : \|\mathbf{A}\|_2 \leq \frac{1}{\lambda}\}$, and letting $\langle \cdot, \cdot \rangle$ denote the Frobenius inner product.*

The proof of Theorem 3.2 is deferred to Appendix B.

Notably, it can be shown that Muon with the additional Nesterov momentum step also falls within our Stochastic Frank-Wolfe. Due to space constraints, we defer the details to Appendix B.

Theorem 3.3 below provides the convergence rate guarantee for Algorithm 3, and its proof can be found in Appendix B. We emphasize that the theorem considers the case where the user employs a batch size $m_t$ of samples to construct a stochastic gradient estimate $\bar{\mathbf{g}}_t = \nabla f(\mathbf{x}_t; \Xi_t) = \frac{1}{m_t} \sum_{i=1}^{m_t} \nabla f(\mathbf{x}_t; \xi_{t,i})$ at each iteration $t$. The batch size controls the variance of the stochastic gradient. In particular, we have $\mathbb{E}\left[\|\bar{\mathbf{g}}_t - \nabla F(\mathbf{x})\|^2\right] \leq \frac{\sigma^2}{m_t}$, when $m_t$ samples are used to obtain $\bar{\mathbf{g}}_t$.

---

**Theorem 3.3.** *Set* $\eta_t = \frac{1}{D\sqrt{T}}$, $\gamma_t = \gamma$, $\beta_{1,t} = \beta$ *and the batch size* $m_t = m$, *for all* $t \geq 1$. *Let* $\mathbf{x}_a$ *be chosen uniformly at random from* $\{\mathbf{x}_t\}_{t=1}^T$. *Then, under Assumptions 2.3-a and 2.4 with* $p = 2$ *(i.e., bounded variance), Algorithm 3 satisfies* $\mathbb{E}[\mathcal{G}(\mathbf{x}_a)] = O\left(\frac{D(F(\mathbf{x}_1)-F(\mathbf{x}_*)+L(1/2+\beta/\gamma))}{T^{1/2}} + \frac{D\sigma}{\sqrt{m}}\left(\frac{\beta}{\gamma(1-\gamma)}+1\right)\right)$, *for any* $\beta \in [0, 1-\gamma]$ *and* $\gamma \in (0, 1)$.

---

Theorem 3.3 establishes an upper bound of the expected Frank-Wolfe gap that holds for any batch size $m$. If one sets $m_t = m = T$, then the following guarantee is implied by Theorem 3.3.

---

**Corollary 3.4.** *In Theorem 3.3, set* $m_t = T$, *for all* $t \geq 1$. *Then, Algorithm 3 satisfies* $\mathbb{E}[\mathcal{G}(\mathbf{x}_a)] = O\left(\frac{D(F(\mathbf{x}_1)-F(\mathbf{x}_*)+L+\sigma)}{T^{1/2}}\right)$.

---

We note that the Corollary 3.4 suggests that the number of calls to stochastic first-order oracle (SFO) (i.e., stochastic gradient computations) for getting an expected $\epsilon$-gap is $O(1/\epsilon^4)$ in the worst case, which matches an existing rate of standard Stochastic FW in Reddi et al. (2016) and the rate of a variant of Stochastic FW considered in Pethick et al. (2025) (see Lemma 5.6 in Pethick et al. (2025)). While Corollary 3.4 employs a batch size that grows with $T$, the same scaling is also necessary in the analysis of standard Stochastic Frank–Wolfe in Reddi et al. (2016) to attain the $O(1/\epsilon^4)$ SFO complexity. Later, we propose three variants (Algorithms 4, 5, and 6) that effectively eliminate the large batch size requirement. Among them, Algorithm 5 achieves with high probability the same SFO complexity as Corollary 3.4 up to a logarithmic factor under the bounded-variance assumption, while using a batch size of one. Algorithm 6 further improves the SFO complexity while also maintaining a batch size of one.

Next, we highlight the implication of the convergence of the Frank-Wolfe gap for Lion and Muon — finding a KKT point of the constrained optimization problem:

$$\min_{\|\mathbf{x}\| \leq \frac{1}{\lambda}} F(\mathbf{x}), \qquad (2)$$

where for Lion, the norm in (2) is $\|\cdot\|_\infty$ (c.f., Theorem 3.1), while for Muon, the norm is the matrix spectral norm $\|\cdot\|_2$ (c.f., Theorem 3.2). We recall that when a pair of primal variable $\mathbf{x}_*$ and a dual variable $\mu_*$ satisfies the KKT conditions, it means that they satisfy (1) primal feasibility, i.e., $\|\mathbf{x}_*\| \leq \frac{1}{\lambda}$, (2) dual feasibility, i.e., $\mu_* \geq 0$, (3) stationarity, i.e., $0 \in \nabla F(\mathbf{x}_*) + \mu_* \partial \|\mathbf{x}_*\|$, and (4) complementary slackness, i.e., $\mu_*(\|\mathbf{x}_*\| - \frac{1}{\lambda}) = 0$. In this case, we say that $\mathbf{x}_*$ is a KKT point, following the terminology of Xie & Li (2024), who provide a precise equivalent characterization of a KKT point, stated as follows:

**Lemma 3.5** (Lemma 3.8 in Xie & Li (2024)). $\mathbf{x}$ *is a KKT point of* (2) *if and only if* $\|\mathbf{x}\| \leq \frac{1}{\lambda}$ *and* $\langle -\lambda\mathbf{x}, \nabla F(\mathbf{x})\rangle = \|\nabla F(\mathbf{x})\|_*$.

Xie & Li (2024) show that if AdamW (Loshchilov & Hutter, 2018) converges with non-increasing step sizes, then it must converge to a KKT point of (2) under the $\ell_\infty$-norm constraint asymptotically. On the other hand, we find that the Frank-Wolfe gap serves as a metric for measuring convergence to a KKT point. To be completely precise, when the constraint set $\mathcal{C}$ is a norm-ball constraint, we have

$$\begin{aligned}\mathcal{G}(\mathbf{x}) &= \max_{\|\mathbf{v}\| \leq \frac{1}{\lambda}} \langle \mathbf{v} - \mathbf{x}, -\nabla F(\mathbf{x})\rangle \\ &= \frac{1}{\lambda} \|\nabla F(\mathbf{x})\|_* + \langle \mathbf{x}, \nabla F(\mathbf{x})\rangle. \qquad (3)\end{aligned}$$

The above expression together with Lemma 3.5 shows that converging to a Frank-Wolfe gap is exactly equivalent to obtaining a KKT point.

In the context of Lion, we note that Dong et al. (2024) present convergence guarantees in terms of the quantity, $\frac{1}{T}\sum_{t=1}^T \mathbb{E}\left[\lambda\langle\nabla F(\mathbf{x}_t), \mathbf{x}_t\rangle + \|\nabla F(\mathbf{x}_t)\|_1\right]$, which is exactly the expected Frank-Wolfe gap up to a multiplication constant $\lambda$, since it holds that $\frac{1}{T}\sum_{t=1}^T \mathbb{E}\left[\lambda\langle\nabla F(\mathbf{x}_t), \mathbf{x}_t\rangle + \|\nabla F(\mathbf{x}_t)\|_1\right] = \lambda\frac{1}{T}\sum_{t=1}^T \mathbb{E}[\mathcal{G}(\mathbf{x}_t)] = \lambda\mathbb{E}[\mathcal{G}(\mathbf{x}_a)]$. Their result shows an iteration complexity of $O\left(\frac{d^2}{\epsilon^4}\right)$ to converge to an $\epsilon$-gap, where $d$ is the dimension. Similarly, in the case of Muon with weight decay, Chen et al. (2025) employ a KKT score function $\mathcal{S}(\mathbf{X})$, and derive their results using the quantity $\frac{1}{T}\sum_{t=1}^T \mathbb{E}[\mathcal{S}(\mathbf{X}_t)] = \frac{1}{T}\sum_{t=1}^T \mathbb{E}\left[\langle\lambda\mathbf{X}_t, \nabla F(\mathbf{X}_t)\rangle + \|\nabla F(\mathbf{X}_t)\|_{tr}\right]$, which is analogous to the Frank-Wolfe gap in the matrix setting. The provided iteration complexity matches the bound in Theorem 3.3. In comparison with these works, the unifying perspective of *Lion and Muon as Stochastic FW* presented in our work yields a more general result, as the convergence result in Theorem 3.3 applies to both Lion and Muon.

# 4 A deeper investigation of Stochastic FW methods under the bounded moment noise

In this section, we conduct a deeper investigation of stochastic FW methods under Assumption 2.4, which characterizes the moment of the noise present in stochastic gradient estimates.

## 4.1 Light-tailed regime

Corollary 3.4 in the previous section shows a required $O(1/\epsilon^4)$ calls to SFO to guarantee an expected $\epsilon$ gap. However, there has been extensive research in recent years aimed at improving the complexity of stochastic Frank-Wolfe methods. These efforts integrate variance reduction techniques under various assumptions to obtain an $O(1/\epsilon^3)$ complexity in terms of SFO calls, see, e.g., Yurtsever et al. (2019); Hassani et al. (2020); Zhang et al. (2020b); Nazykov et al. (2024); Beznosikov et al. (2023); Weber & Sra (2022). In Appendix A, we summarize the most relevant results in Table 1, and provide a more detailed discussion.

Here, we propose Algorithm 4 and provide its convergence guarantees in Theorem 4.1 and Corollary 4.2. The idea of the algorithmic design is to integrate a variance reduction technique, STORM (Cutkosky & Orabona, 2019), into Algorithm 3.

**Theorem 4.1.** *Set* $\eta_t = \frac{1}{DT^{2/3}}$, $\gamma_t = \frac{1}{T^{2/3}}$, $\beta_{1,t} = 1 - \frac{1}{T^{1/3}}$, *for all* $t \geq 1$, *and the batch size* $m_t = m$, *for all* $t > 1$. *Let* $\mathbf{x}_a$ *be chosen uniformly at random from* $\{\mathbf{x}_t\}_{t=1}^T$. *Then, under Assumptions 2.1, 2.2, and 2.4 with* $p = 2$ *(i.e., bounded variance), Algorithm 4 satisfies* $\mathbb{E}[\mathcal{G}(\mathbf{x}_a)] = O\left(\frac{D\left(F(\mathbf{x}_1) - F(\mathbf{x}_*) + L^2 + \frac{\sigma^2}{m}\right)}{T^{1/3}} + \frac{D\sigma^2}{m_1}\right).$

**Corollary 4.2.** *In Theorem 4.1, set* $m_1 = T^{1/3}$ *and* $m_t = 1$, *for all* $t > 1$. *Then, Algorithm 4 satisfies* $\mathbb{E}[\mathcal{G}(\mathbf{x}_a)] = O\left(\frac{D\left(F(\mathbf{x}_1) - F(\mathbf{x}_*) + L^2 + \sigma^2\right)}{T^{1/3}}\right).$

Corollary 4.2 implies that the number of iterations is $T = O(1/\epsilon^3)$ to obtain an expected $\epsilon$-gap. Here, we recall that $m_t$ is the batch size to construct the stochastic gradient estimate $\bar{\mathbf{g}}_t$. Hence, the corresponding number of SFO calls indicated by Corollary 4.2 is $m_1 + (T - 1)m = O\left(\frac{1}{\epsilon^3}\right)$, which matches the best-known complexity bounds in the literature (see Table 1). However, we note that Algorithm 4 requires a large batch size only at initialization, and its amortized average batch size is at most 2, thereby avoiding the need for large batches throughout the iterations, unlike Yurtsever et al. (2019); Hassani et al. (2020). Furthermore, compared to Zhang et al. (2020b), our result does not require

---

**Algorithm 4** Stochastic FW with Variance Reduction

**Required:** Momentum parameters $\{\gamma_t\}$, step size $\{\eta_t\}$.
**Initialize:** $\mathbf{g}_0 = 0$, and $\mathbf{x}_1 \in \mathcal{C}$.
**for** $t = 1, 2, \dots$ **do**
  Sample $\Xi_t \sim \mathcal{D}$.
  Compute $\bar{\mathbf{g}}_t = \nabla f(\mathbf{x}_t; \Xi_t)$.
  Set $\mathbf{g}_t = (1 - \gamma_t)\mathbf{g}_{t-1} + \gamma_t \bar{\mathbf{g}}_t$
        $+ (1 - \gamma_t)(\bar{\mathbf{g}}_t - \mathbb{1}_{t \geq 2}\nabla f(\mathbf{x}_{t-1}; \Xi_t))$.
  Set $\hat{\mathbf{g}}_t = \frac{\beta_{1,t}}{(1-\gamma_t)}\mathbf{g}_t + \left(1 - \frac{\beta_{1,t}}{(1-\gamma_t)}\right)\bar{\mathbf{g}}_t$.
  Obtain $\mathbf{u}_t = \arg\min_{\mathbf{v} \in \mathcal{C}}\langle \mathbf{v}, \hat{\mathbf{g}}_t\rangle$.
  Update $\mathbf{x}_{t+1} = (1 - \eta_t)\mathbf{x}_t + \eta_t \mathbf{u}_t$.
**end for**

---

**Algorithm 5** Stochastic FW with Clipping

**Required:** Step size $\{\eta_t\}$, momentum parameters $\{\beta_{1,t}\}$, $\{\gamma_t\}$, and clipping parameter $M$.
**Initialize:** $\mathbf{g}_0 = 0$, and $\mathbf{x}_1 \in \mathcal{C}$.
**for** $t = 1, 2, \dots$ **do**
  Sample $\Xi_t \sim \mathcal{D}$.
  Get $\bar{\mathbf{g}}_t = \left(1 \wedge \frac{M}{\|\nabla f(\mathbf{x}_t; \Xi_t)\|}\right)\nabla f(\mathbf{x}_t; \Xi_t)$.
  Update $\mathbf{g}_t = (1 - \gamma_t)\mathbf{g}_{t-1} + \gamma_t \bar{\mathbf{g}}_t$.
  Set $\hat{\mathbf{g}}_t = \frac{\beta_{1,t}}{(1-\gamma_t)}\mathbf{g}_t + \left(1 - \frac{\beta_{1,t}}{(1-\gamma_t)}\right)\bar{\mathbf{g}}_t$.
  Compute $\mathbf{u}_t = \arg\min_{\mathbf{v} \in \mathcal{C}}\langle \mathbf{v}, \hat{\mathbf{g}}_t\rangle$.
  Update $\mathbf{x}_{t+1} = (1 - \eta_t)\mathbf{x}_t + \eta_t \mathbf{u}_t$.
**end for**

---

any assumptions on the Hessian or additional structural assumptions on the data distribution, while still maintaining a small average batch size. We also note that in the case of SignSGD and its variants (Karimireddy et al., 2019; Safaryan & Richtárik, 2021; Crawshaw et al., 2022), there is a line of work leveraging variance reduction techniques, see e.g., (Chzhen & Schechtman, 2023; Qin et al., 2023; Jiang et al., 2024) and the references therein. On the other hand, based on the connection between Lion and Stochastic FW that we have highlighted in an earlier section, we know that applying Algorithm 4 over the $\ell_\infty$-norm ball can yield an optimization dynamic with variance reduction that uses the sign of a stochastic gradient to update. By Corollary 4.2, this variant achieves an SFO complexity guarantee of $O(1/\epsilon^3)$, which parallels that in Jiang et al. (2024) and Arjevani et al. (2023) for finding an $\epsilon$-stationary point. However, we emphasize that our result of Stochastic FW is applicable to any convex and compact constraint set, not only limited to the $\ell_\infty$-norm ball.

## 4.2 Heavy-tailed regime

In this subsection, we shift gears to develop Stochastic FW methods that enjoy theoretical guarantees under $p$-th moment bounded noise, for any $p \in (1, 2]$. As discussed in the introduction, it is widely observed that stochastic gradients in deep learning—particularly in training large language models—are heavy-tailed (Zhang et al., 2020a; Gurbuzbala-

ban et al., 2021; Hodgkinson & Mahoney, 2021; Kunstner et al., 2023; Ahn et al., 2024). Therefore, the commonly used bounded variance assumption may not be appropriate for designing and analyzing stochastic optimization algorithms. To mitigate the issue of heavy-tailed noise, *clipping* has been used to establish several nice results in SGD recently (Zhang et al., 2020a; Gorbunov et al., 2020; Mai & Johansson, 2021; Cutkosky & Mehta, 2021; Nguyen et al., 2023; Hübler et al., 2024; Schaipp et al., 2025), as well as in adaptive methods such as Adagrad and Adam (Chezhegov et al., 2025; Li & Liu, 2023). However, very little prior research has focused on FW-like methods under heavy-tailed noise, and the work by Tang et al. (2022) is the only one we are aware of. That said, unlike our work and the aforementioned related results, which assume bounded $p$-th moment noise (Assumption 2.4), Tang et al. (2022) adopt a different set of assumptions on the stochastic noise and assume convexity of the function, and hence our results in the following are not directly comparable to them (more detail in Appendix A). We also note that their algorithm requires a large batch size, whereas ours allows the batch size to be one.

We propose Algorithm 5, which can be viewed as integrating the clipping technique from Algorithm 3. We note that from the idea of *Lions and Muons as Stochastic FWs*, one can obtain a variant of Lion and Muon with clipping from Algorithm 5. Specifically, the new variant of Lion (which we call LION+) shares the same steps as Lion, except that Line 5 in Algorithm 1 is replaced with $\bar{\mathbf{g}}_t = \left(1 \wedge \frac{M}{\|\nabla f(\mathbf{x}_t; \Xi_t)\|}\right) \nabla f(\mathbf{x}_t; \Xi_t)$, where $M > 0$ is the parameter of clipping. Similarly, the new variant of Muon with clipping, denoted MUON+, follows the same updates as Algorithm 2, except that Line 5 is replaced with a clipped stochastic gradient. Due to the space limit, we defer the full algorithmic descriptions of LION+ and MUON+ to Appendix C. Theorem 4.3 below provides the convergence rate guarantee of the proposed Stochastic FW with clipping.

**Theorem 4.3.** *Suppose Assumptions 2.3-a, 2.4, and 2.5 hold. Set $\gamma_t = \gamma = T^{\frac{-p}{3p-2}}$, $\beta_{1,t} = \beta = (1 - \gamma)(1 - T^{\frac{-p}{3p-2}})$, $M = \frac{\sigma}{\gamma^{1/p}} \vee 2G$, and $\eta_t = \eta = \frac{1}{\sqrt{LT}D} \wedge \frac{\gamma}{\beta}\frac{1}{D} \wedge \frac{\sqrt{\gamma}}{D\sqrt{\beta TL}} \wedge \frac{1-\gamma}{20\gamma DTM\log\frac{4T}{\delta}} \wedge \frac{1}{2TD(1-\frac{\beta}{1-\gamma})M(1+\gamma)}$. Then, with probability at least $1 - \delta$, Algorithm 5 has $\frac{1}{T}\sum_{t=1}^{T}\mathcal{G}(\mathbf{x}_t) = O\left(\frac{\log\frac{T}{\delta}}{T^{\frac{p-1}{3p-2}}}\right)$, where $p \in (1,2]$.*

We note that the convergence rate $O\left(\log\left(\frac{T}{\delta}\right) T^{\frac{1-p}{3p-2}}\right)$ for the Frank-Wolfe gap achieved by Algorithm 5 parallels those results for SGD with clipping under heavy-tailed noise in the literature (Zhang et al., 2020a; Cutkosky & Mehta, 2021; Hübler et al., 2024), which converges to an $\epsilon$ expected gradient norm. Notably, the result of Theorem 4.3 is estab-

---

**Algorithm 6** Stochastic FW with Clipping and Variance Reduction

**Required:** Step size $\{\eta_t\}$, momentum parameters $\{\beta_{1,t}\}$, $\{\gamma_t\}$, and clipping parameter $M$.
**Initialize:** $\mathbf{g}_0 = 0$, and $\mathbf{x}_1 \in \mathcal{C}$.
**for** $t = 1, 2, \ldots$ **do**
  Sample $\Xi_t \sim \mathcal{D}$.
  Set $\bar{\mathbf{g}}_t = \left(1 \wedge \frac{M}{\|\nabla f(\mathbf{x}_t;\Xi_t)\|}\right) \nabla f(\mathbf{x}_t; \Xi_t)$.
  Update $\mathbf{g}_t = (1 - \gamma_t)\mathbf{g}_{t-1} + \gamma_t \bar{\mathbf{g}}_t$
    $+ (1 - \gamma_t)\mathbb{1}_{t \geq 2}\left(\nabla f(\mathbf{x}_t; \Xi_t) - \nabla f(\mathbf{x}_{t-1}; \Xi_t)\right)$.
  Set $\hat{\mathbf{g}}_t = \frac{\beta_{1,t}}{(1-\gamma_t)}\mathbf{g}_t + \left(1 - \frac{\beta_{1,t}}{(1-\gamma_t)}\right) \bar{\mathbf{g}}_t$.
  Compute $\mathbf{u}_t = \arg\min_{\mathbf{v}\in\mathcal{C}} \langle\mathbf{v}, \hat{\mathbf{g}}_t\rangle$.
  Update $\mathbf{x}_{t+1} = (1 - \eta_t)\mathbf{x}_t + \eta_t \mathbf{u}_t$.
**end for**

---

lished even when the batch size of stochastic gradients is fixed to 1 at each iteration.

We also propose another algorithm that incorporates both the clipping operation and variance reduction, i.e., Algorithm 6. Theorem 4.4 shows the convergence rate of the proposed method, and we note that, similarly to Theorem 4.3, this guarantee holds even when the batch size of the stochastic gradients is 1 at each iteration.

In particular, Theorem 4.4 shows an improved complexity, compared to Theorem 4.3. We also note that Theorem 4.4 for the proposed Stochastic FW can be seen as a counterpart to a notable result by Liu et al. (2023) on SGD for non-convex stochastic optimization under heavy-tailed noise, where the authors establish the same rate for obtaining an $\epsilon$-small expected gradient norm via normalized SGD with clipping and variance reduction.

**Theorem 4.4.** *Suppose Assumptions 2.3-a, 2.3-b, 2.4, and 2.5 hold. Set $\gamma_t = \gamma = T^{\frac{-p}{2p-1}}$, $\beta_{1,t} = \beta = (1 - \gamma)\left(1 - T^{\frac{-p}{2p-1}}\right)$, $M = \frac{\sigma}{\gamma^{1/p}} \vee 2G$, and $\eta_t = \eta = \frac{1}{\sqrt{LT}D} \wedge \frac{\gamma}{\beta}\frac{1}{D} \wedge \frac{\gamma^{1/4}}{D\sqrt{9TL\beta\log\frac{3T}{\delta}}} \wedge \frac{1-\gamma}{20\gamma DTM\beta\log\frac{4T}{\delta}} \wedge \frac{1}{2TD(1-\frac{\beta}{1-\gamma})M(1+\gamma)}$. Then, with probability at least $1 - \delta$, Algorithm 6 has $\frac{1}{T}\sum_{t=1}^{T}\mathcal{G}(\mathbf{x}_t) = O\left(\frac{\log\frac{T}{\delta}}{T^{\frac{p-1}{2p-1}}}\right)$, where $p \in (1,2]$.*

To our knowledge, both Theorem 4.3 and Theorem 4.4 are the first results of this kind for Stochastic FW-type methods under heavy-tailed noise in *non-convex* optimization. Theorem 4.4 shows that with an additional assumption on the problem structure (i.e., Assumption 2.3-b), the rate can be improved to $\tilde{O}\left(T^{\frac{1-p}{2p-1}}\right)$ from $\tilde{O}\left(T^{\frac{1-p}{3p-2}}\right)$.

## 5 Experiments

In this section, we evaluate the performance of LION+ and MUON+ through numerical experiments. Specifically, we train a nanoGPT [1] on the Shakespeare dataset. We compare the performance of the proposed algorithms against Lion (Chen et al., 2024b) and Muon (Jordan et al., 2024). In the cases of Muon and MUON+, we use the efficient Newton-Schulz iteration, proposed by Bernstein & Newhouse (2024) for the orthogonalization step, and we use AdamW (Loshchilov & Hutter, 2018) to optimize the network's one-dimensional (i.e., vector) parameters. To ensure a fair comparison, we keep AdamW's hyperparameters fixed. For all the evaluation algorithms, we employ a cosine learning rate scheduler. The dropout rate is set to 0.2. We repeat all the experiments with five different seed values and report the average. We refer the reader to Appendix D for the detailed training configuration and hyperparameter tuning for each of the comparison algorithms. All experiments are conducted on one NVIDIA A100 GPU.

**Results:** Figure 1 shows the training and validation loss curves averaged over five runs and evaluated every 10 and 50 steps, respectively. The results indicate that methods with gradient clipping consistently obtain a lower validation loss compared to their unclipped counterparts. Specifically, by comparing the number of steps to reach a validation loss below 1.47, LION+ requires 2950 steps, approximately 19.18% less steps than Lion (3650 steps). Similarly, MUON+ achieves the target validation loss in 4000 steps, demonstrating an approximately 5.88% reduction in steps compared to Muon (4250 steps).

Additional experimental results are provided in Appendix E.

## 6 Discussion and Future Directions

In this work, we have presented a generalized formulation of Stochastic Frank-Wolfe algorithms, and we have established a comprehensive set of previously missing theoretical results in the non-convex setting, including smooth functions with bounded variance (Theorem 3.3), smooth stochastic functions with bounded variance and variance reduction (Theorem 4), and with clipping in the presence of more heavy-tailed noise assumptions (Theorems 4.3 and 4.4). In particular, we have demonstrated that the proposed Stochastic Frank-Wolfe unifies Lion and Muon under suitable norm constraints, while also extending to *any* convex constraint set, which in turn broadens the scope of algorithms to which its convergence guarantees apply. In the case of norm-ball constraints, as in Lion or Muon, we have shown that our convergence guarantees in terms of the Frank-Wolfe gap translate into convergence to a KKT point of the constrained

[1] https://github.com/karpathy/nanoGPT

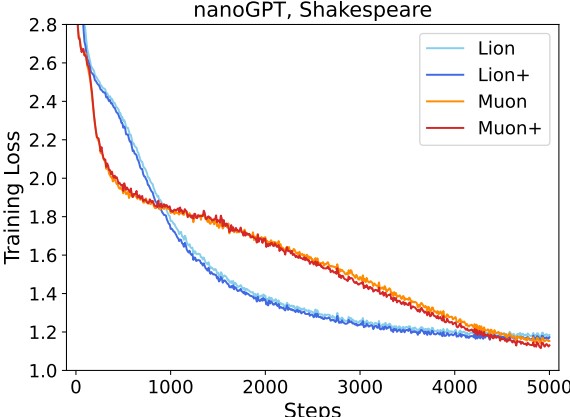

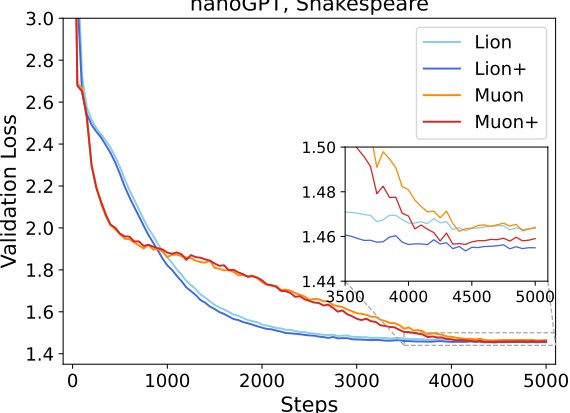

*Figure 1.* Loss curves for nanoGPT training on the Shakespeare dataset. We plotted the training and validation loss per 10 and 50 steps, respectively. The results are averaged across five seed values.

problem. Furthermore, we have demonstrated that by incorporating a variance reduction term, we can obtain an improved convergence rate in terms of stochastic gradient evaluations. Finally, we have considered the heavy-tailed noise case, where the stochastic gradients satisfy a weaker assumption than bounded variance, and we have provided variants with gradient clipping and variance reduction that satisfy high-probability bounds. Our experimental results validated the theoretical guarantees by demonstrating the improved convergence of the enhanced variants over the baseline ones.

It still remains an open problem whether the heavy-tailed analysis can be modified to account for the noise level $\sigma$ in the bound. More precisely, in the absence of noise, the high-probability bounds we have obtained do not adapt to an improved convergence rate as expected in the noiseless setting, unlike the in-expectation bounds we have provided for the bounded variance case. We also note that it is an interesting question whether the convergence analysis of these algorithms can be established under more general norms in Banach spaces. Importantly, our theoretical analysis mea-

sures the constraint set diameter using Hilbert-space norms. Since norms are equivalent only up to dimension-dependent constants in finite-dimensional spaces, incorporating a constraint set defined by a different norm could introduce an explicit dependence on the dimension. Another theoretical limitation is that our Stochastic Frank-Wolfe analysis does not extend to the Lion and Muon variants *without* weight decay, since Frank-Wolfe methods inherently address constrained optimization problems. A natural next step is to investigate how varying the constraint set can yield new algorithmic variants with potentially improved dynamics and performance, which we leave for future work.

## Acknowledgements

The authors thank the anonymous reviewers for their feedback. The authors also appreciate the support from NSF CCF-2403392, as well as the Google Gemma Academic Program and Google Cloud Credits.

## Impact Statement

This paper presents work whose goal is to advance the field of Machine Learning. There are many potential societal consequences of our work, none of which we feel must be specifically highlighted here.

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

# A    More Related Works

**Lion.** The Lion optimizer, proposed by Chen et al. (2024b), is a recent algorithm discovered via program search that has been experimentally shown to have superior generalization properties on various tasks compared to AdamW, which in turns has drawn further research attention in improving the algorithm and theoretical foundation (Chen et al., 2024a; Liu et al., 2024; Dong et al., 2024; Kosson et al., 2024b;a; Liang et al., 2024; 2025; Zhao et al., 2025). Notably, Chen et al. (2024a) introduce a general family of Lion-$\mathcal{K}$ algorithms by developing a Lyapunov function for the optimization dynamics and show that Lion solves an $\ell_\infty$-norm constrained optimization problem. Liu et al. (2024) present an adaptation of Lion within the distributed optimization setting, allowing efficient reduction of communication costs. Dong et al. (2024) provide a convergence analysis of Lion to a KKT point of the constrained problem and extend the analysis to stationary points in the unconstrained case. Kosson et al. (2024b) analyze the influence of the interplay between weight decay and learning rate in the update dynamics of various optimizers, including Lion. A subsequent work by the same authors proposes two Lion variants that scale the update size and adjust the hyperparameter choices to be more comparable to those of AdamW (Kosson et al., 2024a). Liang et al. (2024) present a conversion of Lion to an online subspace descent algorithm, a process that enables memory efficiency by utilizing the low-rank structure of the gradients and restricting the update states to a dynamically changing subspace. Another variant that accelerates the loss decrease in momentum-based methods by inspecting the direction consistency between the current gradient and the update step was developed by Liang et al. (2025), who present C-Lion, a cautious version of Lion. Zhao et al. (2025) conduct an empirical analysis of hyperparameter stability and performance of Lion alongside other optimization algorithms and explore their equivalence in terms of optimal performance. More recently, Yuan et al. (2025) develop a preconditioned optimization framework by incorporating variance reduction into various adaptive gradient methods, with one variant based on Lion. While their setting offers interesting insights, we note that, in contrast to this work, it does not incorporate the weight decay term used in Lion, and convergence guarantees for the Lion variant are not included. Additionally, their analysis requires a stronger assumption of positive-definite preconditioners, a condition that depends on the evolving algorithmic iterates.

**Muon and Shampoo.** Muon (Jordan et al., 2024), on the other hand, is a preconditioned gradient method. The design of Muon was motivated from improving Shampoo (Gupta et al., 2018), which aims to maintain necessary second-order information while being efficient in optimization over tensor spaces. Shampoo regained prominence after winning the external tuning track at the 2024 AlgoPerf: Training Algorithms competition (Dahl et al. (2023); Vyas et al. (2025); Morwani et al. (2025)). Anil et al. (2021) further extended Shampoo to obtain a scalable version, capable of handling large model architectures. The work of Vyas et al. (2025) establishes an equivalence between Shampoo and Adafactor applied in the eigenbasis defined by the Shampoo preconditioner. Extending this observation, they propose SOAP, which allows applying Adam in Shampoo's eigenspace. Notably, a recent work by Morwani et al. (2025) showed an interesting connection between Shampoo's preconditioner and the optimal Kronecker product approximation of certain matrices, allowing a more precise understanding of its dynamics. Building on Shampoo, Jordan et al. (2024) propose Muon, an update rule that can be interpreted as a variant of Shampoo without the use of preconditioner accumulators. Bernstein & Newhouse (2024) demonstrate that Muon can also be viewed as a steepest descent method under the spectral norm and provide a more efficient Newton-Schulz iteration scheme to perform approximate SVD. A recent work by Pethick et al. (2025) showcased Muon without the additional Nesterov-momentum step as an instance of one of their proposed methods labeled Unconstrained Stochastic Conditional Gradient Method, with a guarantee on the expected gradient norm, and also recovered Muon with weight decay from a constrained variant, with a guarantee on the expected Frank-Wolfe gap. Their work further establishes a connection between the linear minimization oracle in Stochastic Frank-Wolfe and other norm-constrained optimizers, offering a valuable step toward unifying these approaches. While offering important contributions, their constrained variant does not incorporate the momentum extrapolation step, and as a result, does not recover Lion or the Nesterov-momentum variant of Muon. Since then, several works have analyzed the convergence properties of Muon in the absence of a weight-decay term (Li & Hong, 2025; Shen et al., 2025; An et al., 2026; Kovalev, 2025; Riabinin et al., 2025). Notably, Chen et al. (2025) extend this line of work by studying the convergence of Muon with weight decay within the Lion-$\mathcal{K}$ framework. Muon with weight decay has been shown experimentally to outperform vanilla Muon in the over-train regime (Liu et al., 2025a). We emphasize that, while convergence guarantees for Muon with weight decay can be recovered from the proposed Stochastic Frank-Wolfe in this paper, our results are established for the general Stochastic Frank-Wolfe and therefore extend to a broad class of algorithms. Moreover, new variants of Muon can be derived within our proposed methods, for which we also recover the corresponding convergence guarantees. Several recent variants of Muon have additionally been proposed, including Ma et al. (2025); Liu et al. (2025b); Lau et al. (2025); Huang et al. (2025); He et al. (2025); Si et al. (2025); Liu et al. (2025a).

| Method | Assumptions | Batch | LMO | SFO |
|---|---|---|---|---|
| SVFW-S
Reddi et al. (2016) | $F$ is $L$-smooth
$\|\nabla f(\mathbf{x};\xi)\| \leq G$ | $O\left(1/\epsilon^{4/3}\right)$ | $O\left(1/\epsilon^2\right)$ | $O\left(1/\epsilon^{10/3}\right)$ |
| SPIDER-FW
Yurtsever et al. (2019) | $f(\cdot;\xi)$ is $L$-smooth
$\mathbb{E}\left[\|\nabla f(\mathbf{x};\xi) - \nabla F(\mathbf{x})\|^2\right] \leq \sigma^2$ | $O\left(1/\epsilon\right)$ | $O\left(1/\epsilon^2\right)$ | $O\left(1/\epsilon^3\right)$ |
| SFW++
Hassani et al. (2020) | $\|f(\mathbf{x};\xi)\| \leq B$
$\|\nabla f(\mathbf{x};\xi)\| \leq G$
$\mathbb{E}\left[\|\nabla \log p(\mathbf{x};\xi)\|^4\right] \leq G_p^4$
$\|\nabla^2 f(\mathbf{x};\xi)\| \leq L_f$
$\mathbb{E}\left[\|\nabla^2 \log p(\mathbf{x};\xi)\|^2\right] \leq L_p^2$
$\nabla^2 f(\mathbf{x};\xi)$ is $L_{2,f}$-Lipschitz
$\nabla^2 \log p(\mathbf{x};\xi)$ is $L_{2,p}$-Lipschitz | $O\left(1/\epsilon\right)$ | $O\left(1/\epsilon^2\right)$ | $O\left(1/\epsilon^3\right)$ |
| 1-SFW
Zhang et al. (2020b) | $\|f(\mathbf{x};\xi)\| \leq B$
$\|\nabla f(\mathbf{x};\xi)\| \leq G$
$\mathbb{E}\left[\|\nabla \log p(\mathbf{x};\xi)\|^4\right] \leq G_p^4$
$\|\nabla^2 f(\mathbf{x};\xi)\| \leq L_f$
$\mathbb{E}\left[\|\nabla^2 \log p(\mathbf{x};\xi)\|^2\right] \leq L_p^2$ | 1 | $O\left(1/\epsilon^3\right)$ | $O\left(1/\epsilon^3\right)$ |

*Table 1.* Convergence guarantees of stochastic Frank-Wolfe methods with variance reduction for non-convex optimization in the stochastic setting with the bounded variance assumption, i.e., $\mathbb{E}_\xi\left[\|\nabla f(\mathbf{x},\xi) - \nabla F(\mathbf{x})\|^2\right] \leq \sigma^2$. Here, $p(\mathbf{x};\xi)$ is the distribution from which the stochastic gradient is sampled. We refer the reader to the references therein for details.

**SignSGD with variance reduction.** Optimization methods that use the sign of the gradient at each iteration have gained interest due to their communication efficiency properties. While introduced in Seide et al. (2014) as a gradient compression scheme, sign stochastic gradient descent (SignSGD) was first rigorously analyzed by Bernstein et al. (2018), who showed that SignSGD with large batch sizes achieves $O(1/\epsilon^4)$ complexity for non-convex optimization problems. Following this, a growing body of work has been developed to extend sign-based methods (Bernstein et al., 2019; Karimireddy et al., 2019; Chen et al., 2019; Safaryan & Richtárik, 2021; Crawshaw et al., 2022; Sun et al., 2023; Xiang & Su, 2023; Jin et al., 2025). In the direction of variance reduction, Chzhen & Schechtman (2023) introduce SignSVRG, which incorporates the variance reduction ideas from SVRG (Johnson & Zhang, 2013) into SignSGD, and establish a convergence rate of $O(m/\epsilon^2)$ in the finite-sum setting, where $m$ is the number of component functions. Building on similar ideas from SVRG and using random reshuffling, Qin et al. (2023) propose SignRVR and its momentum variant, SignRVM, both achieving a convergence rate of $O(m/\epsilon^4)$ in the finite-sum setting. More recently, Jiang et al. (2024) utilize the variance reduction technique of STORM (Cutkosky & Orabona, 2019) in SignSGD to develop SSVR, achieving an improved convergence rate of $O(1/\epsilon^3)$ in the general stochastic setting. We note all of the aforementioned complexity results of SignSGD and its varaints concern finding an $\epsilon$-small expected gradient norm.

**SFW with variance reduction.** Variance reduction techniques have been adopted in Stochastic Frank-Wolfe (SFW) algorithms to produce various algorithmic variants for the *non-convex stochastic setting*. The work of Reddi et al. (2016) is the first to propose a variance-reduced SFW algorithm (SVFW-S) that achieves $O\left(1/\epsilon^{10/3}\right)$ SFO and $O\left(1/\epsilon^2\right)$ LMO complexities to obtain an $\epsilon$-approximate solution, improving the $O\left(1/\epsilon^4\right)$ SFO and $O\left(1/\epsilon^2\right)$ LMO rates of classical SFW. Yurtsever et al. (2019) introduce a similar variant (SPIDER-FW) by combining the ideas of SPIDER (Fang et al., 2018) and SFW, and show a superior complexity rate of $O\left(1/\epsilon^3\right)$ for SFO, while preserving the same LMO complexity. Hassani et al. (2020) propose another variant (SFW++), which applies variance reduction by using an unbiased estimator of the gradient difference and demonstrates an SFO complexity of $O\left(1/\epsilon^3\right)$. An alternative variant (1-SFW) with exactly one call to the SFO per iteration was presented in the work of Zhang et al. (2020b). However, the proposed algorithm exhibits worse computational complexities of $O\left(1/\epsilon^3\right)$ for SFO and LMO. Building on the ideas of SPIDER, Weber & Sra (2022) suggest another variance-reduced variant (SPIDER-RFW) for the Riemannian setting, demonstrating an SFO complexity of $O\left(1/\epsilon^3\right)$. Nazykov et al. (2024) propose a unified framework of SFW methods, leading to the development of various new algorithms, including variance reduction alternatives. Some other works have explored the use of variance reduction in stochastic Frank-Wolfe variants for the *non-convex finite-sum setting* (Reddi et al., 2016; Yurtsever et al., 2019; Qu et al.,

2018; Beznosikov et al., 2023).

**Heavy-tailed noise.** Stochastic gradients in deep learning are widely known to exhibit heavy-tailed noise (Zhang et al., 2020a; Gurbuzbalaban et al., 2021; Hodgkinson & Mahoney, 2021; Kunstner et al., 2023; Ahn et al., 2024). The heavy-tailed noise condition in SGD was first explored in the work of Zhang et al. (2020a), where the authors provide convergence guarantees in expectation for SGD with clipping. Gorbunov et al. (2020) show high probability bounds for SGD under the bounded variance assumption in the smooth convex case. Subsequent works by Gorbunov et al. (2022) and Gorbunov et al. (2024) extend the applicability of gradient clipping and moment-based analysis to variational inequality problems. Another work by Mai & Johansson (2021) studies the stability and convergence properties of clipped SGD for convex and weakly convex functions with rapidly growing subgradients. Cutkosky & Mehta (2021) demonstrate that combining gradient clipping, momentum, and normalized gradient descent leads to high-probability convergence under the $p_{th}$ bounded moment assumption for general non-convex functions. Adaptive gradient methods have also been studied under the same assumption. Specifically, Li & Liu (2023) derive high-probability convergence and generalization bounds for clipped SGD and its momentum and adaptive variants in the non-convex setting, and Chezhegov et al. (2025) similarly provide convergence guarantees for clipped Adam and AdaGrad under heavy-tailed noise in both convex and non-convex regimes. Nguyen et al. (2023) present an alternative analysis method for showing high-probability convergence in various clipped gradient methods and prove convergence gurantees for the convex and nonconvex setting. Later, Hübler et al. (2024) provide an in-expectation convergence of normalized SGD that does not require the specification of any algorithmic parameters. A noteworthy insight is uncovered by the recent work of Schaipp et al. (2025), which reveals that clipped gradient methods implicitly perform a median estimation over iterations.

# B Proofs of the theoretical results in Section 3

## B.1 Proof of Theorem 3.1

**Theorem B.1.** *(Lion as Stochastic FW) Lion (Algorithm 1) is an instance of a Stochastic Frank-Wolfe (Algorithm 3) when using parameters $\beta_{1,t} = \beta_1$, $\gamma_t = 1 - \beta_2$, $\eta_t^{\text{Alg }3} = \lambda \eta_t^{\text{Alg }1}$, for all t, setting $\mathcal{C} = \{\mathbf{v} \in \mathbb{R}^n : \|\mathbf{v}\|_\infty \leq \frac{1}{\lambda}\}$, and letting $\langle \cdot, \cdot \rangle$ denote the Euclidean inner product.*

*Proof.* We show by induction that for all $t \geq 0$ the updates obtained by Algorithm 1 are equivalent to the updates of an instance of Algorithm 3. Let $\mathbf{x}_t^{\text{Alg }1}$, $\eta_t^{\text{Alg }1}$, $\bar{\mathbf{g}}_t^{\text{Alg }1}$, and $\mathbf{x}_t^{\text{Alg }3}$, $\eta_t^{\text{Alg }3}$, and $\bar{\mathbf{g}}_t^{\text{Alg }3}$, denote the iterates, step-sizes and stochastic gradients used at step $t$ of Algorithm 1 and Algorithm 3, respectively. Then, by setting $\beta_{1,t} \leftarrow \beta_1$, $\gamma_t \leftarrow 1 - \beta_2$, $\eta_t^{\text{Alg }3} \leftarrow \lambda \eta_t^{\text{Alg }1}$, for all $t$ and letting $\mathcal{C} = \{\mathbf{v} : \|\mathbf{v}\|_\infty \leq \frac{1}{\lambda}\}$, we show that the following equations are maintained for all $t \geq 0$.

$$\mathbf{x}_{t+1}^{\text{Alg }1} = \mathbf{x}_{t+1}^{\text{Alg }3} \tag{4}$$

$$\mathbf{m}_t = \mathbf{g}_t \tag{5}$$

$$\mathbf{c}_{t+1} = \hat{\mathbf{g}}_{t+1} \tag{6}$$

Note that the objects on the left hand-side of the equalities correspond to Algorithm 1 and the objects on the right hand-side correspond to Algorithm 3. For $t = 0$, we have that by initialization of the algorithms $\mathbf{m}_0 = \mathbf{g}_0 = 0$ and $\mathbf{x}_1^{\text{Alg }1} = \mathbf{x}_1^{\text{Alg }3}$. Furthermore, we observe that for all $t \geq 0$ we have (4),(5) $\Rightarrow$ (6). That is, because if $\mathbf{m}_t = \mathbf{g}_t$, $\mathbf{x}_{t+1}^{\text{Alg }1} = \mathbf{x}_{t+1}^{\text{Alg }3}$ and using

the parameter choices from before, we have

$$
\begin{aligned}
\mathbf{c}_{t+1} &= \beta_1 \mathbf{m}_t + (1 - \beta_1) \bar{\mathbf{g}}_{t+1}^{\text{Alg } 1} \\
&\overset{(i)}{=} \beta_1 \mathbf{g}_t + (1 - \beta_1) \nabla f \left( \mathbf{x}_{t+1}^{\text{Alg } 1}; \Xi_{t+1} \right) \\
&\overset{(ii)}{=} \beta_1 \mathbf{g}_t + (1 - \beta_1) \nabla f \left( \mathbf{x}_{t+1}^{\text{Alg } 3}; \Xi_{t+1} \right) \\
&= \beta_1 \mathbf{g}_t + (1 - \beta_1) \bar{\mathbf{g}}_{t+1}^{\text{Alg } 3} \\
&= \beta_1 \mathbf{g}_t + \frac{\beta_1}{\beta_2}(1 - \beta_2) \bar{\mathbf{g}}_{t+1}^{\text{Alg } 3} + \left( 1 - \frac{\beta_1}{\beta_2} \right) \bar{\mathbf{g}}_{t+1}^{\text{Alg } 3} \\
&= \frac{\beta_1}{\beta_2} \left( \beta_2 \mathbf{g}_t + (1 - \beta_2) \bar{\mathbf{g}}_{t+1}^{\text{Alg } 3} \right) + \left( 1 - \frac{\beta_1}{\beta_2} \right) \bar{\mathbf{g}}_{t+1}^{\text{Alg } 3} \\
&= \frac{\beta_1}{\beta_2} \mathbf{g}_{t+1} + \left( 1 - \frac{\beta_1}{\beta_2} \right) \bar{\mathbf{g}}_{t+1}^{\text{Alg } 3} \\
&= \hat{\mathbf{g}}_{t+1},
\end{aligned}
$$

where (i) is by $\mathbf{m}_t = \mathbf{g}_t$ and (ii) is by $\mathbf{x}_{t+1}^{\text{Alg } 1} = \mathbf{x}_{t+1}^{\text{Alg } 3}$.

Now, we can show that (4) and (5) hold for $t > 0$ using induction. Assume that they hold up to some $t \geq 0$. Then, it follows that (6) also holds up to $t \geq 0$ and we have

$$
\mathbf{x}_{t+2}^{\text{Alg } 1} = \mathbf{x}_{t+1}^{\text{Alg } 1} - \eta_{t+1}^{\text{Alg } 1} \left( \text{sign}(\mathbf{c}_{t+1}) + \lambda \mathbf{x}_{t+1}^{\text{Alg } 1} \right) = \mathbf{x}_{t+1}^{\text{Alg } 3} + \eta_{t+1}^{\text{Alg } 3} \left( \mathbf{u}_{t+1} - \mathbf{x}_{t+1}^{\text{Alg } 3} \right) = \mathbf{x}_{t+2}^{\text{Alg } 3},
$$

and

$$
\begin{aligned}
\mathbf{m}_{t+1} &= \beta_2 \mathbf{m}_t + (1 - \beta_2) \bar{\mathbf{g}}_{t+1}^{\text{Alg } 1} \\
&= \beta_2 \mathbf{m}_t + (1 - \beta_2) \nabla f \left( \mathbf{x}_{t+1}^{\text{Alg } 1}; \Xi_{t+1} \right) \\
&= \beta_2 \mathbf{g}_t + (1 - \beta_2) \nabla f \left( \mathbf{x}_{t+1}^{\text{Alg } 3}; \Xi_{t+1} \right) \\
&= \beta_2 \mathbf{g}_t + (1 - \beta_2) \bar{\mathbf{g}}_{t+1}^{\text{Alg } 3} \\
&= \mathbf{g}_{t+1}.
\end{aligned}
$$

Finally, since (4) and (5) hold for $t + 1$, we obtain that (6) also holds for $t + 1$. This completes the proof. $\square$

## B.2   Proof of Theorem 3.2

**Theorem B.2.** *(Muon as Stochastic FW) Muon (Algorithm 2) is an instance of a Stochastic Frank-Wolfe (Algorithm 3) when using the parameters $\beta_{1,t} = \mu$, $\gamma_t = 1 - \mu$, $\eta_t^{\text{Alg } 3} = \lambda \eta_t^{\text{Alg } 2}$, for all t, setting $\mathcal{C} = \{ \mathbf{A} \in \mathbb{R}^{m \times n} : \|\mathbf{A}\|_2 \leq \frac{1}{\lambda} \}$, and letting $\langle \cdot, \cdot \rangle$ denote the Frobenius inner product.*

*Proof.* We show by induction that for all $t$ the updates obtained by Algorithm 2 are equivalent to the updates of an instance of Algorithm 3. Let $\eta_t^{\text{Alg } 2}$ and $\eta_t^{\text{Alg } 3}$ denote the step-sizes used at step $t$ of Algorithm 2 and Algorithm 3, respectively. Then, by setting $\beta_{1,t} \leftarrow \mu$, $\gamma_t \leftarrow 1 - \mu$, $\eta_t^{\text{Alg } 3} \leftarrow \lambda \eta_t^{\text{Alg } 2}$, for all $t$, $\mathcal{C} = \{ \mathbf{A} \in \mathbb{R}^{m \times n} : \|\mathbf{A}\|_2 \leq \frac{1}{\lambda} \}$ and letting $\langle \cdot, \cdot \rangle$ denote the Frobenius inner product, we show that the following equations are maintained for all $t \geq 0$.

$$
\mathbf{B}_t = \frac{\mathbf{g}_t}{1 - \mu} \tag{7}
$$

$$
\mathbf{O}_t = -\lambda \mathbf{u}_t \tag{8}
$$

$$
\mathbf{X}_{t+1} = \mathbf{x}_{t+1} \tag{9}
$$

Note that with these parameter choices $\hat{\mathbf{g}}_t = \mathbf{g}_t$, for all $t \geq 0$ in Algorithm 3. Here, the objects on the left hand-side of the equality correspond to Algorithm 2 and the objects on the right hand-side correspond to Algorithm 3. For $t = 0$, we have

that by initialization of the algorithms $\mathbf{B}_0 = \frac{\mathbf{g}_0}{1-\mu} = 0$ and $\mathbf{X}_1 = \mathbf{x}_1$. Furthermore, for any $t \geq 0$, we have (7) $\Rightarrow$ (8). That is, because if $\mathbf{B}_t = \frac{\mathbf{g}_t}{1-\mu}$, then

$$
\begin{aligned}
\mathbf{O}_t &= \arg \min_{\mathbf{A} \in \mathcal{O}_{m \times n}} \|\mathbf{A} - \mathbf{B}_t\|_F \\
&= -\lambda \cdot \arg \min_{\|\mathbf{A}\|_2 \leq \frac{1}{\lambda}} \langle \mathbf{A}, \mathbf{B}_t \rangle \\
&= -\lambda \cdot \arg \min_{\|\mathbf{A}\|_2 \leq \frac{1}{\lambda}} \langle \mathbf{A}, \frac{\mathbf{g}_t}{1-\mu} \rangle \\
&= -\lambda \cdot \arg \min_{\|\mathbf{A}\|_2 \leq \frac{1}{\lambda}} \langle \mathbf{A}, \mathbf{g}_t \rangle \\
&= -\lambda \mathbf{u}_t.
\end{aligned}
$$

Now, we can show that (7) and (9) hold for $t > 0$ by induction. Assume that they hold up to some $t \geq 0$. Then, we have

$$
\begin{aligned}
\mathbf{B}_{t+1} &= \mu \mathbf{B}_t + \mathbf{G}_{t+1} \\
&= \mu \mathbf{B}_t + \nabla f(\mathbf{X}_{t+1}; \Xi_{t+1}) \\
&= \mu \frac{\mathbf{g}_t}{1-\mu} + \nabla f(\mathbf{x}_{t+1}; \Xi_{t+1}) \\
&= \frac{\mathbf{g}_{t+1}}{1-\mu}.
\end{aligned}
$$

Therefore, we also have that (8) holds for $t + 1$, i.e. $\mathbf{O}_{t+1} = -\lambda \mathbf{u}_{t+1}$. Finally,

$$
\begin{aligned}
\mathbf{X}_{t+2} &= \mathbf{X}_{t+1} - \eta_{t+1}^{\text{Alg 2}} (\mathbf{O}_{t+1} + \lambda \mathbf{X}_{t+1}) \\
&= \mathbf{x}_{t+1} - \frac{\eta_{t+1}^{\text{Alg 3}}}{\lambda} (-\lambda \mathbf{u}_{t+1} + \lambda \mathbf{x}_{t+1}) \\
&= \left(1 - \eta_{t+1}^{\text{Alg 3}}\right) \mathbf{x}_{t+1} + \eta_{t+1} \mathbf{u}_{t+1},
\end{aligned}
$$

which implies that (9) holds for $t + 1$. This completes the proof. $\qquad\square$

### B.3  Muon with Nesterov-momentum

In this section, we show that Muon with Nesterov momentum (Algorithm 7) is also an instance of our Stochastic Frank-Wolfe formulation (Algorithm 3).

---

**Algorithm 7** Muon with Nesterov momentum (Jordan et al., 2024)

---

**Required:** Momentum parameter $\mu$, step size $\{\eta_t\}$, weight decay parameter $\lambda$.
**Initialize:** $\mathbf{B}_0 = 0$ and $\mathbf{X}_1 \in \mathcal{C}$.
**for** $t = 1, 2, \dots$ **do**
    Sample $\Xi_t \sim \mathcal{D}$.
    Compute $\mathbf{G}_t = \nabla f(\mathbf{X}_t; \Xi_t) \in \mathbb{R}^{m \times n}$.
    Update $\mathbf{B}_t = \mu \mathbf{B}_{t-1} + \mathbf{G}_t$.
    Update $\overline{\mathbf{B}}_t = \mu \mathbf{B}_t + \mathbf{G}_t$.
    Update $\mathbf{O}_t = \arg \min_{\mathbf{A} \in \mathcal{O}_{m \times n}} \|\mathbf{A} - \overline{\mathbf{B}}_t\|_F$
    Update $\mathbf{X}_{t+1} = \mathbf{X}_t - \eta_t (\mathbf{O}_t + \lambda \mathbf{X}_t)$.
**end for**

---

**Theorem B.7.** *(Muon with Nesterov momentum as Stochastic FW) Muon with Nesterov momentum (Algorithm 7) is an instance of a Stochastic Frank-Wolfe (Algorithm 3) when using the parameters $\beta_{1,t} = \mu^2$, $\gamma_t = 1 - \mu$, $\eta_t^{\text{Alg 3}} = \lambda \eta_t^{\text{Alg 7}}$, for all $t$, setting $\mathcal{C} = \{\mathbf{A} \in \mathbb{R}^{m \times n} : \|\mathbf{A}\|_2 \leq \frac{1}{\lambda}\}$, and letting $\langle \cdot, \cdot \rangle$ denote the Frobenius inner product.*

*Proof.* Let $\eta_t^{\text{Alg 7}}$ and $\eta_t^{\text{Alg 3}}$ denote the step-sizes used at step $t$ of Algorithm 7 and Algorithm 3, respectively. Then, by setting $\beta_{1,t} \leftarrow \mu^2$, $\gamma_t \leftarrow 1 - \mu$, $\eta_t^{\text{Alg 3}} \leftarrow \lambda \eta_t^{\text{Alg 7}}$, for all $t$, $\mathcal{C} = \{\mathbf{A} \in \mathbb{R}^{m \times n} : \|\mathbf{A}\|_2 \leq \frac{1}{\lambda}\}$ and letting $\langle \cdot, \cdot \rangle$ denote the

Frobenius inner product, we show by induction that the following equations are maintained for all $t \geq 0$.

$$\mathbf{B}_t = \frac{\mathbf{g}_t}{1 - \mu} \tag{10}$$

$$\overline{\mathbf{B}}_t = \frac{\hat{\mathbf{g}}_t}{1 - \mu} \tag{11}$$

$$\mathbf{O}_t = -\lambda \mathbf{u}_t \tag{12}$$

$$\mathbf{X}_{t+1} = \mathbf{x}_{t+1} \tag{13}$$

The rest of the proof proceeds analogously to the proof of Theorem 3.2. $\qquad\square$

## B.4   Proof of Theorem 3.3

**Theorem B.3.** *Set $\eta_t = \frac{1}{D\sqrt{T}}$ and the batch size $m_t = m$, for all $t \geq 1$. Let $\mathbf{x}_a$ be chosen uniformly at random from $\{\mathbf{x}_t\}_{t=1}^T$. Then, under Assumptions 2.3-a and 2.4 with $p = 2$ (i.e., bounded variance), Algorithm 3 satisfies*

$$\mathbb{E}[\mathcal{G}(\mathbf{x}_a)] = O\left( \frac{D\left(F(\mathbf{x}_1) - F(\mathbf{x}_*) + L(1/2 + \beta/\gamma)\right)}{T^{1/2}} + \frac{D\sigma}{\sqrt{m}}\left( \frac{\beta}{\gamma(1-\gamma)} + 1\right)\right), \tag{14}$$

*for any $\beta \in [0, 1-\gamma]$ and $\gamma \in (0,1)$.*

*Proof.* We recall the Frank-Wolfe gap:

$$\mathcal{G}(\mathbf{x}) = \max_{\mathbf{v} \in \mathcal{C}} \langle \mathbf{v} - \mathbf{x}, -\nabla F(\mathbf{x}) \rangle.$$

In the following, we denote the point $\hat{\mathbf{v}}_t$ as

$$\hat{\mathbf{v}}_t := \arg \max_{\mathbf{v} \in \mathcal{C}} \langle \mathbf{v}, -\nabla F(\mathbf{x}_t) \rangle$$

for some fixed time $t \geq 1$. We also denote $\nabla f(\mathbf{x}_t; \Xi_t) := \frac{1}{m_t} \sum_{i=1}^{m_t} \nabla_\mathbf{x} f\left(\mathbf{x}_t; \xi_t^i\right)$, the mini-batch of stochastic gradients at $t$ for brevity, where $\Xi_t$ denotes the randomness at $t$.

On the other hand, we have

$$
\begin{aligned}
F\left(\mathbf{x}_{t+1}\right) &= F\left(\mathbf{x}_t + \eta_t(\mathbf{u}_t - \mathbf{x}_t)\right) \\
&\leq F\left(\mathbf{x}_t\right) + \eta_t \langle \nabla F\left(\mathbf{x}_t\right), \mathbf{u}_t - \mathbf{x}_t \rangle + \frac{\eta_t^2 L}{2} \|\mathbf{u}_t - \mathbf{x}_t\|^2 \\
&\leq F\left(\mathbf{x}_t\right) + \eta_t \langle \nabla F\left(\mathbf{x}_t\right), \mathbf{u}_t - \mathbf{x}_t \rangle + \frac{L}{2}\eta_t^2 D^2 \\
&= F\left(\mathbf{x}_t\right) + \eta_t \langle \hat{\mathbf{g}}_t, \mathbf{u}_t - \mathbf{x}_t \rangle + \eta_t \langle \nabla F\left(\mathbf{x}_t\right) - \hat{\mathbf{g}}_t, \mathbf{u}_t - \mathbf{x}_t \rangle + \frac{L}{2}\eta_t^2 D^2 \\
&\leq F\left(\mathbf{x}_t\right) + \eta_t \langle \hat{\mathbf{g}}_t, \hat{\mathbf{v}}_t - \mathbf{x}_t \rangle + \eta_t \langle \nabla F\left(\mathbf{x}_t\right) - \hat{\mathbf{g}}_t, \mathbf{u}_t - \mathbf{x}_t \rangle + \frac{L}{2}\eta_t^2 D^2 \\
&= F\left(\mathbf{x}_t\right) + \eta_t \langle \hat{\mathbf{g}}_t, \hat{\mathbf{v}}_t - \mathbf{x}_t - \mathbf{u}_t + \mathbf{x}_t \rangle + \eta_t \langle \nabla F\left(\mathbf{x}_t\right), \mathbf{u}_t - \mathbf{x}_t \rangle + \frac{L}{2}\eta_t^2 D^2 \\
&= F\left(\mathbf{x}_t\right) + \eta_t \langle \hat{\mathbf{g}}_t, \hat{\mathbf{v}}_t - \mathbf{u}_t \rangle + \eta_t \langle \nabla F\left(\mathbf{x}_t\right), \mathbf{u}_t - \hat{\mathbf{v}}_t + \hat{\mathbf{v}}_t - \mathbf{x}_t \rangle + \frac{L}{2}\eta_t^2 D^2 \\
&= F\left(\mathbf{x}_t\right) + \eta_t \langle \nabla F\left(\mathbf{x}_t\right), \hat{\mathbf{v}}_t - \mathbf{x}_t \rangle + \eta_t \langle \nabla F\left(\mathbf{x}_t\right) - \hat{\mathbf{g}}_t, \mathbf{u}_t - \hat{\mathbf{v}}_t \rangle + \frac{L}{2}\eta_t^2 D^2 \\
&\leq F\left(\mathbf{x}_t\right) - \eta_t \mathcal{G}(\mathbf{x}_t) + \eta_t D \|\nabla F\left(\mathbf{x}_t\right) - \hat{\mathbf{g}}_t\| + \frac{L}{2}\eta_t^2 D^2, \tag{15}
\end{aligned}
$$

where the first inequality is from the $L$-smoothness of $F$, the second inequality follows by the diameter $D$ of the set $\mathcal{C}$, the third inequality follows from the optimality of $\mathbf{u}_t$, and the fourth inequality follows from the definition of the Frank-Wolfe

gap, the Cauchy-Schwarz inequality and the diameter of $\mathcal{C}$. Furthermore,

$$
\begin{aligned}
\|\nabla F(\mathbf{x}_t) - \hat{\mathbf{g}}_t\| &= \|\nabla F(\mathbf{x}_t) - \mathbf{g}_t + \mathbf{g}_t - \hat{\mathbf{g}}_t\| \\
&= \left\| \nabla F(\mathbf{x}_t) - \mathbf{g}_t + \left(1 - \frac{\beta_{1,t}}{1-\gamma_t}\right)(\mathbf{g}_t - \bar{\mathbf{g}}_t) \right\| \\
&= \left\| \nabla F(\mathbf{x}_t) - \mathbf{g}_t + \left(1 - \frac{\beta_{1,t}}{1-\gamma_t}\right)(\mathbf{g}_t - \nabla F(\mathbf{x}_t) + \nabla F(\mathbf{x}_t) - \bar{\mathbf{g}}_t) \right\| \\
&= \left\| \frac{\beta_{1,t}}{1-\gamma_t}(\nabla F(\mathbf{x}_t) - \mathbf{g}_t) + \left(1 - \frac{\beta_{1,t}}{1-\gamma_t}\right)(\nabla F(\mathbf{x}_t) - \bar{\mathbf{g}}_t) \right\| \\
&\le \left| \frac{\beta_{1,t}}{1-\gamma_t} \right| \|\nabla F(\mathbf{x}_t) - \mathbf{g}_t\| + \left|1 - \frac{\beta_{1,t}}{1-\gamma_t}\right| \|\nabla F(\mathbf{x}_t) - \bar{\mathbf{g}}_t\|.
\end{aligned}
$$

Denote $\epsilon_t := \mathbf{g}_t - \nabla F(\mathbf{x}_t)$. Combining all the above, we obtain

$$
\begin{aligned}
&\mathbb{E}[F(\mathbf{x}_{t+1})] \\
&\le \mathbb{E}[F(\mathbf{x}_t)] - \eta_t \mathbb{E}[\mathcal{G}(\mathbf{x}_t)] + \eta_t \left|\frac{\beta_{1,t}}{1-\gamma_t}\right| D \mathbb{E}[\|\epsilon_t\|] + \eta_t \left|1 - \frac{\beta_{1,t}}{1-\gamma_t}\right| D \underbrace{\mathbb{E}[\|\nabla F(\mathbf{x}_t) - \bar{\mathbf{g}}_t\|]}_{:=\theta_t \le \frac{\sigma}{\sqrt{m_t}}} + \frac{L}{2}\eta_t^2 D^2.
\end{aligned} \tag{16}
$$

Then, from the update in Line 6 of Algorithm 3 we obtain

$$
\underbrace{\mathbf{g}_t - \nabla F(\mathbf{x}_t)}_{:=\epsilon_t} = (1-\gamma_t)\underbrace{(\mathbf{g}_{t-1} - \nabla F(\mathbf{x}_{t-1}))}_{:=\epsilon_{t-1}} + (1-\gamma_t)(\nabla F(\mathbf{x}_{t-1}) - \nabla F(\mathbf{x}_t)) + \gamma_t(\bar{\mathbf{g}}_t - \nabla F(\mathbf{x}_t)).
$$

By the triangle inequality and $L$-Lipschitz gradient assumption we have

$$
\begin{aligned}
\mathbb{E}[\|\epsilon_t\|] &\le (1-\gamma_t)\mathbb{E}[\|\epsilon_{t-1}\|] + (1-\gamma_t)\mathbb{E}[\|\nabla F(\mathbf{x}_{t-1}) - \nabla F(\mathbf{x}_t)\|] + \gamma_t \theta_t \\
&\le (1-\gamma_t)\mathbb{E}[\|\epsilon_{t-1}\|] + (1-\gamma_t)\eta_{t-1} L \mathbb{E}[\|\mathbf{u}_{t-1} - \mathbf{x}_{t-1}\|] + \gamma_t \theta_t \\
&\le (1-\gamma_t)\mathbb{E}[\|\epsilon_{t-1}\|] + (1-\gamma_t)\eta_{t-1} L D + \gamma_t \theta_t.
\end{aligned} \tag{17}
$$

Now define a potential function $\Psi_t := \mathbb{E}[F(\mathbf{x}_t)] + C_t \mathbb{E}[\|\epsilon_t\|]$, where $\{C_t\}_{t\ge1}$ is a sequence of positive numbers to be determined later. Using (16), (17) and the definition of the potential function we obtain

$$
\begin{aligned}
\Psi_{t+1} &= \mathbb{E}[F(\mathbf{x}_{t+1})] + C_{t+1}\mathbb{E}[\|\epsilon_{t+1}\|] \\
&\le \mathbb{E}[F(\mathbf{x}_t)] - \eta_t \mathbb{E}[\mathcal{G}(\mathbf{x}_t)] + \eta_t \left|\frac{\beta_{1,t}}{1-\gamma_t}\right| D\mathbb{E}[\|\epsilon_t\|] + \eta_t\left|1 - \frac{\beta_{1,t}}{1-\gamma_t}\right| D\theta_t + \frac{L}{2}\eta_t^2 D^2 \\
&\quad + C_{t+1}[(1-\gamma_{t+1})\mathbb{E}[\|\epsilon_t\|] + (1-\gamma_{t+1})\eta_t L D + \gamma_{t+1}\theta_{t+1}] \\
&= \mathbb{E}[F(\mathbf{x}_t)] - \eta_t \mathbb{E}[\mathcal{G}(\mathbf{x}_t)] + \left[C_{t+1}(1-\gamma_{t+1}) + \eta_t\left|\frac{\beta_{1,t}}{1-\gamma_t}\right| D\right]\mathbb{E}[\|\epsilon_t\|] \\
&\quad + \eta_t\left|1 - \frac{\beta_{1,t}}{1-\gamma_t}\right| D\theta_t + \frac{L}{2}\eta_t^2 D^2 + C_{t+1}[(1-\gamma_{t+1})\eta_t L D + \gamma_{t+1}\theta_{t+1}].
\end{aligned} \tag{18}
$$

We can specify the sequences $C_t, \gamma_t, \eta_t$ and $\beta_{1,t}$ such that for all $t$

$$
C_{t+1}(1-\gamma_{t+1}) + \eta_t\left|\frac{\beta_{1,t}}{1-\gamma_t}\right| D \le C_t. \tag{19}
$$

Then, (18) can further be upper-bounded as

$$
\Psi_{t+1} \le \Psi_t - \eta_t \mathbb{E}[\mathcal{G}(\mathbf{x}_t)] + \eta_t\left|1 - \frac{\beta_{1,t}}{(1-\gamma_t)}\right| D\theta_t + \frac{L}{2}\eta_t^2 D^2 + C_{t+1}[(1-\gamma_{t+1})\eta_t L D + \gamma_{t+1}\theta_{t+1}].
$$

Rearranging and summing from $t = 1, \ldots, T$ we obtain

$$\sum_{t=1}^{T} \eta_t \mathbb{E}[\mathcal{G}(\mathbf{x}_t)]$$

$$\leq \Psi_1 - \Psi_{T+1} + \sum_{t=1}^{T} \left( \eta_t \left| 1 - \frac{\beta_{1,t}}{(1-\gamma_t)} \right| D\theta_t + \frac{L}{2}\eta_t^2 D^2 + C_{t+1} \left[ (1-\gamma_{t+1})\eta_t LD + \gamma_{t+1}\theta_{t+1} \right] \right)$$

$$\leq F(\mathbf{x}_1) + C_1 \mathbb{E}[\|\epsilon_1\|] - F(\mathbf{x}_*)$$

$$+ \sum_{t=1}^{T} \left( \eta_t \left| 1 - \frac{\beta_{1,t}}{(1-\gamma_t)} \right| D\theta_t + \frac{L}{2}\eta_t^2 D^2 + C_{t+1} \left[ (1-\gamma_{t+1})\eta_t LD + \gamma_{t+1}\theta_{t+1} \right] \right)$$

$$= F(\mathbf{x}_1) - F(\mathbf{x}_*) + C_1 \mathbb{E}[\|\epsilon_1\|]$$

$$+ \sum_{t=1}^{T} \left( \eta_t \left| 1 - \frac{\beta_{1,t}}{(1-\gamma_t)} \right| D\theta_t + \frac{L}{2}\eta_t^2 D^2 + C_{t+1} \left[ (1-\gamma_{t+1})\eta_t LD + \gamma_{t+1}\theta_{t+1} \right] \right). \tag{20}$$

The sequence $\{C_t\}_t$ can be chosen such that $C_t = C$, for all $t$. Then, the constraint (19) is reduced to

$$\frac{\eta_t D \left| \frac{\beta_{1,t}}{1-\gamma_t} \right|}{\gamma_{t+1}} \leq C,$$

for all $t$. If we set $\eta_t = \frac{1}{D\sqrt{T}}$, $\beta_{1,t} = \beta$, $\gamma_t = \gamma$, with $\beta + \gamma \leq 1$, for all $t$, then we can choose

$$C \leftarrow \frac{\beta}{\gamma(1-\gamma)\sqrt{T}}.$$

Using these parameter choices and the bound $\theta_t \leq \frac{\sigma}{\sqrt{m_t}}$, from (20) we get

$$\frac{\sqrt{T}}{D} \frac{1}{T} \sum_{t=1}^{T} \mathbb{E}[\mathcal{G}(\mathbf{x}_t)] \leq F(\mathbf{x}_1) - F(\mathbf{x}_*) + \frac{\beta}{\gamma(1-\gamma)\sqrt{T}} \mathbb{E}[\|\epsilon_1\|] + \left( \frac{1-\gamma-\beta}{(1-\gamma)\sqrt{T}} \right) \sum_{t=1}^{T} \frac{\sigma}{\sqrt{m_t}} + \frac{L}{2}$$

$$+ L\frac{\beta}{\gamma} + \left( \frac{\beta}{(1-\gamma)\sqrt{T}} \right) \sum_{t=1}^{T} \frac{\sigma}{\sqrt{m_{t+1}}}. \tag{21}$$

We further note that

$$\mathbb{E}[\|\epsilon_1\|] = \mathbb{E}[\|\gamma\bar{g}_1 - \nabla F(\mathbf{x}_1)\|] \leq \gamma \mathbb{E}[\|\bar{g}_1 - \nabla F(\mathbf{x}_1)\|] + (1-\gamma)\|\nabla F(\mathbf{x}_1)\|$$

$$\leq \gamma\frac{\sigma}{\sqrt{m_1}} + (1-\gamma)\|\nabla F(\mathbf{x}_1)\|. \tag{22}$$

Combining (21) and (22), we have

$$\frac{\sqrt{T}}{D} \frac{1}{T} \sum_{t=1}^{T} \mathbb{E}[\mathcal{G}(\mathbf{x}_t)] \leq F(\mathbf{x}_1) - F(\mathbf{x}_*) + \frac{\beta}{(1-\gamma)\sqrt{T}}\frac{\sigma}{\sqrt{m_1}} + \frac{\beta}{\gamma\sqrt{T}}\|\nabla F(\mathbf{x}_1)\|$$

$$+ \left( \frac{1-\gamma-\beta}{(1-\gamma)\sqrt{T}} \right) \sum_{t=1}^{T} \frac{\sigma}{\sqrt{m_t}} + \frac{L}{2} + L\frac{\beta}{\gamma} + \left( \frac{\beta}{(1-\gamma)\sqrt{T}} \right) \sum_{t=1}^{T} \frac{\sigma}{\sqrt{m_{t+1}}}.$$

Let $m_t = m$, for all $t \in [T]$. Then, assuming that the output $\mathbf{x}_a$ is chosen uniformly at random from $\{\mathbf{x}_t\}_{t=1}^{T}$, we have

$$\mathbb{E}[\mathcal{G}(\mathbf{x}_a)] \leq \frac{D}{\sqrt{T}} \left( F(\mathbf{x}_1) - F(\mathbf{x}_*) + L\left( \frac{1}{2} + \frac{\beta}{\gamma} \right) + \frac{\beta}{\gamma\sqrt{T}}\|\nabla F(\mathbf{x}_1)\| \right) + \frac{D\sigma}{\sqrt{m}} \left( \frac{\beta}{\gamma(1-\gamma)} + 1 \right).$$

$$\square$$

# C  Proofs of the theoretical results in Section 4

## C.1  Proof of Theorem 4.1

**Theorem C.1.** *Set $\eta_t = \frac{1}{DT^{2/3}}$, $\gamma_t = \frac{1}{T^{2/3}}$, $\beta_{1,t} = 1 - \frac{1}{T^{1/3}}$, for all $t \geq 1$, and the batch size $m_t = m$, for all $t > 1$. Let $\mathbf{x}_a$ be chosen uniformly at random from $\{\mathbf{x}_t\}_{t=1}^T$. Then, under Assumptions 2.1, 2.2, and 2.4 with $p = 2$ (i.e., bounded variance), Algorithm 4 satisfies $\mathbb{E}[\mathcal{G}(\mathbf{x}_a)] = O\left( \frac{D\left(F(\mathbf{x}_1) - F(\mathbf{x}_*) + L^2 + \frac{\sigma^2}{m}\right)}{T^{1/3}} + \frac{D\sigma^2}{m_1} \right).$*

*Proof.* From (15), we have

$$F\left(\mathbf{x}_{t+1}\right) \leq F\left(\mathbf{x}_t\right) - \eta_t \mathcal{G}(\mathbf{x}_t) + \eta_t D \|\nabla F\left(\mathbf{x}_t\right) - \hat{\mathbf{g}}_t\| + \frac{L}{2} \eta_t^2 D^2$$

$$\leq F\left(\mathbf{x}_t\right) - \eta_t \mathcal{G}(\mathbf{x}_t) + \frac{\eta_t \nu D}{2} + \frac{\eta_t D}{2\nu} \|\nabla F\left(\mathbf{x}_t\right) - \hat{\mathbf{g}}_t\|^2 + \frac{L}{2} \eta_t^2 D^2,$$

where the second inequality follows from Young's inequality and $\nu > 0$ is a constant. Then, we have

$$\mathbb{E}[F(\mathbf{x}_{t+1})] \leq \mathbb{E}[F\left(\mathbf{x}_t\right)] - \eta_t \mathbb{E}[\mathcal{G}(\mathbf{x}_t)] + \frac{\eta_t \nu D}{2} + \frac{\eta_t D}{2\nu} \mathbb{E}[\|\nabla F\left(\mathbf{x}_t\right) - \hat{\mathbf{g}}_t\|^2] + \frac{L}{2} \eta_t^2 D^2. \tag{23}$$

Furthermore,

$$\|\nabla F\left(\mathbf{x}_t\right) - \hat{\mathbf{g}}_t\|^2 = \|\nabla F\left(\mathbf{x}_t\right) - \mathbf{g}_t + \mathbf{g}_t - \hat{\mathbf{g}}_t\|^2$$

$$= \left\| \nabla F\left(\mathbf{x}_t\right) - \mathbf{g}_t + \left(1 - \frac{\beta_{1,t}}{1 - \gamma_t}\right) (\mathbf{g}_t - \bar{\mathbf{g}}_t) \right\|^2$$

$$= \left\| \nabla F\left(\mathbf{x}_t\right) - \mathbf{g}_t + \left(1 - \frac{\beta_{1,t}}{1 - \gamma_t}\right) (\mathbf{g}_t - \nabla F\left(\mathbf{x}_t\right) + \nabla F\left(\mathbf{x}_t\right) - \bar{\mathbf{g}}_t) \right\|^2$$

$$= \left\| \frac{\beta_{1,t}}{1 - \gamma_t} (\nabla F\left(\mathbf{x}_t\right) - \mathbf{g}_t) + \left(1 - \frac{\beta_{1,t}}{1 - \gamma_t}\right) (\nabla F\left(\mathbf{x}_t\right) - \bar{\mathbf{g}}_t) \right\|^2$$

$$\leq 2 \left(\frac{\beta_{1,t}}{1 - \gamma_t}\right)^2 \|\nabla F\left(\mathbf{x}_t\right) - \mathbf{g}_t\|^2 + 2 \left(1 - \frac{\beta_{1,t}}{1 - \gamma_t}\right)^2 \|\nabla F\left(\mathbf{x}_t\right) - \bar{\mathbf{g}}_t\|^2, \tag{24}$$

where the inequality follows from $\|\mathbf{u} + \mathbf{v}\|^2 \leq 2\|\mathbf{u}\|^2 + 2\|\mathbf{v}\|^2$. As before, denote $\epsilon_t := \mathbf{g}_t - \nabla F(\mathbf{x}_t)$. Let $\mathcal{G}(\mathbf{x})$, $\hat{\mathbf{v}}_t$ as previously defined. Let $\nabla f(\mathbf{x}_t; \Xi_t) := \frac{1}{m_t} \sum_{i=1}^{m_t} \nabla_{\mathbf{x}} f\left(\mathbf{x}_t; \xi_t^i\right)$ and $\nabla f(\mathbf{x}_{t-1}; \Xi_t) := \frac{1}{m_t} \sum_{i=1}^{m_t} \nabla_{\mathbf{x}} f\left(\mathbf{x}_{t-1}; \xi_t^i\right)$, where $\Xi_t$ represents the randomness at $t$. By adding and subtracting the full gradient $\nabla F(\mathbf{x}_t)$ from the update in Step 4 in Algorithm 4, we obtain

$$\underbrace{\mathbf{g}_t - \nabla F(\mathbf{x}_t)}_{:=\epsilon_t} = (1 - \gamma_t) \underbrace{(\mathbf{g}_{t-1} - \nabla F(\mathbf{x}_{t-1}))}_{:=\epsilon_{t-1}} + \gamma_t \left(\nabla f(\mathbf{x}_t; \Xi_t) - \nabla F(\mathbf{x}_t)\right)$$

$$+ (1 - \gamma_t) \left(\nabla f(\mathbf{x}_t; \Xi_t) - \nabla f(\mathbf{x}_{t-1}; \Xi_t) + \nabla F(\mathbf{x}_{t-1}) - \nabla F(\mathbf{x}_t)\right).$$

Now, using that $\|\mathbf{u} + \mathbf{v}\|^2 \leq 2\|\mathbf{u}\|^2 + 2\|\mathbf{v}\|^2$ and that $\mathbb{E}[\langle \nabla f(\mathbf{x}_t; \Xi_t) - \nabla f(\mathbf{x}_{t-1}; \Xi_t) + \nabla F(\mathbf{x}_{t-1}) - \nabla F(\mathbf{x}_t), \epsilon_{t-1}\rangle] = 0$ as well as that $\mathbb{E}[\langle \nabla f(\mathbf{x}_t; \Xi_t) - \nabla F(\mathbf{x}_t), \epsilon_{t-1}\rangle] = 0$, we obtain

$$\mathbb{E}[\|\epsilon_t\|^2] \leq 2 (1 - \gamma_t)^2 \mathbb{E}\left[\|\nabla f(\mathbf{x}_t; \Xi_t) - \nabla f(\mathbf{x}_{t-1}; \Xi_t) + \nabla F(\mathbf{x}_{t-1}) - \nabla F(\mathbf{x}_t)\|^2\right]$$

$$+ 2\gamma_t^2 \mathbb{E}\left[\|\nabla f(\mathbf{x}_t; \Xi_t) - \nabla F(\mathbf{x}_t)\|^2\right] + (1 - \gamma_t)^2 \mathbb{E}\left[\|\epsilon_{t-1}\|^2\right]$$

$$\leq 2 (1 - \gamma_t)^2 \mathbb{E}\left[\|\nabla f(\mathbf{x}_t; \Xi_t) - \nabla f(\mathbf{x}_{t-1}; \Xi_t)\|^2\right] + 2 \frac{\gamma_t^2 \sigma^2}{m_t} + (1 - \gamma_t)^2 \mathbb{E}\left[\|\epsilon_{t-1}\|^2\right],$$

where the second inequality uses the fact that $\|\nabla f(\mathbf{x}_t; \Xi_t) - \nabla F(\mathbf{x}_t)\|^2 \leq \frac{\sigma^2}{m_t}$. Next, using the averaged $L$-Lipschitz gradient assumption, we further have

$$\leq 2 (1 - \gamma_t)^2 L^2 \mathbb{E}\left[\|\mathbf{x}_t - \mathbf{x}_{t-1}\|^2\right] + 2 \frac{\gamma_t^2 \sigma^2}{m_t} + (1 - \gamma_t)^2 \mathbb{E}\left[\|\epsilon_{t-1}\|^2\right]$$

Now, from the update step we have $\mathbf{x}_t - \mathbf{x}_{t-1} = \eta_t(\mathbf{u}_t - \mathbf{x}_{t-1})$, and thus

$$= 2(1 - \gamma_t)^2 L^2 \eta_t^2 \mathbb{E}\left[\|\mathbf{u}_t - \mathbf{x}_{t-1}\|^2\right] + 2\frac{\gamma_t^2 \sigma^2}{m_t} + (1 - \gamma_t)^2 \mathbb{E}\left[\|\epsilon_{t-1}\|^2\right]$$

$$\leq 2(1 - \gamma_t)^2 L^2 \eta_t^2 D^2 + 2\frac{\gamma_t^2 \sigma^2}{m_t} + (1 - \gamma_t)^2 \mathbb{E}\left[\|\epsilon_{t-1}\|^2\right], \tag{25}$$

where the last inequality is obtained by the assumption on the diameter $D$ of the constraint set. Combining the above, from (23) we have

$$\mathbb{E}[F(\mathbf{x}_{t+1})] \leq \mathbb{E}[F(\mathbf{x}_t)] - \eta_t \mathbb{E}[\mathcal{G}(\mathbf{x}_t)] + \frac{\eta_t \nu D}{2} + \frac{\eta_t D}{2\nu}\mathbb{E}[\|\nabla F(\mathbf{x}_t) - \hat{\mathbf{g}}_t\|^2] + \frac{L}{2}\eta_t^2 D^2$$

$$\leq \mathbb{E}[F(\mathbf{x}_t)] - \eta_t \mathbb{E}[\mathcal{G}(\mathbf{x}_t)] + \frac{\eta_t \nu D}{2} + 2\frac{\eta_t D}{2\nu}\left(\frac{\beta_{1,t}}{1 - \gamma_t}\right)^2 \mathbb{E}\left[\|\nabla F(\mathbf{x}_t) - \mathbf{g}_t\|^2\right]$$

$$+ 2\frac{\eta_t D}{2\nu}\left(1 - \frac{\beta_{1,t}}{1 - \gamma_t}\right)^2 \mathbb{E}\left[\|\nabla F(\mathbf{x}_t) - \bar{\mathbf{g}}_t\|^2\right] + \frac{L}{2}\eta_t^2 D^2$$

$$\leq \mathbb{E}[F(\mathbf{x}_t)] - \eta_t \mathbb{E}[\mathcal{G}(\mathbf{x}_t)] + \frac{\eta_t \nu D}{2} + \frac{\eta_t D}{\nu}\left(\frac{\beta_{1,t}}{1 - \gamma_t}\right)^2 \mathbb{E}\left[\|\epsilon_t\|^2\right]$$

$$+ \frac{\eta_t D}{\nu}\left(1 - \frac{\beta_{1,t}}{1 - \gamma_t}\right)^2 \frac{\sigma^2}{m_t} + \frac{L}{2}\eta_t^2 D^2,$$

where the second inequality follows from (24) and the third inequality follows from $\|\bar{\mathbf{g}}_t - \nabla F(\mathbf{x}_t)\|^2 \leq \frac{\sigma^2}{m_t}$. Now, defining the potential function $\Phi_t := \mathbb{E}[F(\mathbf{x}_t)] + C_t \mathbb{E}[\|\epsilon_t\|^2]$, and using the above inequality we have

$$\Phi_{t+1} = \mathbb{E}[F(\mathbf{x}_{t+1})] + C_{t+1}\mathbb{E}[\|\epsilon_{t+1}\|^2]$$

$$\leq \mathbb{E}[F(\mathbf{x}_t)] - \eta_t \mathbb{E}[\mathcal{G}(\mathbf{x}_t)] + \frac{\eta_t \nu D}{2} + \frac{\eta_t D}{\nu}\left(\frac{\beta_{1,t}}{1 - \gamma_t}\right)^2 \mathbb{E}\left[\|\epsilon_t\|^2\right]$$

$$+ \frac{\eta_t D}{\nu}\left(1 - \frac{\beta_{1,t}}{1 - \gamma_t}\right)^2 \frac{\sigma^2}{m_t} + \frac{L}{2}\eta_t^2 D^2 + C_{t+1}\mathbb{E}[\|\epsilon_{t+1}\|^2]$$

$$\leq \mathbb{E}[F(\mathbf{x}_t)] - \eta_t \mathbb{E}[\mathcal{G}(\mathbf{x}_t)] + \frac{\eta_t \nu D}{2} + \frac{\eta_t D}{\nu}\left(\frac{\beta_{1,t}}{1 - \gamma_t}\right)^2 \mathbb{E}\left[\|\epsilon_t\|^2\right]$$

$$+ \frac{\eta_t D}{\nu}\left(1 - \frac{\beta_{1,t}}{1 - \gamma_t}\right)^2 \frac{\sigma^2}{m_t} + \frac{L}{2}\eta_t^2 D^2$$

$$+ C_{t+1}\left(2(1 - \gamma_{t+1})^2 L^2 \eta_{t+1}^2 D^2 + 2\frac{\gamma_{t+1}^2 \sigma^2}{m_{t+1}} + (1 - \gamma_{t+1})^2 \mathbb{E}\left[\|\epsilon_t\|^2\right]\right)$$

$$= \mathbb{E}[F(\mathbf{x}_t)] - \eta_t \mathbb{E}[\mathcal{G}(\mathbf{x}_t)] + \frac{\eta_t \nu D}{2} + \left[C_{t+1}(1 - \gamma_{t+1})^2 + \frac{\eta_t D}{\nu}\left(\frac{\beta_{1,t}}{1 - \gamma_t}\right)^2\right] \mathbb{E}\left[\|\epsilon_t\|^2\right]$$

$$+ \frac{\eta_t D}{\nu}\left(1 - \frac{\beta_{1,t}}{1 - \gamma_t}\right)^2 \frac{\sigma^2}{m_t} + \frac{L}{2}\eta_t^2 D^2 + C_{t+1}\left(2(1 - \gamma_{t+1})^2 L^2 \eta_{t+1}^2 D^2 + 2\frac{\gamma_{t+1}^2 \sigma^2}{m_{t+1}}\right),$$

where the second inequality follows from (25). We set $(C_t)_{t \geq 1}$ so that

$$C_{t+1}(1 - \gamma_{t+1})^2 + \frac{\eta_t D}{\nu}\left(\frac{\beta_{1,t}}{1 - \gamma_t}\right)^2 \leq C_t. \tag{26}$$

Then, we have

$$\Phi_{t+1} \leq \Phi_t - \eta_t \mathbb{E}[\mathcal{G}(\mathbf{x}_t)] + \frac{\eta_t \nu D}{2} + \frac{\eta_t D}{\nu}\left(1 - \frac{\beta_{1,t}}{1 - \gamma_t}\right)^2 \frac{\sigma^2}{m_t} + \frac{L}{2}\eta_t^2 D^2$$

$$+ C_{t+1}\left(2(1 - \gamma_{t+1})^2 L^2 \eta_{t+1}^2 D^2 + 2\frac{\gamma_{t+1}^2 \sigma^2}{m_{t+1}}\right).$$

Rearranging and summing the inequality from $t = 1, 2, \ldots, T$, we have

$$\sum_{t=1}^{T} \eta_t \mathbb{E}[\mathcal{G}(\mathbf{x}_t)] \leq \Phi_1 - \Phi_{T+1} + \sum_{t=1}^{T} \frac{\eta_t \nu D}{2} + \sum_{t=1}^{T} \frac{\eta_t D}{\nu} \left(1 - \frac{\beta_{1,t}}{1 - \gamma_t}\right)^2 \frac{\sigma^2}{m_t} + \sum_{t=1}^{T} \frac{L}{2} \eta_t^2 D^2$$

$$+ \sum_{t=1}^{T} C_{t+1} \left(2 \left(1 - \gamma_{t+1}\right)^2 L^2 \eta_{t+1}^2 D^2 + 2 \frac{\gamma_{t+1}^2 \sigma^2}{m_{t+1}}\right).$$

Let $\eta_t = \eta$ be a constant. Then, assuming that the output $\mathbf{x}_a$ is chosen uniformly at random from $\{\mathbf{x}_t\}_{t=1}^T$, we further have

$$T\eta \mathbb{E}[\mathcal{G}(\mathbf{x}_a)] = T\eta \frac{1}{T} \sum_{t=1}^{T} \mathbb{E}[\mathcal{G}(\mathbf{x}_t)]$$

$$\leq \Phi_1 - \Phi_{T+1} + \frac{T\eta \nu D}{2} + \frac{L}{2} T\eta^2 D^2 + \frac{\eta D}{\nu} \sum_{t=1}^{T} \left(1 - \frac{\beta_{1,t}}{1 - \gamma_t}\right)^2 \frac{\sigma^2}{m_t}$$

$$+ \sum_{t=1}^{T} C_{t+1} \left(2 \left(1 - \gamma_{t+1}\right)^2 L^2 \eta^2 D^2 + 2 \frac{\gamma_{t+1}^2 \sigma^2}{m_{t+1}}\right)$$

Therefore,

$$\mathbb{E}[\mathcal{G}(\mathbf{x}_a)] \leq \frac{\Phi_1 - \Phi_{T+1}}{T\eta} + \frac{\nu D}{2} + \frac{L\eta D^2}{2} + \frac{D}{\nu T} \sum_{t=1}^{T} \left(1 - \frac{\beta_{1,t}}{1 - \gamma_t}\right)^2 \frac{\sigma^2}{m_t}$$

$$+ \frac{1}{T\eta} \sum_{t=1}^{T} C_{t+1} \left(2 \left(1 - \gamma_{t+1}\right)^2 L^2 \eta^2 D^2 + 2 \frac{\gamma_{t+1}^2 \sigma^2}{m_{t+1}}\right)$$

$$\leq \frac{F(\mathbf{x}_1) - F(\mathbf{x}^*)}{T\eta} + \frac{C_1 \mathbb{E}\left[\|\epsilon_1\|^2\right]}{T\eta} + \frac{\nu D}{2} + \frac{L\eta D^2}{2} + \frac{D}{\nu T} \sum_{t=1}^{T} \left(1 - \frac{\beta_{1,t}}{1 - \gamma_t}\right)^2 \frac{\sigma^2}{m_t}$$

$$+ \frac{1}{T\eta} \sum_{t=1}^{T} C_{t+1} \left(2 \left(1 - \gamma_{t+1}\right)^2 L^2 \eta^2 D^2 + 2 \frac{\gamma_{t+1}^2 \sigma^2}{m_{t+1}}\right)$$

$$\leq \frac{F(\mathbf{x}_1) - F(\mathbf{x}^*)}{T\eta} + \frac{C_1}{T\eta} \frac{\sigma^2}{m_1} + \frac{\nu D}{2} + \frac{L\eta D^2}{2} + \frac{D}{\nu T} \sum_{t=1}^{T} \left(1 - \frac{\beta_{1,t}}{1 - \gamma_t}\right)^2 \frac{\sigma^2}{m_t}$$

$$+ \frac{1}{T\eta} \sum_{t=1}^{T} C_{t+1} \left(2 \left(1 - \gamma_{t+1}\right)^2 L^2 \eta^2 D^2 + 2 \frac{\gamma_{t+1}^2 \sigma^2}{m_{t+1}}\right). \tag{27}$$

To continue, let $C_t = C$, $\beta_{1,t} = \beta$ and $\gamma_t = \gamma$ be some constants that will be determined soon. Then, (26) becomes $C\left(1 - \gamma\right)^2 + \frac{\eta D}{\nu} \left(\frac{\beta}{1-\gamma}\right)^2 \leq C$. To satisfy the constraint, we can simply set

$$C \leftarrow \frac{\eta D \beta^2}{\nu (1 - \gamma)^2 (1 - (1 - \gamma)^2)}. \tag{28}$$

Substituting the expression of $C$ back into (27), and letting $m_t = m$, for all $t > 1$, we have

$$
\begin{aligned}
\mathbb{E}[\mathcal{G}(\mathbf{x}_a)] \leq\ & \frac{F(\mathbf{x}_1) - F(\mathbf{x}^*)}{T\eta} + \frac{D\beta^2}{\nu(1-\gamma)^2(1-(1-\gamma)^2)T}\frac{\sigma^2}{m_1} \\
& + \frac{\nu D}{2} + \frac{L\eta D^2}{2} + \frac{D}{\nu}\left(1-\frac{\beta}{1-\gamma}\right)^2\frac{\sigma^2}{m} \\
& + \frac{D\beta^2}{\nu(1-\gamma)^2(1-(1-\gamma)^2)}\left[2(1-\gamma)^2 L^2\eta^2 D^2 + 2\frac{\gamma^2\sigma^2}{m}\right] \\
=\ & \frac{F(\mathbf{x}_1) - F(\mathbf{x}^*)}{T\eta} + \frac{D\beta^2}{\nu\gamma(1-\gamma)^2(2-\gamma)T}\frac{\sigma^2}{m_1} \\
& + \frac{\nu D}{2} + \frac{L\eta D^2}{2} + \frac{D}{\nu}\left(1-\frac{\beta}{1-\gamma}\right)^2\frac{\sigma^2}{m} \\
& + \frac{D\beta^2}{\nu\gamma(1-\gamma)^2(2-\gamma)}\left[2(1-\gamma)^2 L^2\eta^2 D^2 + 2\frac{\gamma^2\sigma^2}{m}\right]
\end{aligned}
$$

We can choose parameters $\eta = \frac{1}{DT^{2/3}}$, $\gamma = \frac{1}{T^{2/3}}$, $\beta = 1 - \frac{1}{T^{1/3}}$ and $\nu = \frac{1}{T^{1/3}}$, for which we obtain

$$
\begin{aligned}
\mathbb{E}[\mathcal{G}(\mathbf{x}_a)] \leq\ & \frac{D(F(\mathbf{x}_1) - F(\mathbf{x}^*))}{T^{1/3}} + \frac{DT^{2/3}\left(1-T^{-1/3}\right)^2}{\left(2T^{2/3}-1\right)\left(1-T^{-2/3}\right)^2}\frac{\sigma^2}{m_1} \\
& + \frac{D}{2T^{1/3}} + \frac{LD}{2T^{2/3}} + DT^{1/3}\left(\frac{T^{-1/3}-T^{-2/3}}{1-T^{-2/3}}\right)^2\frac{\sigma^2}{m} \\
& + \frac{DT^{5/3}\left(1-T^{-1/3}\right)^2}{\left(2T^{2/3}-1\right)\left(1-T^{-2/3}\right)^2}\left[\frac{2L^2}{T^{4/3}}\left(1-\frac{2}{T^{2/3}}+\frac{1}{T^{4/3}}\right)+\frac{2}{T^{4/3}}\frac{\sigma^2}{m}\right].
\end{aligned}
$$

Therefore,

$$
\mathbb{E}[\mathcal{G}(\mathbf{x}_a)] = O\left(\frac{D}{T^{1/3}}\left(F(\mathbf{x}_1) - F(\mathbf{x}^*) + \frac{L^2+1}{2} + \frac{3\sigma^2}{2m}\right) + \frac{LD}{2T^{2/3}} + \frac{L^2 D}{2T^{5/3}} + \frac{D\sigma^2}{4m_1}\right).
$$

$\square$

## C.2   LION+ and MUON+

In this section, we present the algorithmic specifications of LION+ and MUON+, which are obtained as instances of Algorithm 5.

---

**Algorithm 8** LION+

**Required:** Momentum parameters $\beta_1, \beta_2$, step-sizes $\{\eta_t\}$, weight decay parameter $\lambda$, and clipping parameter $M$.
**Initialize:** $\mathbf{m}_0 = 0$ and $\mathbf{x}_1 \in \mathcal{C}$.
**for** $t = 1, 2, \dots$ **do**
   Sample $\Xi_t \sim \mathcal{D}$.
   Compute $\bar{\mathbf{g}}_t = \left(1 \wedge \frac{M}{\|\nabla f(\mathbf{x}_t;\Xi_t)\|}\right)\nabla f(\mathbf{x}_t;\Xi_t)$.
   Update $\mathbf{c}_t = \beta_1\mathbf{m}_{t-1} + (1-\beta_1)\bar{\mathbf{g}}_t$.
   Update $\mathbf{x}_{t+1} = \mathbf{x}_t - \eta_t\left(\text{sign}(\mathbf{c}_t) + \lambda\mathbf{x}_t\right)$.
   Update $\mathbf{m}_t = \beta_2\mathbf{m}_{t-1} + (1-\beta_2)\bar{\mathbf{g}}_t$.
**end for**

---

**Algorithm 9** MUON+

**Required:** Momentum parameter $\mu$, step-sizes $\{\eta_t\}$, weight decay parameter $\lambda$, and clipping parameter $M$.
**Initialize:** $\mathbf{B}_0 = 0$ and $\mathbf{X}_1 \in \mathcal{C}$.
**for** $t = 1, 2, \dots$ **do**
   Sample $\Xi_t \sim \mathcal{D}$.
   Compute $\mathbf{G}_t = \left(1 \wedge \frac{M}{\|\nabla f(\mathbf{X}_t;\Xi_t)\|}\right)\nabla f(\mathbf{X}_t;\Xi_t)$.
   Update $\mathbf{B}_t = \mu\mathbf{B}_{t-1} + \mathbf{G}_t$.
   Update $\mathbf{O}_t = \arg\min_{\mathbf{A}\in\mathcal{O}_{m\times n}}\|\mathbf{A} - \mathbf{B}_t\|_F$.
   Update $\mathbf{X}_{t+1} = \mathbf{X}_t - \eta_t\left(\mathbf{O}_t + \lambda\mathbf{X}_t\right)$.
**end for**

---

## C.3 Proof of Theorem 4.3

In the following, we denote

$$\epsilon_t := \begin{cases} \mathbf{g}_t - \nabla F(\mathbf{x}_t) & t \geq 1 \\ -\nabla F(\mathbf{x}_1) & t = 0 \end{cases} \tag{29}$$

$$Z_t := \nabla F(\mathbf{x}_{t-1}) - \nabla F(\mathbf{x}_t) \tag{30}$$

$$\zeta_t := \bar{\mathbf{g}}_t - \nabla F(\mathbf{x}_t), \quad \zeta_t^u := \bar{\mathbf{g}}_t - \mathbb{E}_t[\bar{\mathbf{g}}_t], \quad \zeta_t^b := \mathbb{E}_t[\bar{\mathbf{g}}_t] - \nabla F(\mathbf{x}_t), \tag{31}$$

where $\mathbb{E}_t[\cdot|\mathcal{F}_{t-1}]$ and $\mathcal{F}_{t-1}$ represents the randomness up to and including $t-1$.

We will need a series of technical lemmas, and we build upon some of the machinery developed in the analysis of Normalized SGD with clipping and momentum by Liu et al. (2023) to derive our result for Stochastic FW with clipping.

**Lemma C.2.** *Set the initial point* $\mathbf{x}_1 = \mathbf{x}_0$. *Let* $\gamma_t = \gamma$ *and* $\eta_t = \eta$ *for some constants* $\gamma > 0$ *and* $\eta > 0$. *Then*

$$\epsilon_t = (1-\gamma)^t \epsilon_0 + (1-\gamma)\left(\sum_{s=1}^t (1-\gamma)^{t-s} Z_s\right) + \gamma\left(\sum_{s=1}^t (1-\gamma)^{t-s}\zeta_s\right).$$

*Proof.* For any $t \geq 2$. by expansion, we obtain

$$\epsilon_t = \mathbf{g}_t - \nabla F(\mathbf{x}_t) = (1-\gamma)\,\mathbf{g}_{t-1} + \gamma\bar{\mathbf{g}}_t - \nabla F(\mathbf{x}_t)$$
$$= (1-\gamma)\epsilon_{t-1} + (1-\gamma)Z_t + \gamma\zeta_t, \tag{32}$$

Furthermore, (32) also holds for $t = 1$. Recursively expanding (32) from $t$ back to 1 leads to the result. $\qquad\square$

**Lemma C.3.** *Denote* $D$ *the diameter of the constraint set* $\mathcal{C}$. *For any* $t \in [T]$, *it holds that*

$$\left\|\sum_{s=1}^t (1-\gamma)^{t-s} Z_s\right\| \leq \frac{LD\eta}{\gamma}.$$

*Proof.* We have

$$\left\|\sum_{s=1}^t (1-\gamma)^{t-s} Z_s\right\| \leq \sum_{s=1}^t (1-\gamma)^{t-s}\|Z_s\|$$
$$= \sum_{s=1}^t (1-\gamma)^{t-s}\|\nabla F(\mathbf{x}_{s-1}) - \nabla F(\mathbf{x}_s)\|$$
$$\leq \sum_{s=1}^t (1-\gamma)^{t-s} L\|\mathbf{x}_{s-1} - \mathbf{x}_s\|$$
$$= \sum_{s=1}^t (1-\gamma)^{t-s} L\eta\|\mathbf{x}_{s-1} - \mathbf{u}_{s-1}\|$$
$$\leq \sum_{s=1}^t (1-\gamma)^{t-s} L\eta D$$
$$\leq \frac{LD\eta}{\gamma}.$$

$\square$

**Lemma C.4** (Lemma 5 in Liu et al. (2023)). *For all* $t \in [T]$, *we have* $\|\zeta_t^u\| \leq 2M$. *Furthermore, if* $\|\nabla F(\mathbf{x}_t)\| \leq \frac{M}{2}$, *then the following holds:*

$$\|\zeta_t^b\| \leq 2\sigma^p M^{1-p} \tag{33}$$
$$\mathbb{E}_t\left[\|\zeta_t^u\|^2\right] \leq 10\sigma^p M^{2-p}. \tag{34}$$

**Lemma C.5** (Lemma 10 in Liu et al. (2023)). *Fix any $t \in [T]$. Define*

$$U_s^t := \begin{cases} 0 & s = 0 \\ \text{Sign}(\sum_{i=1}^{s-1} U_i^t) \frac{\langle \sum_{i=1}^{s-1}(1-\gamma)^{t-i}\zeta_i^u, (1-\gamma)^{t-s}\zeta_s^u \rangle}{\|\sum_{i=1}^{s-1}(1-\gamma)^{t-i}\zeta_i^u\|} & s \neq 0 \text{ and } \sum_{i=1}^{s-1}(1-\gamma)^{t-i}\zeta_i^u \neq 0 \\ 0 & s \neq 0 \text{ and } \sum_{i=1}^{s-1}(1-\gamma)^{t-i}\zeta_i^u = 0. \end{cases} \tag{35}$$

*Then, $U_s^t$ is a martingale difference sequence satisfying $|U_s^t| \leq \|(1-\gamma)^{t-s}\zeta_s^u\|$. Also, denote $R_s^t := \|(1-\gamma)^{t-s}\zeta_s^u\|^2 - \mathbb{E}_s\left[\|(1-\gamma)^{t-s}\zeta_s^u\|^2\right]$, which is also a martingale difference sequence. We have*

$$\left\|\sum_{s=1}^{t}(1-\gamma)^{t-s}\zeta_s\right\| \leq \left|\sum_{s=1}^{t} U_s^t\right| + \sqrt{2\left|\sum_{s=1}^{t} R_s^t\right|} + \sqrt{2\sum_{s=1}^{t}\mathbb{E}_s[\|(1-\gamma)^{t-s}\zeta_s^u\|^2]} + \left\|\sum_{s=1}^{t}(1-\gamma)^{t-s}\zeta_s^b\right\| \tag{36}$$

*for all $t \in [T]$.*

**Lemma C.6** (Lemma 11 in Liu et al. (2023)). *Define an event $a_t$ as*

$$a_t := \left\{ \left|\sum_{s=1}^{t} U_s^t\right| \leq \left(\frac{4}{3} + 2\sqrt{\frac{5(\sigma/M)^p}{\gamma}}\right) M \log\frac{4T}{\delta} \text{ or } \sum_{s=1}^{t}\mathbb{E}_s[(U_s^t)^2] > \frac{10\sigma^p M^{2-p}}{\gamma}\log\frac{4T}{\delta} \right\}.$$

*Then,*

$$\Pr[a_t] \geq 1 - \frac{\delta}{2T}, \quad \forall t \in [T],$$

*where $\delta > 0$.*

**Lemma C.7** (Lemma 12 in Liu et al. (2023)). *Define an event $b_t$ as*

$$b_t := \left\{ \left|\sum_{s=1}^{t} R_s^t\right| \leq \left(\frac{16}{3} + 4\sqrt{\frac{5(\sigma/M)^p}{\gamma}}\right) M^2 \log\frac{4T}{\delta} \text{ or } \sum_{s=1}^{t}\mathbb{E}_s[(R_s^t)^2] > \frac{40\sigma^p M^{4-p}}{\gamma}\log\frac{4T}{\delta} \right\}.$$

*Then,*

$$\Pr[b_t] \geq 1 - \frac{\delta}{2T}, \quad \forall t \in [T],$$

*where $\delta > 0$.*

**Lemma C.8.** *Let $\gamma_t = \gamma$, $\eta_t = \eta$, and $\beta_{1,t} = \beta$ for some constants $\gamma > 0$, $\eta > 0$, and $\beta > 0$. Then, for any $\tau \in [T]$,*

$$\eta \sum_{t=1}^{\tau} \mathcal{G}(\mathbf{x}_t) + F(\mathbf{x}_{\tau+1}) \leq F(\mathbf{x}_1) + \frac{L\tau\eta^2 D^2}{2} + \frac{\eta D\beta\|\nabla F(\mathbf{x}_1)\|}{\gamma}$$

$$+ \frac{\eta D\beta}{1-\gamma}\sum_{t=1}^{\tau}\left((1-\gamma)\left\|\sum_{s=1}^{t}(1-\gamma)^{t-s}Z_s\right\| + \gamma\left\|\sum_{s=1}^{t}(1-\gamma)^{t-s}\zeta_s\right\|\right)$$

$$+ \eta D\left(1 - \frac{\beta}{1-\gamma}\right)\sum_{t=1}^{\tau}\|\zeta_t\|.$$

*Proof.* From (15), we have

$$F(\mathbf{x}_{t+1}) \leq F(\mathbf{x}_t) - \eta\mathcal{G}(\mathbf{x}_t) + \eta D\|\nabla F(\mathbf{x}_t) - \hat{\mathbf{g}}_t\| + \frac{L}{2}\eta^2 D^2. \tag{37}$$

Furthermore,

$$\|\nabla F\left(\mathbf{x}_t\right) - \hat{\mathbf{g}}_t\| = \|\nabla F\left(\mathbf{x}_t\right) - \mathbf{g}_t + \mathbf{g}_t - \hat{\mathbf{g}}_t\|$$

$$= \left\|\nabla F\left(\mathbf{x}_t\right) - \mathbf{g}_t + \left(1 - \frac{\beta}{1 - \gamma}\right)\left(\mathbf{g}_t - \overline{\mathbf{g}}_t\right)\right\|$$

$$= \left\|\nabla F\left(\mathbf{x}_t\right) - \mathbf{g}_t + \left(1 - \frac{\beta}{1 - \gamma}\right)\left(\mathbf{g}_t - \nabla F\left(\mathbf{x}_t\right) + \nabla F\left(\mathbf{x}_t\right) - \overline{\mathbf{g}}_t\right)\right\|$$

$$= \left\|\frac{\beta}{1 - \gamma}\left(\nabla F\left(\mathbf{x}_t\right) - \mathbf{g}_t\right) + \left(1 - \frac{\beta}{1 - \gamma}\right)\left(\nabla F\left(\mathbf{x}_t\right) - \overline{\mathbf{g}}_t\right)\right\|$$

$$\leq \left(\frac{\beta}{1 - \gamma}\right)\|\nabla F\left(\mathbf{x}_t\right) - \mathbf{g}_t\| + \left(1 - \frac{\beta}{1 - \gamma}\right)\|\nabla F\left(\mathbf{x}_t\right) - \overline{\mathbf{g}}_t\|. \tag{38}$$

Combining (37) and (38), we have

$$F\left(\mathbf{x}_{t+1}\right) \leq F\left(\mathbf{x}_t\right) - \eta\mathcal{G}(\mathbf{x}_t) + \eta D\left(\left(\frac{\beta}{1 - \gamma}\right)\underbrace{\|\nabla F\left(\mathbf{x}_t\right) - \mathbf{g}_t\|}_{=\|\epsilon_t\|} + \left(1 - \frac{\beta}{1 - \gamma}\right)\underbrace{\|\nabla F\left(\mathbf{x}_t\right) - \overline{\mathbf{g}}_t\|}_{=\|\zeta_t\|}\right)$$

$$+ \frac{L}{2}\eta^2 D^2. \tag{39}$$

Summing the above inequalities from $t = 1$ to $\tau$, we have

$$\eta \sum_{t=1}^{\tau} \mathcal{G}(\mathbf{x}_t) + F\left(\mathbf{x}_{\tau+1}\right)$$

$$\leq F(\mathbf{x}_1) + \frac{L\tau\eta^2 D^2}{2} + \frac{\eta D\beta}{1 - \gamma}\sum_{t=1}^{\tau}\|\epsilon_t\| + \eta D\left(1 - \frac{\beta}{1 - \gamma}\right)\sum_{t=1}^{\tau}\|\zeta_t\|$$

$$\overset{(a)}{\leq} F(\mathbf{x}_1) + \frac{L\tau\eta^2 D^2}{2}$$

$$+ \frac{\eta D\beta}{1 - \gamma}\sum_{t=1}^{\tau}\left\|(1 - \gamma)^t \epsilon_0 + (1 - \gamma)\left(\sum_{s=1}^{t}(1 - \gamma)^{t-s}Z_s\right) + \gamma\left(\sum_{s=1}^{t}(1 - \gamma)^{t-s}\zeta_s\right)\right\|$$

$$+ \eta D\left(1 - \frac{\beta}{1 - \gamma}\right)\sum_{t=1}^{\tau}\|\zeta_t\|$$

$$\leq F(\mathbf{x}_1) + \frac{L\tau\eta^2 D^2}{2}$$

$$+ \frac{\eta D\beta}{1 - \gamma}\sum_{t=1}^{\tau}\left((1 - \gamma)^t\|\epsilon_0\| + (1 - \gamma)\left\|\sum_{s=1}^{t}(1 - \gamma)^{t-s}Z_s\right\| + \gamma\left\|\sum_{s=1}^{t}(1 - \gamma)^{t-s}\zeta_s\right\|\right)$$

$$+ \eta D\left(1 - \frac{\beta}{1 - \gamma}\right)\sum_{t=1}^{\tau}\|\zeta_t\|$$

$$\overset{(b)}{\leq} F(\mathbf{x}_1) + \frac{L\tau\eta^2 D^2}{2} + \frac{\eta D\beta\|\nabla F(\mathbf{x}_1)\|}{\gamma}$$

$$+ \frac{\eta D\beta}{1 - \gamma}\sum_{t=1}^{\tau}\left((1 - \gamma)\left\|\sum_{s=1}^{t}(1 - \gamma)^{t-s}Z_s\right\| + \gamma\left\|\sum_{s=1}^{t}(1 - \gamma)^{t-s}\zeta_s\right\|\right)$$

$$+ \eta D\left(1 - \frac{\beta}{1 - \gamma}\right)\sum_{t=1}^{\tau}\|\zeta_t\|,$$

where (a) uses Lemma C.9 and (b) uses that $\epsilon_0 = -\nabla F(\mathbf{x}_1)$ by the definition of $\epsilon_0$.

$\square$

**Lemma C.9** (Adapted from Lemma 7 in Liu et al. (2023)). *Let $\gamma_t = \gamma$ and $\eta_t = \eta$ for some constants $\gamma > 0$ and $\eta > 0$. Then*

$$\epsilon_t = (1-\gamma)^t \epsilon_0 + (1-\gamma)\left(\sum_{s=1}^t (1-\gamma)^{t-s} Z_s\right) + \gamma\left(\sum_{s=1}^t (1-\gamma)^{t-s} \zeta_s\right).$$

*Proof.* When $t \geq 2$, we have

$$\epsilon_t = \mathbf{g}_t - \nabla F(\mathbf{x}_t) = (1-\gamma)\mathbf{g}_{t-1} + \gamma\bar{\mathbf{g}}_t + (1-\gamma)\left(\nabla f(\mathbf{x}_t; \Xi_t) - \nabla f(\mathbf{x}_{t-1}; \Xi_t)\right) - \nabla F(\mathbf{x}_t)$$
$$= (1-\gamma)\epsilon_{t-1} + (1-\gamma)Z_t + \gamma\zeta_t \tag{40}$$

Recursively expanding the above equation from $t$ back to 1 leads to the result. We also note that (40) holds when $t = 1$.

$\square$

**Theorem C.10.** *Suppose Assumptions 2.3-a, 2.4, and 2.5 hold. Set $\gamma_t = \gamma = T^{\frac{-p}{3p-2}}$, $\beta_{1,t} = \beta = (1-\gamma)(1-T^{\frac{-p}{3p-2}})$, $M = \frac{\sigma}{\gamma^{1/p}} \vee 2G$, and $\eta_t = \eta = \frac{1}{\sqrt{LTD}} \wedge \frac{\gamma}{\beta}\frac{1}{D} \wedge \frac{\sqrt{\gamma}}{D\sqrt{\beta TL}} \wedge \frac{1-\gamma}{20\gamma DTM \log \frac{4T}{\delta}} \wedge \frac{1}{2TD(1-\frac{\beta}{1-\gamma})M(1+\gamma)}$. Then, with probability at least $1 - \delta$, Algorithm 5 has $\frac{1}{T}\sum_{t=1}^T \mathcal{G}(\mathbf{x}_t) = O\left(\frac{\log\frac{T}{\delta}}{T^{\frac{p-1}{3p-2}}}\right)$, where $p \in (1, 2]$.*

*Proof.* First, let us denote $\Phi := F(\mathbf{x}_1) - \min_{\mathbf{x}} F(\mathbf{x}) + 4 + \|\nabla F(\mathbf{x}_1)\|$. Furthermore, define the following events:

$$\mathfrak{E}_\tau^F := \left\{\eta\sum_{s=1}^t \mathcal{G}(\mathbf{x}_s) \leq \Phi, \quad \forall t \leq \tau\right\}; \qquad \mathfrak{E}_\tau^A := \cap_{t=1}^\tau a_t; \qquad \mathfrak{E}_\tau^B := \cap_{t=1}^\tau b_t. \tag{41}$$

Now we are going to use proof by induction to show that $\mathfrak{E}_\tau^E := \mathfrak{E}_\tau^F \cap \mathfrak{E}_\tau^A \cap \mathfrak{E}_\tau^B$ holds for probability at least $1 - \frac{2\tau\delta}{T}$ for any $\tau \in \{0, 1, \ldots, T\}$, which implies that

$$\frac{1}{T}\sum_{t=1}^T \mathcal{G}(\mathbf{x}_t) \leq \frac{\Phi}{\eta T}$$
$$= O\left(\frac{\sqrt{L}D}{\sqrt{T}}\Phi \vee \frac{\beta}{\gamma}\frac{D}{T}\Phi \vee \frac{D\sqrt{L\beta}}{\sqrt{\gamma}\sqrt{T}}\Phi \vee \frac{\gamma DM\log(4T/\delta)}{1-\gamma}\Phi \vee 2D\left(1-\frac{\beta}{1-\gamma}\right)M(1+\gamma)\Phi\right)$$
$$= O\left(\frac{\sqrt{L}D}{T^{1/2}}\Phi \vee \frac{D}{T^{\frac{2(p-1)}{3p-2}}}\Phi \vee \frac{D\sqrt{L}}{T^{\frac{p-1}{3p-2}}}\Phi \vee \frac{\sigma D\log(4T/\delta)}{T^{\frac{p-1}{3p-2}}}\Phi \vee \frac{DG\log(4T/\delta)}{T^{\frac{p}{3p-2}}}\Phi \vee \frac{D\sigma}{T^{\frac{p-1}{3p-2}}}\Phi \vee \frac{DG}{T^{\frac{p}{3p-2}}}\Phi\right),$$

where we used the parameter choice $\gamma = T^{\frac{-p}{3p-2}}$, $M = \frac{\sigma}{\gamma^{1/p}} \vee 2G$, and also that $1 - \frac{\beta}{1-\gamma} = T^{-\frac{p}{3p-2}}$.

When $\tau = 0$, we have $\mathfrak{E}_0^E = \mathfrak{E}_0^F = \{0 \leq \Phi\}$, which is trivially true.

Assume that at time $\tau - 1 \in [T]$, with probability at least $1 - \frac{2(\tau-1)\delta}{T}$, we have that the event $\mathfrak{E}_{\tau-1}^E$ holds. By Lemma C.6 and Lemma C.7, each of the events $\mathfrak{E}_\tau^A$ and $\mathfrak{E}_\tau^B$ holds with probability at least $1 - \frac{\delta}{2T}$. Now consider the event $\mathfrak{E}_{\tau-1}^E \cap \mathfrak{E}_\tau^A \cap \mathfrak{E}_\tau^B$,

which holds with probability at least $1 - \frac{2\tau\delta}{T}$ by the union bound. By Lemma C.8, under $\mathfrak{E}^E_{\tau-1} \cap \mathfrak{E}^A_\tau \cap \mathfrak{E}^B_\tau$, we have

$$
\eta \sum_{t=1}^{\tau} \mathcal{G}(\mathbf{x}_t) + F(\mathbf{x}_{\tau+1}) \leq F(\mathbf{x}_1) + \frac{L\tau\eta^2 D^2}{2} + \frac{\eta D\beta \|\nabla F(\mathbf{x}_1)\|}{\gamma} + \eta D \left(1 - \frac{\beta}{1-\gamma}\right) \sum_{t=1}^{\tau} \|\zeta_t\|
$$

$$
+ \frac{\eta D\beta}{1-\gamma} \sum_{t=1}^{\tau} \left( (1-\gamma) \left\| \sum_{s=1}^{t} (1-\gamma)^{t-s} Z_s \right\| + \gamma \left\| \sum_{s=1}^{t} (1-\gamma)^{t-s} \zeta_s \right\| \right)
$$

$$
\overset{(i)}{\leq} F(\mathbf{x}_1) + \frac{L\tau\eta^2 D^2}{2} + \frac{\eta D\beta \|\nabla F(\mathbf{x}_1)\|}{\gamma} + \eta D \left(1 - \frac{\beta}{1-\gamma}\right) \sum_{t=1}^{\tau} \|\zeta_t\|
$$

$$
+ \eta D\beta\tau \frac{LD\eta}{\gamma} + \frac{\gamma\eta D\beta}{1-\gamma} \sum_{t=1}^{\tau} \left\| \sum_{s=1}^{t} (1-\gamma)^{t-s} \zeta_s \right\|
$$

$$
\overset{(ii)}{\leq} F(\mathbf{x}_1) + \frac{L\tau\eta^2 D^2}{2} + \frac{\eta D\beta \|\nabla F(\mathbf{x}_1)\|}{\gamma} + \eta D \left(1 - \frac{\beta}{1-\gamma}\right) \sum_{t=1}^{\tau} \|\zeta_t\|
$$

$$
+ \eta D\beta\tau \frac{LD\eta}{\gamma} + \frac{20\gamma\eta D\tau M\beta}{1-\gamma} \log \frac{4T}{\delta}
$$

$$
\overset{(iii)}{\leq} F(\mathbf{x}_1) + \frac{L\tau\eta^2 D^2}{2} + \frac{\eta D\beta \|\nabla F(\mathbf{x}_1)\|}{\gamma} + 2\eta D\tau \left(1 - \frac{\beta}{1-\gamma}\right) M(1+\gamma)
$$

$$
+ \eta D\beta\tau \frac{LD\eta}{\gamma} + \frac{20\gamma\eta D\tau M\beta}{1-\gamma} \log \frac{4T}{\delta}, \tag{42}
$$

where (i) is by Lemma C.3, (iii) is due to $\gamma \in (0,1)$ and that $\|\zeta_t\| \leq \|\zeta_t^u\| + \|\zeta_t^b\| \leq 2M + 2\sigma^p M^{1-p} \leq 2M + 2M\gamma$ by Lemma C.4 and the choice of $M$, and (ii) is because by Lemma C.5,

$$
\left\| \sum_{s=1}^{t} (1-\gamma)^{t-s} \zeta_s \right\| \leq \left| \sum_{s=1}^{t} U_s^t \right| + \sqrt{2 \left| \sum_{s=1}^{t} R_s^t \right|} + \sqrt{2 \sum_{s=1}^{t} \mathbb{E}_s[\|(1-\gamma)^{t-s} \zeta_s^u\|^2]} + \left\| \sum_{s=1}^{t} (1-\gamma)^{t-s} \zeta_s^b \right\|
$$

$$
\overset{\clubsuit}{\leq} \left( \frac{4}{3} + 2\sqrt{\frac{5(\sigma/M)^p}{\gamma}} \right) M \log \frac{4T}{\delta} + \sqrt{2 \left( \frac{16}{3} + 4\sqrt{\frac{5(\sigma/M)^p}{\gamma}} \right) M^2 \log \frac{4T}{\delta}}
$$

$$
+ \sqrt{2 \sum_{s=1}^{t} (1-\gamma)^{2(t-s)} (10\sigma^p M^{2-p}) + \frac{1}{\gamma} 2\sigma^p M^{1-p}}
$$

$$
\overset{\spadesuit}{\leq} 6M \log \frac{4T}{\delta} + 6M \sqrt{\log \frac{4T}{\delta}} + 5M + 2M \leq 20M \log \frac{4T}{\delta}.
$$

Above, $\clubsuit$ holds for the following reason. By the choice of $M$, we have $\|\nabla f(\mathbf{x}_t)\| \leq \frac{M}{2}$, since $\mathbf{x}_t \in \mathcal{C}$ and $\|\nabla f(\mathbf{x}_t)\| \leq G \leq \frac{M}{2}$.

Hence, we can use Lemma C.4 to get $\|\zeta_t^u\| \leq 2M$, $\|\zeta_t^b\| \leq 2\sigma^p M^{1-p}$, and $\mathbb{E}_t\left[\|\zeta_t^u\|^2\right] \leq 10\sigma^p M^{2-p}$ for any $t \in [T]$. Then, from Lemma C.5 and Lemma C.6, we further have

$$
\sum_{s=1}^{t} \mathbb{E}_s[(U_s^t)^2] \leq \sum_{s=1}^{t} \mathbb{E}_s\left[\|(1-\gamma)^{t-s} \zeta_s^u\|^2\right] \leq \frac{10\sigma^p M^{2-p}}{1-(1-\gamma)^2} \leq \frac{10\sigma^p M^{2-p}}{\gamma} \tag{43}
$$

$$
\sum_{s=1}^{t} \mathbb{E}_s[(R_s^t)^2] \leq \sum_{s=1}^{t} \mathbb{E}_s\left[\|(1-\gamma)^{t-s} \zeta_s^u\|^4\right] \leq \frac{40\sigma^p M^{4-p}}{1-(1-\gamma)^4} \leq \frac{40\sigma^p M^{4-p}}{\gamma}. \tag{44}
$$

Combining (43), (44), Lemma C.6, and Lemma C.7, we have

$$
\left| \sum_{s=1}^{t} U_s^t \right| \leq \left( \frac{4}{3} + 2\sqrt{\frac{5(\sigma/M)^p}{\gamma}} \right) M \log \frac{4T}{\delta} \quad \text{and} \quad \left| \sum_{s=1}^{t} R_s^t \right| \leq \left( \frac{16}{3} + 4\sqrt{\frac{5(\sigma/M)^p}{\gamma}} \right) M^2 \log \frac{4T}{\delta},
$$

for any $t \leq \tau$, under the event $\mathfrak{E}^E_{\tau-1} \cap \mathfrak{E}^A_\tau \cap \mathfrak{E}^B_\tau \cap \mathfrak{E}^C_\tau$. For $\spadesuit$, we used $\frac{(\sigma/M)^p}{\gamma} \leq 1$ and that $\gamma \in (0,1)$.

To continue, let

$$\eta = \min\left\{ \frac{1}{\sqrt{LT}D}, \frac{\gamma}{\beta}\frac{1}{D}, \frac{\sqrt{\gamma}}{D\sqrt{\beta T L}}, \frac{1-\gamma}{20\gamma DTM\beta\log\frac{4T}{\delta}}, \frac{1}{2TD(1-\frac{\beta}{1-\gamma})M(1+\gamma)} \right\}.$$

Then, from (42), we obtain

$$
\begin{aligned}
\eta\sum_{t=1}^\tau \mathcal{G}(\mathbf{x}_t) &\leq F(\mathbf{x}_1) + \frac{L\tau\eta^2 D^2}{2} + \frac{\eta D\beta\|\nabla F(\mathbf{x}_1)\|}{\gamma} + 2\eta D\tau\left(1 - \frac{\beta}{1-\gamma}\right)M(1+\gamma) \\
&\quad + \eta D\beta\tau\frac{LD\eta}{\gamma} + \frac{20\gamma\eta D\tau M\beta}{1-\gamma}\log\frac{4T}{\delta}, \\
&\leq \underbrace{F(\mathbf{x}_1) - \min_\mathbf{x} F(\mathbf{x}) + 4 + \|\nabla F(\mathbf{x}_1)\|}_{=\Phi},
\end{aligned}
$$

which means that $\mathfrak{E}^F_\tau$ holds under the event $\mathfrak{E}^E_{\tau-1} \cap \mathfrak{E}^A_\tau \cap \mathfrak{E}^B_\tau$. This also implies that $\mathfrak{E}^E_\tau = \mathfrak{E}^F_\tau \cap \mathfrak{E}^A_\tau \cap \mathfrak{E}^B_\tau$ holds for probability at least $1 - \frac{2\tau\delta}{T}$. Therefore, we have completed the induction.

$\square$

## C.4 LION++ and MUON++

In this section, we present the algorithmic specifications of LION++ and MUON++, which are obtained as special cases of Algorithm 6.

---

**Algorithm 10** LION++

**Required:** Momentum parameters $\beta_1, \beta_2$, step-sizes $\{\eta_t\}$, weight decay parameter $\lambda$, and clipping parameter $M$.
**Initialize:** $\mathbf{m}_0 = 0$ and $\mathbf{x}_1 \in \mathcal{C}$.
**for** $t = 1, 2, \ldots$ **do**
  Sample $\Xi_t \sim \mathcal{D}$.
  Compute $\bar{\mathbf{g}}_t = \left(1 \wedge \frac{M}{\|\nabla f(\mathbf{x}_t;\Xi_t)\|}\right) \nabla f(\mathbf{x}_t;\Xi_t)$.
  Update $\mathbf{c}_t = \beta_1\mathbf{m}_{t-1} + (1 - \beta_1)\bar{\mathbf{g}}_t + \beta_1\mathbb{1}_{t\geq 2}\left(\nabla f(\mathbf{x}_t;\Xi_t) - \nabla f(\mathbf{x}_{t-1};\Xi_t)\right)$.
  Update $\mathbf{x}_{t+1} = \mathbf{x}_t - \eta_t(\text{sign}(\mathbf{c}_t) + \lambda\mathbf{x}_t)$.
  Update $\mathbf{m}_t = \beta_2\mathbf{m}_{t-1} + (1 - \beta_2)\bar{\mathbf{g}}_t + \beta_2\mathbb{1}_{t\geq 2}\left(\nabla f(\mathbf{x}_t;\Xi_t) - \nabla f(\mathbf{x}_{t-1};\Xi_t)\right)$.
**end for**

---

**Algorithm 11** MUON++

**Required:** Momentum parameter $\mu$, step-sizes $\{\eta_t\}$, weight decay parameter $\lambda$, and clipping parameter $M$.
**Initialize:** $\mathbf{B}_0 = 0$ and $\mathbf{X}_1 \in \mathcal{C}$.
**for** $t = 1, 2, \ldots$ **do**
  Sample $\Xi_t \sim \mathcal{D}$.
  Compute $\mathbf{G}_t = \left(1 \wedge \frac{M}{\|\nabla f(\mathbf{X}_t;\Xi_t)\|}\right) \nabla f(\mathbf{X}_t;\Xi_t)$.
  Update $\mathbf{B}_t = \mu\mathbf{B}_{t-1} + \mathbf{G}_t + \frac{\mu}{1-\mu}\mathbb{1}_{t\geq 2}\left(\nabla f(\mathbf{X}_t;\Xi_t) - \nabla f(\mathbf{X}_{t-1};\Xi_t)\right)$.
  Update $\mathbf{O}_t = \arg\min_{\mathbf{A}\in\mathcal{O}_{m\times n}}\|\mathbf{A} - \mathbf{B}_t\|_F$.
  Update $\mathbf{X}_{t+1} = \mathbf{X}_t - \eta_t(\mathbf{O}_t + \lambda\mathbf{X}_t)$.
**end for**

---

## C.5 Proof of Theorem 4.4

**Lemma C.11.** *Denote $D$ the diameter of the constraint set $\mathcal{C}$. For any $t \in [T]$, with probability at least $1 - \frac{\delta}{T}$, it holds that*

$$\left\|\sum_{s=1}^t (1-\gamma)^{t-s}Z_s\right\| \leq 9\eta LD\left(\frac{\log\frac{3T}{\delta}}{\sqrt{2\gamma - \gamma^2}}\right).$$

*Proof.* The proof is a modification of the Proof of Lemma 9 in Liu et al. (2023). Recall that $Z_s := \mathbb{1}\{t \geq 2\}\left(\nabla f(\mathbf{x}_s, \Xi_s) - \nabla f(\mathbf{x}_{s-1}, \Xi_s) - (\nabla F(\mathbf{x}_s) - \nabla F(\mathbf{x}_{s-1}))\right)$. Therefore, $\mathbb{E}_s[Z_s] = 0$, and hence $\mathbb{E}_s[(1-\gamma)^{t-s}Z_s] = 0$.

Furthermore,

$$
\begin{aligned}
\|(1-\gamma)^{t-s} Z_s\| &\leq \|\nabla f(\mathbf{x}_s, \Xi_s) - \nabla f(\mathbf{x}_{s-1}, \Xi_s)\| + \|\nabla F(\mathbf{x}_s) - \nabla F(\mathbf{x}_{s-1})\| \\
&\overset{(i)}{\leq} 2L\|\mathbf{x}_s - \mathbf{x}_{s-1}\| \\
&= 2L\eta\|\mathbf{x}_{s-1} - \mathbf{u}_{s-1}\| \\
&\overset{(ii)}{\leq} 2\eta LD,
\end{aligned}
$$

where (i) is due to the L-smoothness and (ii) uses that the diameter of the constraint set is bounded by $D$.

Furthermore,

$$
\begin{aligned}
\mathbb{E}_s \left[ \|(1-\gamma)^{t-s} Z_s\|^2 \right] &\leq (1-\gamma)^{2(t-s)} \mathbb{E}_s \left[ \|\nabla f(\mathbf{x}_s, \Xi_s) - \nabla f(\mathbf{x}_{s-1}, \Xi_s) - (\nabla F(\mathbf{x}_s) - \nabla F(\mathbf{x}_{s-1}))\|^2 \right] \\
&\leq (1-\gamma)^{2(t-s)} \mathbb{E}_s \left[ \|\nabla f(\mathbf{x}_s, \Xi_s) - \nabla f(\mathbf{x}_{s-1}, \Xi_s)\|^2 \right] \\
&\leq (1-\gamma)^{2(t-s)} L^2 \|\mathbf{x}_s - \mathbf{x}_{s-1}\|^2 \\
&\leq (1-\gamma)^{2(t-s)} \eta^2 L^2 D^2,
\end{aligned}
$$

where the last inequality uses the update and the bound of the diameter of the constraint set $D$.

Since $Z_s$ is a martingale difference sequence, we can use Lemma 14 in Liu et al. (2023) (replicated in Lemma C.12 below) to obtain that the following holds for all $\tau \in [t]$ with probability at least $1 - \delta$,

$$
\begin{aligned}
\left\| \sum_{s=1}^{\tau} (1-\gamma)^{t-s} Z_s \right\| &\leq 6\eta LD \log \frac{3}{\delta} + 3\sqrt{\sum_{s=1}^{\tau} (1-\gamma)^{2(t-s)} \eta^2 L^2 D^2 \log \frac{3}{\delta}} \\
&\leq 3\eta LD \left( 2\log \frac{3}{\delta} + \sqrt{\frac{\log \frac{3}{\delta}}{1-(1-\gamma)^2}} \right) \\
&\leq 9\eta LD \left( \frac{\log \frac{3}{\delta}}{\sqrt{2\gamma - \gamma^2}} \right).
\end{aligned}
$$

Then, set $\tau \leftarrow t$ and $\delta \leftarrow \delta/T$ leads to the result.

$\square$

**Lemma C.12** (Lemma 14 in Liu et al. (2023)). *Suppose $X_{t \in [T]}$ is a martingale sequence adapted to the filtration $\mathcal{F}_{t \in [T]}$ in a Hilbert Space satisfying $\|X_t\| \leq R$ almost surely for some constant $R$ and $\mathbb{E}[\|X_t\|^2|\mathcal{F}_{t-1}] \leq \sigma^2$ almost surely for some constant $\sigma_t^2$. Then with probability at least $1 - \delta$, for any $t \in [T]$,*

$$
\left\| \sum_{s=1}^{t} X_s \right\| \leq 3R \log \frac{3}{\delta} + 3\sqrt{\sum_{s=1}^{T} \sigma_s^2 \log \frac{3}{\delta}}.
$$

**Theorem C.13.** *Suppose Assumptions 2.3-a, 2.3-b, 2.4, and 2.5 hold. Set $\gamma_t = \gamma = T^{\frac{-p}{2p-1}}$, $\beta_{1,t} = \beta = (1-\gamma)\left(1 - T^{\frac{-p}{2p-1}}\right)$, $M = \frac{\sigma}{\gamma^{1/p}} \vee 2G$, and $\eta_t = \eta = \frac{1}{\sqrt{LTD}} \wedge \frac{\gamma}{\beta}\frac{1}{D} \wedge \frac{\gamma^{1/4}}{D\sqrt{9TL\beta \log \frac{3T}{\delta}}} \wedge \frac{1-\gamma}{20\gamma DTM\beta \log \frac{4T}{\delta}} \wedge \frac{1}{2TD(1-\frac{\beta}{1-\gamma})M(1+\gamma)}$. Then, with probability at least $1-\delta$, Algorithm 6 has $\frac{1}{T}\sum_{t=1}^{T} \mathcal{G}(\mathbf{x}_t) = O\left( \frac{\log \frac{T}{\delta}}{T^{\frac{p-1}{2p-1}}} \right)$, where $p \in (1, 2]$.*

*Proof.* The proof uses some of the technical tools developed in the analysis of Normalized SGD with clipping and momentum by Liu et al. (2023).

First, let us denote $\Phi := F(\mathbf{x}_1) - \min_{\mathbf{x}} F(\mathbf{x}) + 4 + \|\nabla F(\mathbf{x}_1)\|$. Furthermore, define the following events:

$$\mathfrak{E}_\tau^F := \left\{ \eta \sum_{s=1}^{t} \mathcal{G}(\mathbf{x}_s) \le \Phi, \quad \forall t \le \tau \right\}; \qquad \mathfrak{E}_\tau^A := \cap_{t=1}^{\tau} a_t; \qquad \mathfrak{E}_\tau^B := \cap_{t=1}^{\tau} b_t;$$

$$\mathfrak{E}_\tau^C := \cap_{t=1}^{\tau} c_t := \cap_{t=1}^{\tau} \left\{ \left\| \sum_{s=1}^{t} (1-\gamma)^{t-s} Z_s \right\| \le 9\eta L D \left( \frac{\log \frac{3T}{\delta}}{\sqrt{2\gamma - \gamma^2}} \right) \right\}.$$

Now we are going to use proof by induction to show that $\mathfrak{E}_\tau^E := \mathfrak{E}_\tau^F \cap \mathfrak{E}_\tau^A \cap \mathfrak{E}_\tau^B \cap \mathfrak{E}_\tau^C$ holds for probability at least $1 - \frac{2\tau\delta}{T}$ for any $\tau \in \{0, 1, \ldots, T\}$, which implies that

$$\frac{1}{T} \sum_{t=1}^{T} \mathcal{G}(\mathbf{x}_t) \le \frac{\Phi}{\eta T}$$

$$= O\left( \frac{\sqrt{L}D}{\sqrt{T}}\Phi \vee \frac{\beta}{\gamma}\frac{D}{T}\Phi \vee \frac{\sqrt{L\beta \log \frac{3T}{\delta}}D}{\gamma^{1/4}\sqrt{T}}\Phi \vee \frac{\gamma D M \beta \log(4T/\delta)}{1-\gamma}\Phi \vee 2D\left(1 - \frac{\beta}{1-\gamma}\right)M(1+\gamma) \right)$$

$$= \Phi \cdot O\left( \frac{\sqrt{L}D}{T^{1/2}} \vee \frac{D}{T^{\frac{p-1}{2p-1}}} \vee \frac{\sqrt{L\log\frac{3T}{\delta}}D}{T^{\frac{3p-2}{4(2p-1)}}} \vee \frac{\sigma D \log(4T/\delta)}{T^{\frac{p-1}{2p-1}}} \vee \frac{DG\log(4T/\delta)}{T^{\frac{p}{2p-1}}} \vee \frac{D\sigma}{T^{\frac{p-1}{2p-1}}} \vee \frac{DG}{T^{\frac{p}{2p-1}}} \right),$$

where we used the parameter choice $\gamma = T^{\frac{-p}{2p-1}}$ and $M = \frac{\sigma}{\gamma^{1/p}} \vee 2G$ for the last line and also that $1 - \frac{\beta}{1-\gamma} = T^{-\frac{p}{2p-1}}$.

When $\tau = 0$, we have $\mathfrak{E}_0^E = \mathfrak{E}_0^F = \{0 \le \Phi\}$, which is trivially true.

Assume that at time $\tau - 1 \in [T]$, with probability at least $1 - \frac{2(\tau-1)\delta}{T}$, we have that the event $\mathfrak{E}_{\tau-1}^E$ holds. By Lemma C.6, Lemma C.7, and Lemma C.11, each of the events $\mathfrak{E}_\tau^A$ and $\mathfrak{E}_\tau^B$ holds with probability at least $1 - \frac{\delta}{2T}$, while $\mathfrak{E}_\tau^C$ holds with probability at least $1 - \frac{\delta}{T}$. Now consider the event $\mathfrak{E}_{\tau-1}^E \cap \mathfrak{E}_\tau^A \cap \mathfrak{E}_\tau^B \cap \mathfrak{E}_\tau^C$, which holds with probability at least $1 - \frac{2\tau\delta}{T}$ by the union bound. By Lemma C.8, under $\mathfrak{E}_{\tau-1}^E \cap \mathfrak{E}_\tau^A \cap \mathfrak{E}_\tau^B \cap \mathfrak{E}_\tau^C$, we have

$$\eta \sum_{t=1}^{\tau} \mathcal{G}(\mathbf{x}_t) + F(\mathbf{x}_{\tau+1}) \le F(\mathbf{x}_1) + \frac{L\tau\eta^2 D^2}{2} + \frac{\eta D \beta \|\nabla F(\mathbf{x}_1)\|}{\gamma} + \eta D\left(1 - \frac{\beta}{1-\gamma}\right)\sum_{t=1}^{\tau}\|\zeta_t\|$$

$$+ \frac{\eta D \beta}{1-\gamma}\sum_{t=1}^{\tau}\left((1-\gamma)\left\|\sum_{s=1}^{t}(1-\gamma)^{t-s}Z_s\right\| + \gamma\left\|\sum_{s=1}^{t}(1-\gamma)^{t-s}\zeta_s\right\|\right)$$

$$\overset{(i)}{\le} F(\mathbf{x}_1) + \frac{L\tau\eta^2 D^2}{2} + \frac{\eta D \beta \|\nabla F(\mathbf{x}_1)\|}{\gamma} + \eta D\left(1 - \frac{\beta}{1-\gamma}\right)\sum_{t=1}^{\tau}\|\zeta_t\|$$

$$+ \eta D \beta \tau \frac{9\eta L D \log\frac{3T}{\delta}}{\sqrt{2\gamma - \gamma^2}} + \frac{\gamma\eta D \beta}{1-\gamma}\sum_{t=1}^{\tau}\left\|\sum_{s=1}^{t}(1-\gamma)^{t-s}\zeta_s\right\|$$

$$\overset{(ii)}{\le} F(\mathbf{x}_1) + \frac{L\tau\eta^2 D^2}{2} + \frac{\eta D \beta \|\nabla F(\mathbf{x}_1)\|}{\gamma} + \eta D\left(1 - \frac{\beta}{1-\gamma}\right)\sum_{t=1}^{\tau}\|\zeta_t\|$$

$$+ \eta D \beta \tau \frac{9\eta L D \log\frac{3T}{\delta}}{\sqrt{2\gamma - \gamma^2}} + \frac{20\gamma\eta D \tau M \beta}{1-\gamma}\log\frac{4T}{\delta}$$

$$\overset{(iii)}{\le} F(\mathbf{x}_1) + \frac{L\tau\eta^2 D^2}{2} + \frac{\eta D \beta \|\nabla F(\mathbf{x}_1)\|}{\gamma} + 2\eta D\tau\left(1 - \frac{\beta}{1-\gamma}\right)M(1+\gamma)$$

$$+ \eta D \beta \tau \frac{9\eta L D \log\frac{3T}{\delta}}{\sqrt{\gamma}} + \frac{20\gamma\eta D \tau M \beta}{1-\gamma}\log\frac{4T}{\delta}, \tag{45}$$

where (i) holds under $\mathfrak{E}_\tau^C$, (iii) is due to $\gamma \in (0,1)$ and that $\|\zeta_t\| \le \|\zeta_t^u\| + \|\zeta_t^b\| \le 2M + 2\sigma^p M^{1-p} \le 2M + 2M\gamma$ by

Lemma C.4 and the choice of $M$, and (ii) is because by Lemma C.5,

$$\left\|\sum_{s=1}^{t}(1-\gamma)^{t-s}\zeta_s\right\| \leq \left|\sum_{s=1}^{t}U_s^t\right| + \sqrt{2\left|\sum_{s=1}^{t}R_s^t\right|} + \sqrt{2\sum_{s=1}^{t}\mathbb{E}_s[\|(1-\gamma)^{t-s}\zeta_s^u\|^2]} + \left\|\sum_{s=1}^{t}(1-\gamma)^{t-s}\zeta_s^b\right\|$$

$$\overset{\clubsuit}{\leq} \left(\frac{4}{3}+2\sqrt{\frac{5(\sigma/M)^p}{\gamma}}\right)M\log\frac{4T}{\delta} + \sqrt{2\left(\frac{16}{3}+4\sqrt{\frac{5(\sigma/M)^p}{\gamma}}\right)M^2\log\frac{4T}{\delta}}$$

$$+ \sqrt{2\sum_{s=1}^{t}(1-\gamma)^{2(t-s)}(10\sigma^pM^{2-p}) + \frac{1}{\gamma}2\sigma^pM^{1-p}}$$

$$\overset{\spadesuit}{\leq} 6M\log\frac{4T}{\delta} + 6M\sqrt{\log\frac{4T}{\delta}} + 5M + 2M \leq 20M\log\frac{4T}{\delta}.$$

Above, $\clubsuit$ holds for the following reason. By the choice of $M$, we have $\|\nabla f(\mathbf{x}_t)\| \leq \frac{M}{2}$, since $\mathbf{x}_t \in \mathcal{C}$ and $\|\nabla f(\mathbf{x}_t)\| \leq G \leq \frac{M}{2}$. Hence, we can use Lemma C.4 to get $\|\zeta_t^u\| \leq 2M$, $\|\zeta_t^b\| \leq 2\sigma^pM^{1-p}$, and $\mathbb{E}_t[\|\zeta_t^u\|^2] \leq 10\sigma^pM^{2-p}$ for any $t \in [T]$. Then, from Lemma C.5 and Lemma C.6, we further have

$$\sum_{s=1}^{t}\mathbb{E}_s[(U_s^t)^2] \leq \sum_{s=1}^{t}\mathbb{E}_s\left[\|(1-\gamma)^{t-s}\zeta_s^u\|^2\right] \leq \frac{10\sigma^pM^{2-p}}{1-(1-\gamma)^2} \leq \frac{10\sigma^pM^{2-p}}{\gamma} \tag{46}$$

$$\sum_{s=1}^{t}\mathbb{E}_s[(R_s^t)^2] \leq \sum_{s=1}^{t}\mathbb{E}_s\left[\|(1-\gamma)^{t-s}\zeta_s^u\|^4\right] \leq \frac{40\sigma^pM^{4-p}}{1-(1-\gamma)^4} \leq \frac{40\sigma^pM^{4-p}}{\gamma}. \tag{47}$$

Combining (46), (47), Lemma C.6, and Lemma C.7, we have

$$\left|\sum_{s=1}^{t}U_s^t\right| \leq \left(\frac{4}{3}+2\sqrt{\frac{5(\sigma/M)^p}{\gamma}}\right)M\log\frac{4T}{\delta} \text{ and } \left|\sum_{s=1}^{t}R_s^t\right| \leq \left(\frac{16}{3}+4\sqrt{\frac{5(\sigma/M)^p}{\gamma}}\right)M^2\log\frac{4T}{\delta},$$

for any $t \leq \tau$, under the event $\mathfrak{E}_{\tau-1}^E \cap \mathfrak{E}_\tau^A \cap \mathfrak{E}_\tau^B \cap \mathfrak{E}_\tau^C$. For $\spadesuit$, we used $\frac{(\sigma/M)^p}{\gamma} \leq 1$ and that $\gamma \in (0,1)$. To continue, let

$$\eta = \min\left\{\frac{1}{\sqrt{LTD}}, \frac{\gamma}{\beta}\frac{1}{D}, \frac{\gamma^{1/4}}{D\sqrt{9TL\beta\log\frac{3T}{\delta}}}, \frac{1-\gamma}{20\gamma DTM\beta\log\frac{4T}{\delta}}, \frac{1}{2TD\left(1-\frac{\beta}{1-\gamma}\right)M(1+\gamma)}\right\}.$$

Then, from (45), we obtain

$$\eta\sum_{t=1}^{\tau}\mathcal{G}(\mathbf{x}_t) \leq F(\mathbf{x}_1) + \frac{L\tau\eta^2D^2}{2} + \frac{\eta D\beta\|\nabla F(\mathbf{x}_1)\|}{\gamma} + 2\eta D\tau\left(1-\frac{\beta}{1-\gamma}\right)M(1+\gamma)$$

$$+ \eta D\beta\tau\frac{9\eta LD\log\frac{3T}{\delta}}{\sqrt{\gamma}} + \frac{20\gamma\eta D\tau M\beta}{1-\gamma}\log\frac{4T}{\delta},$$

$$\leq \underbrace{F(\mathbf{x}_1) - \min_{\mathbf{x}}F(\mathbf{x}) + 4 + \|\nabla F(\mathbf{x}_1)\|}_{=\Phi},$$

which means that $\mathfrak{E}_\tau^F$ holds under the event $\mathfrak{E}_{\tau-1}^E \cap \mathfrak{E}_\tau^A \cap \mathfrak{E}_\tau^B \cap \mathfrak{E}_\tau^C$. This also implies that $\mathfrak{E}_\tau^E = \mathfrak{E}_\tau^F \cap \mathfrak{E}_\tau^A \cap \mathfrak{E}_\tau^B \cap \mathfrak{E}_\tau^C$ holds for probability at least $1 - \frac{2\tau\delta}{T}$. Therefore, we have completed the induction.

$\square$

# D Hyperparameter Settings for nanoGPT

For the nanoGPT experiments we use the following training configuration:

- **nanoGPT:** https://github.com/karpathy/nanoGPT/tree/master

For all algorithms, the learning rate, and weight decay were tuned using grid search. For LION+ and MUON+, we additionally tuned the gradient clipping threshold. All experiments were run on one NVIDIA A100 GPU. Table 2 shows the full hyperparameter search space, while Table 3 presents the chosen hyperparameters for each algorithm along with the complete experimental configuration. Figure 2 shows the training and validation loss curves averaged across five different seed values.

| | |
|---|---|
| Learning rate | {1e-1, 5e-2,1e-2, 5e-3, 1e-3, 5e-4, 1e-4, 5e-5, 1e-5} |
| Gradient clipping threshold | {1, 2, 3, 4, 5, $\infty$} |
| Weight decay | {1e-1, 1e-2, 1e-3} |
| Batch size | 64 |

*Table 2.* Hyperparameter search space for nanoGPT training.

| | Lion | LION+ | Muon | MUON+ |
|---|---|---|---|---|
| Max Learning rate | 5e-5 | 5e-5 | 5e-2 | 5e-2 |
| Min Learning rate | 5e-8 | 5e-8 | 5e-4 | 5e-4 |
| Gradient clipping threshold | $\infty$ | 4 | $\infty$ | 5 |
| Weight decay | 1e-3 | 1e-2 | 1e-1 | 1e-1 |
| $(\beta_1, \beta_2)$ | (0.95, 0.98) | (0.95, 0.98) | - | - |
| $\mu$ | - | - | 0.95 | 0.95 |
| Nesterov | - | - | No | No |
| Batch size | | 64 | | |
| Block size | | 256 | | |
| Layers | | 6 | | |
| Heads | | 6 | | |
| Embedding dimension | | 384 | | |
| Warmup steps | | 100 | | |
| Learning rate schedule | | cosine | | |
| Dropout | | 0.2 | | |

*Table 3.* Hyperparameters selected by grid search and setup for nanoGPT training.

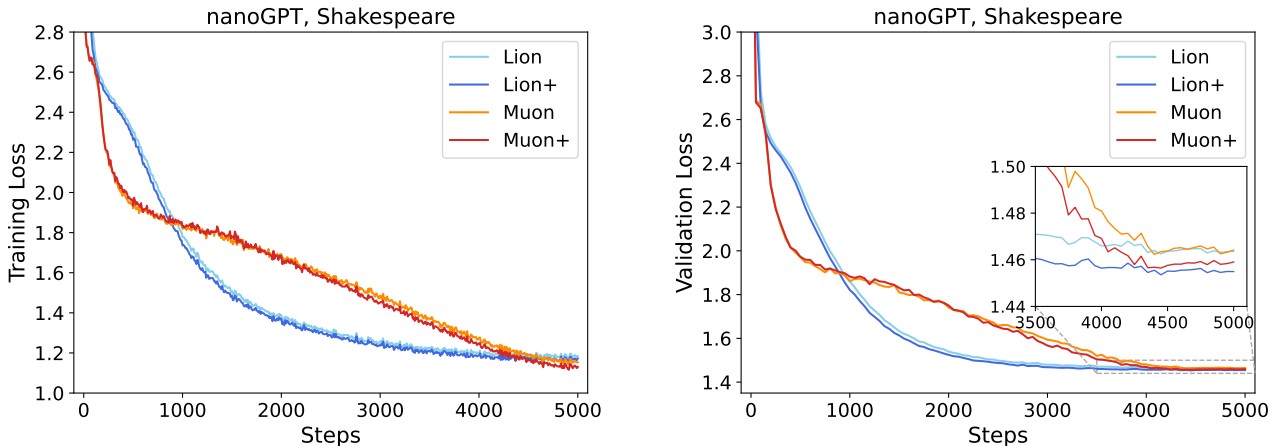

*Figure 2.* Loss curves for nanoGPT training on the Shakespeare dataset. We plotted the training and validation loss per 10 and 50 steps, respectively. The results are averaged across five seed values.

# E    Additional Experiments: Empirical Investigation of Stochastic FW with Clipping and Variance Reduction

## E.1    Synthetic Heavy-tailed Dataset

In this section, we conduct numerical experiments on a synthetic function to assess the performance of LION++ and MUON++. Specifically, we use the experimental setup from Hübler et al. (2025). All experiments are conducted on one NVIDIA A100 GPU. The selected hyperparameters are reported in Tables 4 and 5.

**Lion vs. LION++.** We evaluate the performance of Lion and LION++ on the simple function $F(\mathbf{x}) = \frac{1}{2}\|\mathbf{x}\|^2$, $\mathbf{x} \in \mathbb{R}^d$, for $d = 1$ and $d = 1000$, where $\|\cdot\|$ denotes the Euclidean norm. We introduce three distinct types of noise to the gradient: standard Normal, component-wise symmetrized Pareto with $p = 2.5$, and component-wise symmetrized Pareto with $p = 1.5$. We note that the standard Normal distribution serves as a model for light-tailed noise, while the Pareto distributions are employed to capture heavy-tailed noise, with finite ($p = 2.5$) and infinite ($p = 1.5$) variance. We run each algorithm $10^5$ times for $T = 100$ iterations, and use as convergence criterion the average gradient norm across all $T$ iterations.

Figures 3-5 illustrate the convergence behavior of Lion and LION++ by displaying the median, $\delta$ and $1 - \delta$ (with $\delta = 10^{-4}$) quantiles of the algorithm runs, based on the average gradient norm at $T = 100$.

**Muon vs. MUON++.** To assess the performance of Muon and MUON++, we use the function $F(\mathbf{X}) = \frac{1}{2}\|\mathbf{X}\|^2$, $\mathbf{X} \in \mathbb{R}^{n \times n}$, with $n = 2$ and $n = 30$, where $\|\cdot\|$ denotes the Frobenius norm. We apply the same three types of noise as in the previous section and run each algorithm $10^5$ times for $T = 100$ iterations.

Figures 6-8 show the convergence behavior of Muon and MUON++ by displaying the median, $\delta$ and $1 - \delta$ (with $\delta = 10^{-4}$) quantiles of the algorithm runs, based on the average gradient norm at $T = 100$.

**Results.** The results indicate that in the small-dimensional setting, with $d = 1$ and $n = 2$, respectively, all comparison algorithms perform similarly under light-tailed noise. When the noise is heavy-tailed but has bounded variance, Lion++ and Muon++ show a slight performance advantage. In the case of heavy-tailed noise with unbounded variance, Lion++ and Muon++ demonstrate improved convergence and greater robustness to noise, as reflected in the upper quantiles. For the high-dimensional case, with d=1000 and n=30, respectively, Lion++ and Muon++ tend to achieve lower average gradient norms under both light-tailed and heavy-tailed noise conditions, with the improvement being more notable in the unbounded variance case.

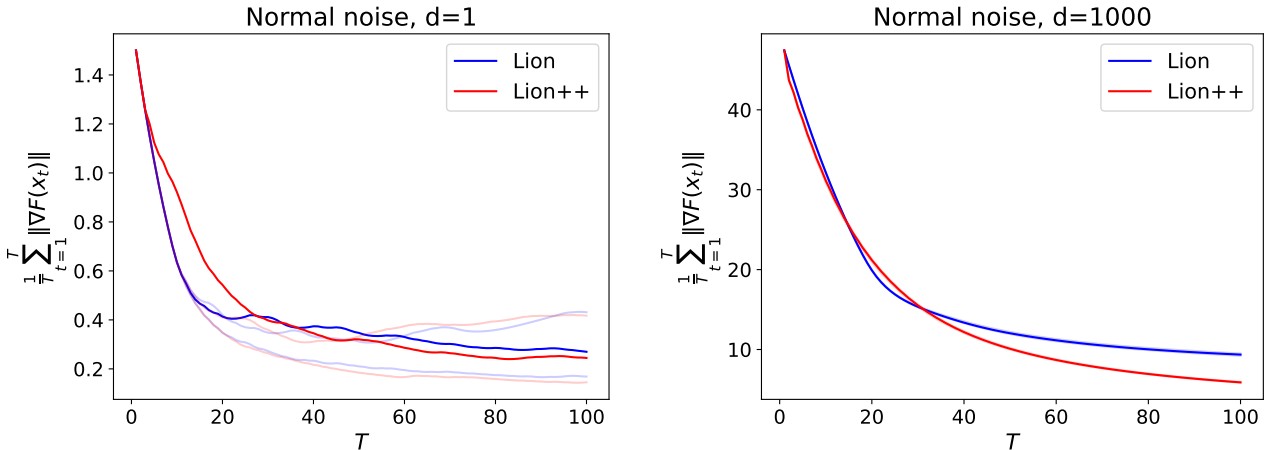

*Figure 3.* Standard Normal noise.

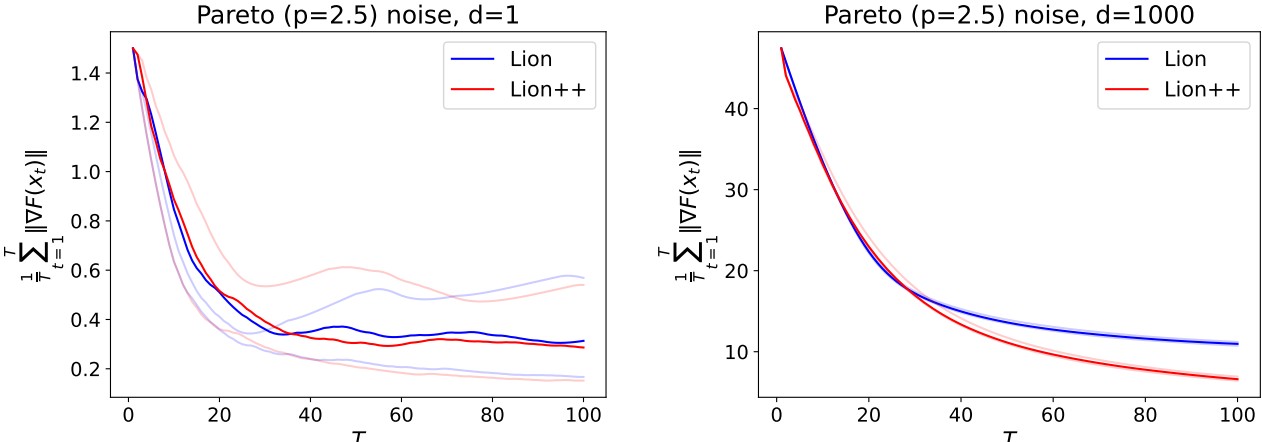

*Figure 4.* Pareto (p=2.5) noise.

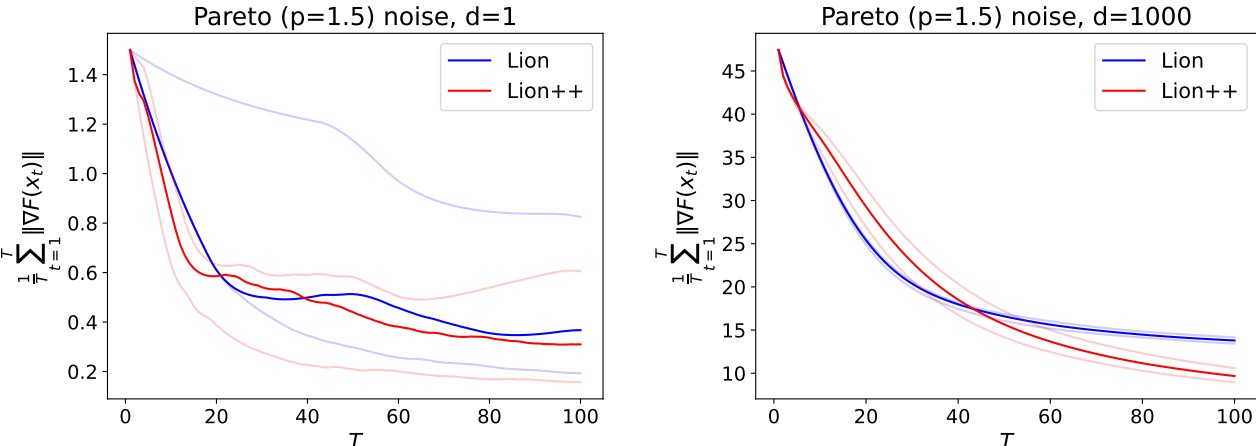

*Figure 5.* Pareto (p=1.5) noise.

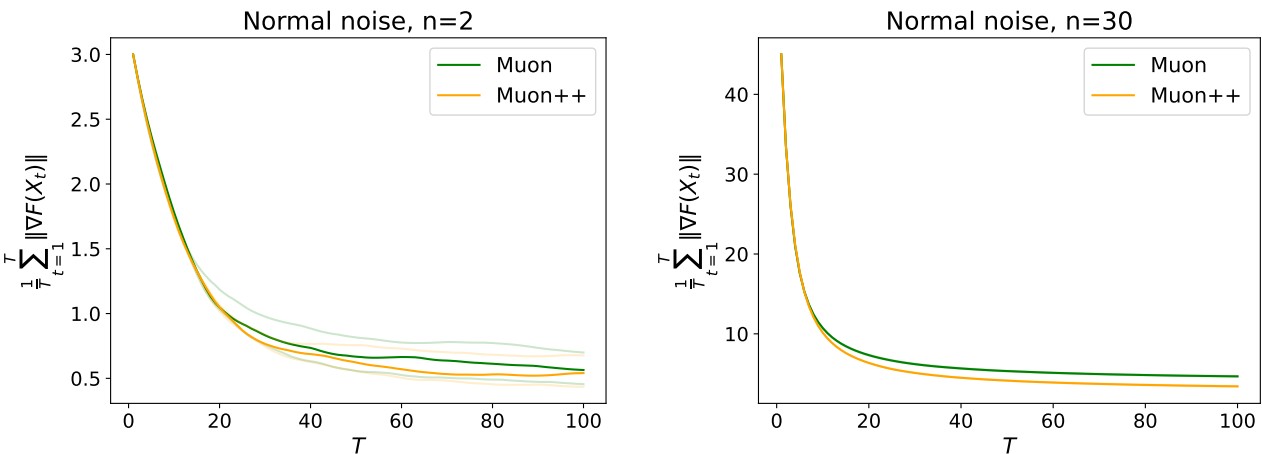

*Figure 6.* Standard Normal noise.

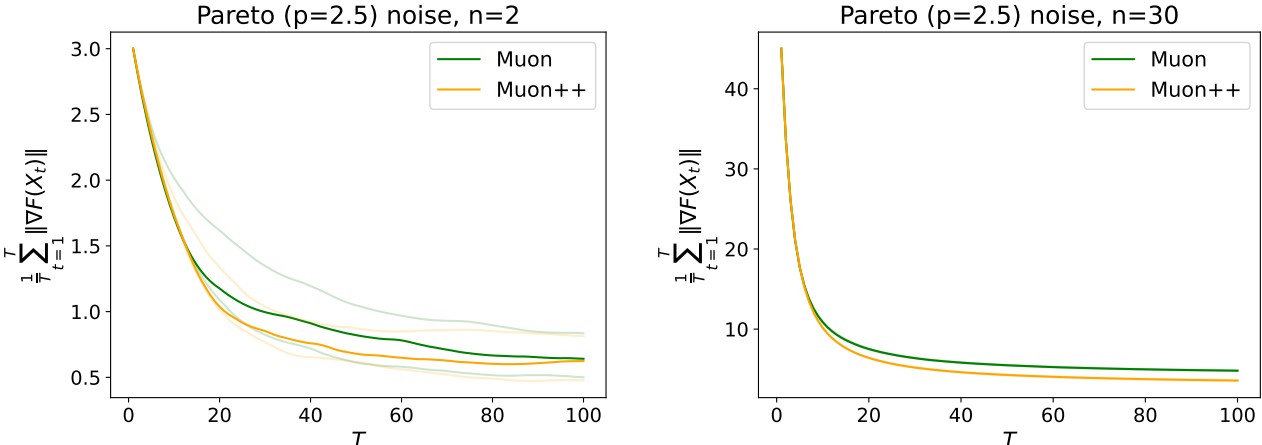

*Figure 7.* Pareto (p=2.5) noise.

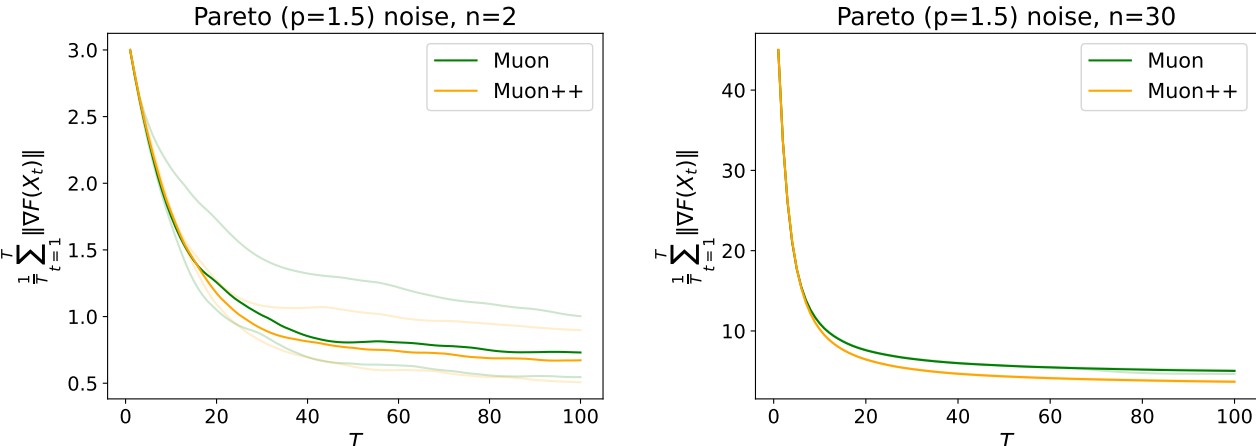

*Figure 8.* Pareto (p=1.5) noise.

|  |  | Lion | Lion++ |
|---|---|---|---|
| Normal noise, d=1 | Learning rate | 1e-1 | 1e-1 |
|  | Gradient clipping | $\infty$ | 1 |
|  | Weight decay | 1 | 1 |
| Normal noise, d=1000 | Learning rate | 5e-2 | 1e-1 |
|  | Gradient clipping | $\infty$ | 1 |
|  | Weight decay | 1 | 3 |
| Pareto (p=2.5) noise, d=1 | Learning rate | 1e-1 | 1e-1 |
|  | Gradient clipping | $\infty$ | 1 |
|  | Weight decay | 1 | 2 |
| Pareto (p=2.5) noise, d=1000 | Learning rate | 5e-2 | 1e-1 |
|  | Gradient clipping | $\infty$ | 3 |
|  | Weight decay | 1 | 1 |
| Pareto (p=1.5) noise, d=1 | Learning rate | 5e-2 | 1e-1 |
|  | Gradient clipping | $\infty$ | 2 |
|  | Weight decay | 1 | 1 |
| Pareto (p=1.5) noise, d=1000 | Learning rate | 5e-2 | 1e-1 |
|  | Gradient clipping | $\infty$ | 3 |
|  | Weight decay | 1 | 1 |

*Table 4.* Hyperparameters selected for the synthetic function $F(\mathbf{x}) = \frac{1}{2}\|\mathbf{x}\|^2$.

|  |  | Muon | Muon++ |
|---|---|---|---|
| Normal noise, n=2 | Learning rate | 1e-1 | 1e-1 |
|  | Gradient clipping | $\infty$ | 1 |
|  | Weight decay | 1 | 1 |
| Normal noise, n=30 | Learning rate | 5e-1 | 5e-1 |
|  | Gradient clipping | $\infty$ | 1 |
|  | Weight decay | 1 | 1 |
| Pareto (p=2.5) noise, n=2 | Learning rate | 1e-1 | 1e-1 |
|  | Gradient clipping | $\infty$ | 1 |
|  | Weight decay | 1 | 1 |
| Pareto (p=2.5) noise, n=30 | Learning rate | 5e-1 | 5e-1 |
|  | Gradient clipping | $\infty$ | 1 |
|  | Weight decay | 1 | 1 |
| Pareto (p=1.5) noise, n=2 | Learning rate | 1e-1 | 1e-1 |
|  | Gradient clipping | $\infty$ | 5 |
|  | Weight decay | 1 | 1 |
| Pareto (p=1.5) noise, n=30 | Learning rate | 5e-1 | 5e-1 |
|  | Gradient clipping | $\infty$ | 1 |
|  | Weight decay | 1 | 1 |

*Table 5.* Hyperparameters selected for the synthetic function $F(\mathbf{X}) = \frac{1}{2}\|\mathbf{X}\|^2$.

### E.2 Multi-class Image Classification

In this section, we evaluate the performance of LION++ and MUON++ on multi-class image classification by training a ResNet18 model (He et al., 2015) on the CIFAR-10 (Krizhevsky & Hinton, 2009) dataset. All experiments are conducted on one NVIDIA A100 GPU. Table 6 shows the full hyperparameter search space, and the selected hyperparameters by grid search are reported in Table 7.

**Results.** Figure 9 displays the test loss and test accuracy curves over 200 epochs. The results show that LION++ consistently outperforms Lion in terms of test loss throughout training. MUON++ also exhibits an overall improvement in test loss compared to Muon, with the difference being particularly notable during the first half of training. In terms of test accuracy, both LION++ and MUON++ exhibit overall improved test accuracy throughout training compared to their respective baselines, with LION++ showing a more consistent and pronounced advantage over Lion during the final epochs.

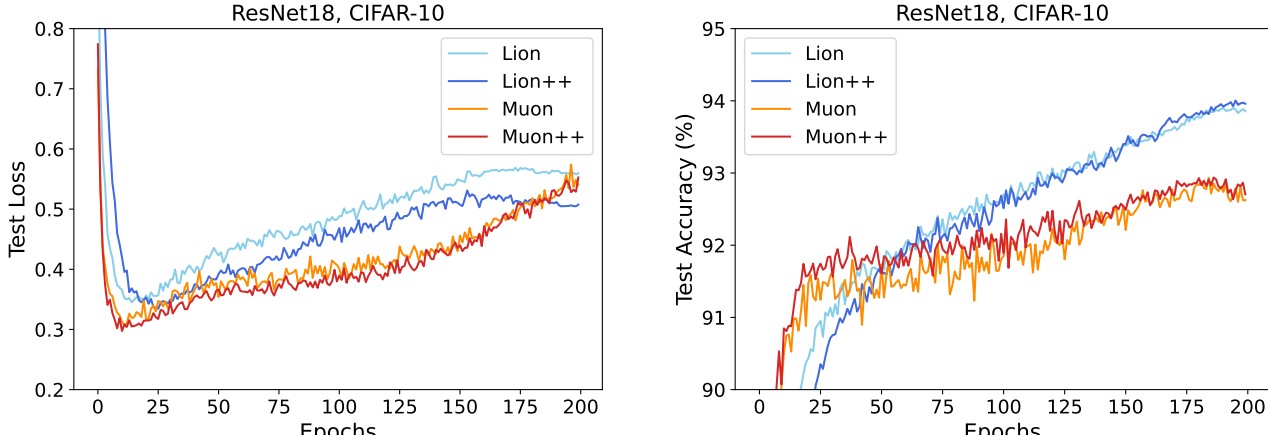

*Figure 9.* Test loss and test accuracy curves for ResNet18 training on the CIFAR-10 dataset. The results are averaged across five seed values.

| Learning rate | {1e-1, 5e-2, 1e-2, 5e-3, 1e-3, 5e-4, 1e-4, 5e-5, 1e-5} |
|---|---|
| Gradient clipping threshold | $\{1, 2, 3, 4, 5, \infty\}$ |
| Weight decay | {1e-1, 1e-2, 1e-3} |
| Batch size | 128 |

*Table 6.* Hyperparameter search space for ResNet18 training.

| | Lion | LION++ | Muon | MUON++ |
|---|---|---|---|---|
| Learning rate | 1e-4 | 1e-4 | 5e-2 | 5e-2 |
| Gradient clipping threshold | $\infty$ | 5 | $\infty$ | 5 |
| Weight decay | 1e-2 | 1e-3 | 1e-3 | 1e-3 |
| $(\beta_1, \beta_2)$ | (0.9, 0.99) | (0.9, 0.99) | - | - |
| $\mu$ | - | - | 0.95 | 0.95 |
| Nesterov | - | - | No | No |
| Batch size | | 128 | | |
| Learning rate schedule | | cosine | | |

*Table 7.* Hyperparameters selected by grid search and setup for ResNet18 training.

