# OpenReview forum: "Lions and Muons: Optimization via Stochastic Frank-Wolfe under Heavy-Tailed Noise"
_ICML.cc/2026/Conference — ICML 2026 regular_

### Official Review · Reviewer_mKKS · 2026-03-01

**Soundness:** 3
**Presentation:** 3
**Significance:** 2
**Originality:** 2
**Overall Recommendation:** 3
**Confidence:** 4

**Summary:**

Lion and Muon are two emerging optimization methods for deep learning. However, the theoretical guarantees of these two methods are not well understood. In this paper, the authors provide a unifying perspective by interpreting these two methods through the lens of Stochastic Frank-Wolfe (SFW). Note that SFW is originally designed for constrained optimization. However, Lion and Muon are mainly used for unconstrained optimization. Interestingly, the authors show that Lion and Muon with weight decay can be viewed as instances of SFW with some manual constraints. From such a connection, the authors provide some convergence guarantees for Lion and Muon with weight decay by using the theoretical analysis framework of SFW. Furthermore, the authors also extend SFW and these two methods into the setting with heavy-tailed noises. Some experiments have also been conducted to verify the performance of these methods.

**Compliance With Llm Reviewing Policy:**

Affirmed.

**Final Justification:**

Due to the concern about the significance, I keep the original score.

**Key Questions For Authors:**

1) It would be better if the authors could highlight the technical challenges in their analysis, instead of only discussing the differences between their results and related results.
2) In Figure 9, I am a bit confused why the test loss of Muon (and Muon+) is better than Lion (Lion+) at least during the first 150 epochs, but the test accuracy of Muon (and Muon+) is worse.

**Limitations:**

yes

**Strengths And Weaknesses:**

## Soundness and Presentation

This paper is well-written and easy to follow. The only suggestion on the writing is that a specific section for introducing and discussing related work may be better. In the current version, the authors incorporate the introduction and discussion of related work into the Introduction section, which is acceptable but may not be deep enough.

This paper is also technically sound. All theoretical guarantees have detailed proofs, though I cannot check the details of all proofs. The authors also provide some experiments to demonstrate the performance of their method, especially the proposed variants for heavy-tailed noises.

## Significance and Originality
Since Lion and Muon have received great attention recently due to their practical performance, I first agree that it is also appealing to establish theoretical guarantees for them. However, there are some concerns about the significance and originality of this paper.
1) There already exist some theoretical guarantees on the convergence of Muon and Lion, especially through the lens of linear minimization oracle (LMO) [1][2]. Since SFW is one of the most famous LMO-based algorithms, given these existing studies, it is not surprising to further establish some connection between these two methods and SFW. Actually, Pethick et al. [1] have also provided some connection between these methods and SFW. Although the authors emphasized that previous studies may not consider some specific tricks used by Lion or Muon, e.g., the momentum extrapolation, such extensions seem to be incremental.
2) I agree that it is valuable to establish theories of these emerging methods. However, I believe that a good theory should provide some additional insights into how to use these methods, instead of only telling us they can converge. A good example is the work of Pethick et al. [1], which provides some suggestions on using LMO with different norm constraints for different layers. Although the heavy-tailed extension of SFW and these two methods in this paper provide some insights into the practical use of  Lion or Muon, the improvement shown in the experimental results is tiny. Moreover, it seems that such an extension is based on some standard technique.

[1] T. Pethick et al. Training Deep Learning Models with Norm-Constrained LMOs. In ICML, 2025.

[2] A. Riabinin. Gluon: Making Muon & Scion great again! (bridging theory and practice of LMO-based optimizers for LLMs). In arXiv, 2025.

---

> ### Author Rebuttal · Authors · 2026-03-31
>
> We appreciate the time the reviewer took to carefully read our paper and provide feedback.
>
> - **Significance and originality**: While our investigation of the dynamics of Lions and Muons as a specific instance of Stochastic Frank-Wolfe in Section 3 is motivated by a sequence of prior works, including Bernstein and Newhouse (2024), Xie and Li (2024), and Pethick et al. (2025), the primary objective of our work diverges in scope and contributions. Compared to the work of Pethick et al. (2025), which investigates norm-constrained LMOs for training deep neural networks, our objective consists of enhancing the theoretical foundations of Stochastic Frank-Wolfe methods in general constrained optimization settings. In this context, our proposed Stochastic Frank-Wolfe variant with the additional extrapolation step enables convergence guarantees for Lion and Muon with Nesterov momentum, which were not established in their work. In fact, the extrapolation step is essential for the unification of Lion and Muon under SFW. More importantly, the latter part of our paper departs entirely from the setup in Pethick et al. (2025). We introduce **three additional SFW variants** with improved convergence guarantees. These include a variance-reduced variant which relaxes the assumptions and batch-size requirements of previous work of SFW with variance-reduction, as well as two variants that can accommodate heavy-tailed noise. To the best of our knowledge, our later results constitute the first convergence guarantees for SFW methods in **non-convex optimization under heavy-tailed noise**.  Overall, the contribution of this work lies in enhancing the understanding and scope of SFW methods, with all algorithms and associated convergence guarantees stated within the general SFW formulation, rather than in investigating specific optimizer instances for practical deep learning deployment.
>
> - **Discussion of technical challenges**: We thank the reviewer for this suggestion. We note that the emphasis on specific technical challenges inherently involves a subjective element, and our presentation prioritizes a clear and faithful presentation of the analysis and obtained results. In particular, many of the key technical challenges are reflected through the comparisons with prior work, in terms of required assumptions and obtained complexities. However, we also emphasize that the main limitations of our analysis are discussed in the final section of the paper.
>
> - **Discussion of related work**: We note that the introduction section already includes a detailed discussion of the most relevant prior work to properly motivate our contributions, while additional references are discussed throughout all sections of the paper to facilitate comparison with existing results. Furthermore, a more comprehensive review of related work is presented in Appendix A.
>
> - **Test loss and accuracy in Figure 9**: The test accuracy depends only on the correctness of the predicted label, whereas the cross-entropy loss is sensitive to the full confidence distribution, i.e., the probabilities assigned to every possible label. Specifically, the cross-entropy loss is heavily penalized by confident errors, but it is also reduced by high-confidence correct predictions. As a result, a model can achieve higher accuracy but also higher loss if its mistakes are made with high confidence. In our experiments, Lion (and Lion++) attains higher accuracy but also higher loss, which suggests that although it predicts the correct label more often, its errors tend to be more overconfident. In contrast, Muon (and Muon++) achieves lower loss but also lower accuracy, indicating more moderate and better-calibrated predictions, which avoid large penalties from incorrect predictions.
>
> We hope our response has addressed the reviewer’s concerns and clarified the positioning of our contributions within the broader literature. We would be happy to provide additional clarification if needed.

---

> > ### Author Rebuttal · Reviewer_mKKS · 2026-04-01
> >
> > Thanks for the authors' responses. However, I am still concerned about the significance of this paper, i.e., the theoretical results do not provide sufficient new insights for using Lions and Muons.

---

> > > ### Author Response · Authors · 2026-04-06
> > >
> > > We are thankful for the reviewer’s input and glad that our rebuttal has addressed most of the reviewer's concerns. On the other hand, we are very sorry that we are not able to see where the reviewer's comment "_the theoretical results do not provide sufficient new insights for using Lions and Muons_" is coming from. We have carefully contextualized the position of our paper as early as in our current abstract, and from the introduction, we have provided a sufficient overview of the presentation and have highlighted our major contributions regarding Stochastic Frank-Wolfe in Section 4, where the by-product of Section 4 is the generation of new variants of deep learning optimizers, such as Lion+ and Muon+, built upon our results in Section 3. We also note that in the abstract, the introduction, and Section 3, we have reiterated that our results in Section 3 are motivated by prior perspective from several related works, and our contributions further reinforce that. For example, our proposed variants in Section 4 relax the large batch-size requirements, as we highlighted in the paper. Furthermore, extending beyond the standard bounded variance assumption, we showed that the proposed algorithmic variants can accommodate heavy-tailed noise, which is not addressed by the existing analyses for Lion and Muon. Since we have carefully contextualized our contributions and main message throughout our paper, we think that any perceived bias regarding the main message perhaps comes from the current paper title. If that is the case, we are open to modifying the paper title. We genuinely hope the reviewer can kindly reevaluate our work and consider increasing the score. Thank you!

---

### Official Review · Reviewer_Chtx · 2026-03-05

**Soundness:** 3
**Presentation:** 3
**Significance:** 3
**Originality:** 3
**Overall Recommendation:** 4
**Confidence:** 3

**Summary:**

This paper studies the connection between recently proposed deep learning optimizers, notably Lion and Muon, and the classical stochastic Frank–Wolfe (SFW) optimization framework. The authors show that these optimizers can be interpreted as instances of stochastic Frank–Wolfe methods under different norm constraints. In particular, Lion corresponds to SFW under an $\ell_\infty$-ball constraint, while Muon with weight decay corresponds to SFW under a spectral-norm ball constraint.

Building on this observation, the paper develops a general stochastic Frank–Wolfe algorithm with extrapolation and establishes convergence guarantees for the Frank–Wolfe gap in nonconvex settings. The analysis is further extended to the case of heavy-tailed stochastic gradients, where the authors propose clipped stochastic Frank–Wolfe variants and derive high-probability convergence guarantees. These theoretical developments lead to modified practical optimizers (LION+ and MUON+). Preliminary experiments on nanoGPT training suggest that the clipped variants can achieve improved training efficiency compared with the original optimizers.

Overall, the paper aims to provide a unified theoretical perspective on modern deep learning optimizers through the lens of stochastic Frank–Wolfe methods, while also extending the theoretical understanding of SFW under heavy-tailed noise.

**Compliance With Llm Reviewing Policy:**

Affirmed.

**Final Justification:**

Thank you for the response. I will keep my score.

**Key Questions For Authors:**

The theoretical analysis focuses on constrained optimization formulations. Could the authors further clarify how this perspective translates to typical deep learning training setups that do not explicitly enforce norm constraints?

The experiments show improvements using clipping-based variants. How sensitive are the results to the clipping threshold and related hyperparameters?

Do the authors expect the benefits of LION+ and MUON+ to persist for larger-scale models or more challenging datasets?

Could the authors comment on whether the proposed SFW framework suggests other potentially useful optimizer designs beyond Lion and Muon?

**Limitations:**

Yes.

**Strengths And Weaknesses:**

Strengths
(1) Interesting unifying perspective.
The paper provides a clear and conceptually appealing interpretation of Lion and Muon as instances of stochastic Frank–Wolfe methods under different norm constraints. This connection is insightful and helps clarify the implicit optimization geometry behind these recently proposed optimizers.

(2) Nontrivial theoretical development.
Beyond the reinterpretation, the paper develops a general stochastic Frank–Wolfe algorithm with extrapolation and establishes convergence guarantees for the Frank–Wolfe gap in nonconvex settings. The extension to heavy-tailed gradient noise is technically meaningful and addresses an important regime relevant to large-scale stochastic optimization.

(3) Robust optimization variant with theoretical guarantees.
The clipped stochastic Frank–Wolfe methods proposed in the paper provide a natural way to handle heavy-tailed gradients. The theoretical results include high-probability convergence guarantees and sample complexity bounds, which extend the existing literature on stochastic Frank–Wolfe methods.

(4) Practical implications.
The theory motivates practical optimizer variants (LION+ and MUON+) that incorporate clipping mechanisms. The preliminary experiments suggest that these modifications may provide improvements in training stability and efficiency.

Overall, the work connects modern deep learning optimization practice with classical optimization theory in a coherent manner, which may be valuable to both communities.

Weaknesses

(1) Limited experimental validation.
The empirical evaluation is relatively limited. Experiments are mainly conducted on nanoGPT training with a single dataset (Shakespeare). Given that the paper aims to provide insights into widely used optimizers in modern deep learning, broader empirical validation would strengthen the claims. In particular, it would be helpful to see experiments on larger language models, vision benchmarks, or other standard tasks.

(2) Practical relevance of the constrained formulation.
The theoretical framework relies on viewing the optimization problem as a norm-constrained problem, while many practical training setups are effectively unconstrained. The paper discusses weight decay as inducing such constraints, but the practical implications of this interpretation could be elaborated further.

(3) Modest novelty in the optimizer reinterpretation itself.
While the identification of Lion and Muon as instances of stochastic Frank–Wolfe is elegant, the connection between steepest-descent directions under dual norms and Frank–Wolfe updates is relatively classical in optimization. Thus, part of the novelty lies more in the systematic reinterpretation and extension rather than in a fundamentally new optimization principle.

(4) Heavy-tailed assumptions and parameter choices.
The heavy-tailed analysis assumes bounded $p$-th moments of stochastic gradients. While this is a reasonable theoretical model, the practical implications for choosing clipping thresholds and related hyperparameters are not fully explored. It would be helpful to better understand how these theoretical parameters translate to real training settings.

---

> ### Author Rebuttal · Authors · 2026-03-31
>
> We thank the reviewer for the positive feedback and the constructive comments.
> - **Constrained formulation**: Through the Stochastic Frank-Wolfe interpretation of Lion and Muon, we demonstrate in the first part of our work that the addition of the weight decay term is **equivalent** to solving a constrained optimization problem, even in practical training setups that appear unconstrained. This provides an interpretation of weight decay not only as a regularization term, but as a mechanism that naturally induces the hard constraints required by the SFW, thus offering deeper insight into the optimization dynamics. A precise characterization of the practical implications of these constraints, and their specific role in shaping the model’s internal representations requires further investigation and remains an important research direction.
> - **Novelty**: We clarify that our contributions aim to enhance the theoretical landscape of **Stochastic Frank-Wolfe algorithms** and not only provide a reinterpretation of Lion and Muon. In particular, the generality of the proposed SFW methods enables us to derive convergence guarantees uniformly across all induced variants from any convex and compact constraint set. In the second part of our work, we introduce **three additional SFW variants** with improved convergence guarantees. Specifically, our variance-reduced algorithm matches the best-known SFO complexity bound of $O(1/\epsilon^3)$ in the literature of SFW with variance reduction [1], [2], [3]. Simultaneously, our variant uses an amortized average batch size of at most 2, thereby avoiding the need for large batches throughout the iterations, unlike [1], [2]. Furthermore, in contrast to [3], our theoretical guarantees hold without requiring Hessian-based assumptions or structural constraints on the data distribution. Furthermore, we extend our SFW to the heavy-tailed setup by proposing two variants that can accommodate bounded $p$-th moment noise. To the best of our knowledge, [4] is the only prior work to address the heavy-tailed setting in SFW algorithms. However, [4] adopt a different set of assumptions on the stochastic noise and assume convexity of the function. Therefore, our work is the first to systematically study Stochastic Frank-Wolfe methods for **non-convex optimization under heavy-tailed noise**.
> - **Heavy-tailed assumptions and parameters**: Please refer to our response to reviewer ndSg.
> - **Experimental evaluation**: We have provided experimental results across **three** diverse tasks and model architectures that consistently support our theoretical results. However, we emphasize that all the algorithmic variants and corresponding convergence guarantees in the paper are formulated in terms of SFW algorithms. Accordingly, our experimental evaluation is intended to validate the robustness of our SFW theory rather than to introduce a new deep learning optimizer for competitive benchmarking. Given that thorough hyperparameter tuning at large scale is a computationally intensive task, we have focused our empirical efforts on demonstrating that our theoretical findings hold across a variety of architectures and tasks. Overall, our proposed algorithms are fairly robust to hyperparameter selection, yielding consistently higher performance than the baselines across various step sizes and clipping levels. To ensure reliability, we plot the average run across five seeds for each configuration. The consistent improvement indicates that the results are not tied to a specific model or dataset, and thus we would expect to see similar improvements in larger-scale models. In any case, we note that this work’s contribution is primarily theoretical in scope, and it should be evaluated on that basis.
> - **Other optimizers**: We appreciate the reviewer's insightful question. We think a potential follow-up is to extend our Stochastic Frank-Wolfe results to the composite Frank-Wolfe setting. We note that a very recent work by Jiang et al. [5] demonstrates that their proposed spectral clipping method can be viewed as a Stochastic Frank-Wolfe method on a composite objective, and their algorithm has shown some enhanced empirical performance in LLM training. Therefore, we believe it is valuable to extend our SFW contributions in the second part of our paper to composite objective settings, with the goal of potentially not only advancing the foundations of SFW by improving the complexity for composite objectives (under both light-tailed and heavy-tailed regimes) but also leading to enhanced variants of optimizers for LLM training.
>
> [1] Yurtsever et al. (2019), Conditional gradient methods via stochastic path-integrated differential estimator
>
> [2] Hassani et al. (2020), Stochastic conditional gradient++
>
> [3] Zhang et al. (2020), One sample stochastic frank-wolfe
>
> [4] Tang et al. (2022), High-probability bounds for robust stochastic frank-wolfe algorithm
>
> [5] Jiang et al. (2026), Enhancing LLM Training via Spectral Clipping

---

> > ### Author Rebuttal · Reviewer_Chtx · 2026-04-02
> >
> > Thank you for the response; your clarification that the main contribution goes beyond reinterpretation to advancing SFW theory (including VR and heavy-tailed nonconvex settings) is helpful, but the practically testable implications of the “weight decay induces hard constraints” view, the operational guidance for parameter/clipping choices under heavy-tailed assumptions, and stronger, fairer empirical comparisons (including standard clipping baselines and matched tuning) still need to be strengthened. Nevertheless,  my recommendation as a weak accept remains.

---

> > > ### Author Response · Authors · 2026-04-06
> > >
> > > We would like to thank the reviewer again for their encouraging feedback and effort in reviewing our work! With respect to the reviewer's remarks, we note that the practical implications of the weight decay term are highlighted through the improved empirical convergence observed in recent works, such as [1], [2], [3]. Moreover, our theoretical results for the heavy-tailed regime indicate that the clipping threshold should be chosen to balance the bias-variance trade-off, by capturing the magnitude of the true gradient signal via $G$, while also suppressing extreme gradient noise via $\sigma$. Similarly, the step size choice demonstrates the required scaling in terms of the horizon $T$ and problem-dependent quantities. Both of these results are consistent with prior work that investigates the bounded $p$-th moment noise assumption, see e.g., [4], [5], [6]. Regarding the clipping baselines, we note that the employed gradient norm clipping is the most common clipping method in practice due to its simplicity. Accordingly, our experiments aim to illustrate the benefits of this standard clipping method under heavy-tailed noise, thus providing empirical validation for our theory.
> > >
> > >
> > >
> > > [1] Liu et al. (2025), Muon is scalable for LLM training
> > >
> > > [2] Chen et al. (2024), Lion Secretly Solves Constrained Optimization: As Lyapunov Predicts
> > >
> > > [3] Zhao et al. (2025), Deconstructing What Makes a Good Optimizer for Language Models
> > >
> > > [4] Cutkosky and Mehta (2021), High-probability bounds for Non-Convex Stochastic Optimization with Heavy Tails
> > >
> > > [5] Zhang et al. (2020), Why are adaptive methods good for attention models?
> > >
> > > [6] Liu et al. (2023), Breaking the Lower Bound with (Little) Structure: Acceleration in Non-Convex Stochastic Optimization with Heavy-Tailed Noise

---

### Official Review · Reviewer_ndSg · 2026-03-06

**Soundness:** 3
**Presentation:** 3
**Significance:** 2
**Originality:** 2
**Overall Recommendation:** 4
**Confidence:** 3

**Summary:**

This paper presents a Frank-Wolfe (FW) perspective on Lion and Muon with weight decay, showing that these methods can be interpreted as stochastic FW algorithms over norm-ball constrained problems. The paper further argues that, in these settings, convergence of the FW gap implies convergence to a KKT point. Building on this viewpoint, the authors develop stochastic FW methods for non-convex optimization under both bounded-variance and heavy-tailed noise, including clipped and variance-reduced variants, and instantiate these results in Lion/Muon-style algorithms. The empirical section reports improvements for clipped variants on nanoGPT/Shakespeare.

**Compliance With Llm Reviewing Policy:**

Affirmed.

**Final Justification:**

I thank the authors for the clarification provided during the rebuttal phase. After careful consideration of the response and the subsequent discussion, my overall evaluation of the paper remains the same.

**Key Questions For Authors:**

1. What simple, robust tuning rules for clipping thresholds and schedule parameters work reliably in practice, and how sensitive are your results to these choices across models and datasets?
  1. How do the proposed variants compare against strong and directly relevant baselines (AdamW, AdamW+clipping, normalized/clipped SGD, and other recent optimizers) in both steps-to-quality and wall-clock?
  1. Do you observe empirical evidence that the assumptions needed for the heavy-tailed analysis are approximately satisfied during training (e.g., boundedness on the induced constraint set)?

**Limitations:**

Yes.

**Strengths And Weaknesses:**

**Strengths**
- The paper offers a clean and conceptually useful unification of Lion and Muon+weight-decay by viewing them as instances of stochastic Frank--Wolfe over norm-ball constraint sets, which deepens understanding of these popular optimizers.
 - Using the Frank--Wolfe gap as the stationarity measure is methodologically appropriate for projection-free constrained optimization and keeps the analysis aligned with the FW literature.
 - The connection from FW-gap convergence to KKT-type stationarity provides a clear optimization interpretation of what the algorithms converge to under the constrained formulation.
 - The heavy-tailed gradient setting is practically relevant, and the clipping-based (and clipping+VR) FW variants provide a principled robustness story with high-probability guarantees.
  - The proposed robust variants are instantiated explicitly as optimizer-like procedures (e.g., LION+/MUON+ and LION++/MUON++), which makes the theoretical contributions easier to translate into practice.
  - The work is likely to be of interest to both theory and practitioner communities by offering a perspective that can inspire additional constrained-optimization views of modern optimizers.

**Weaknesses**
- The heavy-tailed guarantees require technical schedules and assumptions (including boundedness-type conditions on the constraint set) that may not reflect typical deep-learning tuning and may limit how directly the theory explains real training runs.
- The empirical evidence is limited in scale and baseline breadth, and largely compares to unclipped Lion/Muon; stronger baselines and wall-clock reporting are needed to establish practical significance.
- The tuning guidance for clipping thresholds and schedule parameters is not yet sufficiently distilled, which affects usability and reproducibility for practitioners.
- Some aspects of novelty rely on synthesizing known tools (clipping, variance reduction) within the FW framework; clearer differentiation from closely related works would strengthen the originality case.

---

> ### Author Rebuttal · Authors · 2026-03-31
>
> We thank the reviewer for the positive evaluation and thoughtful comments.
>
> - **Boundedness of the constraint set**: We would like to clarify that the first part of our paper shows an exact one-to-one correspondence between the optimization dynamics of algorithms for unconstrained optimization problems (i.e., Lion and Muon with weight decay) and the optimization dynamics of our Stochastic Frank–Wolfe over a bounded convex constraint set. In the second part of our paper, we propose a few algorithms to advance the foundations of Stochastic Frank–Wolfe for general bounded convex sets (under both light-tailed and heavy-tailed regimes). Building on the connection of the dynamics established in the first part of the paper, these results may translate into new optimization dynamics for unconstrained optimization problems (e.g., Lion+, Muon+).
>
> - **Hyperparameters in the heavy-tailed regime**: We emphasize that it is common in the analysis of optimization methods under heavy-tailed noise to state guarantees in terms of hyperparameters that depend on the problem constants $G$ and $\sigma$, see e.g. [1], [2], [3]. For the clipping threshold $M$, the theoretical result implies that it should be chosen large enough to avoid excessive bias from over-clipping informative gradients, while still being small enough to mitigate the effect of large, noisy stochastic gradients. The quantities $G$ and $\sigma$ capture this tradeoff. Similarly, the choice of the step size $\eta$ reflects a scaling using problem-dependent quantities and the horizon $T$ to obtain provable worst-case guarantees. In our experiments, we select these hyperparameters via a grid search over a range of scales, as reported in Tables 2 and 6. Overall, our proposed algorithms are fairly robust to hyperparameter selection, yielding consistently higher performance than the baselines across various step sizes and clipping levels. To ensure reliability of the results, we plot the average run across five different seeds for each tuned configuration.
>
> - **Differentiation from closely related works**: In Appendix A, we provide a comprehensive comparison with prior work on SFW methods with variance reduction as well as prior work on gradient clipping under heavy-tailed noise. Specifically, our variance-reduced algorithm matches the best-known SFO complexity bound of $O(1/\epsilon^3)$ in the literature of SFW with variance reduction [4], [5], [6]. Simultaneously, our variant uses an amortized average batch size of at most 2, thereby avoiding the need for large batches throughout the iterations, unlike [4] and [5]. Furthermore, in contrast to [6], our theoretical guarantees hold without requiring Hessian-based assumptions or structural constraints on the data distribution. On the other hand, the literature concerning FW-type algorithms in the presence of heavy-tailed noise is quite limited, with [7] being the only prior work to address this setting. However, unlike our work, which assumes bounded $p$-th moment noise, [7] adopt a different set of assumptions on the stochastic noise and assume convexity of the function. Therefore, our work is the first to systematically study Stochastic Frank-Wolfe methods for non-convex optimization under heavy-tailed noise.
>
> - **Empirical evidence and baseline comparison**: We have provided experimental results across three diverse tasks and model architectures that support our theoretical results, where the proposed variants consistently yield faster convergence. We would also like to clarify that our work primarily aims to contribute to the theoretical and algorithmic foundations of Stochastic Frank-Wolfe (SFW) methods, and as such, all algorithms and convergence proofs in our work are for SFWs and are not tied to Lion and Muon. In particular, the empirical results are intended to validate the robustness of our SFW theory rather than to introduce a brand-new deep learning optimizer for competitive benchmarking. Furthermore, we note that comparative evaluations between Lion, Muon, and optimizers like AdamW have already been studied in prior works, see e.g. [8], [9], [10].
>
> [1] Cutkosky and Mehta (2021), High-probability bounds for Non-Convex Stochastic Optimization with Heavy Tails
>
> [2] Zhang et al. (2020), Why are adaptive methods good for attention models?
>
> [3] Liu et al. (2023), Breaking the Lower Bound with (Little) Structure: Acceleration in Non-Convex Stochastic Optimization with Heavy-Tailed Noise
>
> [4] Yurtsever et al. (2019), Conditional gradient methods via stochastic path-integrated differential estimator
>
> [5] Hassani et al. (2020), Stochastic conditional gradient++
>
> [6] Zhang et al. (2020), One sample stochastic frank-wolfe
>
> [7] Tang et al. (2022), High-probability bounds for robust stochastic frank-wolfe algorithm
>
> [8] Chen et al. (2024), Symbolic discovery of optimization algorithms
>
> [9] Liu et al. (2025), Muon is scalable for LLM training
>
> [10] Shah et al. (2025), Practical Efficiency of Muon for Pretraining

---

> > ### Author Rebuttal · Reviewer_ndSg · 2026-04-01
> >
> > I thank the authors for their response and my recommendation as a weak accept remains.

---

> > > ### Author Response · Authors · 2026-04-06
> > >
> > > We would like to thank the reviewer again for their supportive and thorough review!

---

### Official Review · Reviewer_Fx2h · 2026-03-12

**Soundness:** 2
**Presentation:** 3
**Significance:** 3
**Originality:** 3
**Overall Recommendation:** 4
**Confidence:** 4

**Summary:**

In this paper, authors studiy modern optimizers such as Lion and Muon in the form of Stochastic Frank-Wolfe method. They also propose a theoretical guarantees, which are consistent with convergence toward a KKT point of the original problem under a norm constraint. Moreover, different modifications (variance reduced procedure and gradient clipping for the heavy-tailed regime) were investigated. To clarify theoretical findings, authors validate extensive numerical experiments.

**Compliance With Llm Reviewing Policy:**

Affirmed.

**Final Justification:**

The authors addressed all of my concerns. In my view, the paper has one remaining drawback: the conceptual limiting transition in the absence of the weight decay parameter. The framework proposed by the authors does not allow this case to be recovered, even though analyses of this kind for Muon/Lion methods already exist.

**Key Questions For Authors:**

1) Do the authors think it is possible to extend the lemma to obtain bias and variance bounds for the clipped stochastic gradient in the case where the norm comes from a Banach space without any critical dimension-dependent factor?

**Limitations:**

Yes

**Strengths And Weaknesses:**

### Strengths

I would like to highlight the following points:

1) **Interpretation of popular approaches.** For me, the main contribution of this work lies precisely in interpreting methods such as Lion/Muon with weight decay through the lens of stochastic Frank-Wolfe. This is a very vivid example of how new optimizers can incorporate ideas from classical approaches.
2) **Theoretical results.** While checking the proof, I did not find any critical flaws.
3) **Numerical experiments.** The authors also validate their findings by conducting experiments that correlate with real-world scenarios.

### Weaknesses

Despite the advantages above, I can emphasize the only one drawback.

***Why reformulate an unconstrained optimization problem as a constrained one?*** If I understood correctly, the final convergence bounds stated in the theorems and corollaries depend on the diameter of the feasible set $D$ in the constrained problem. As mentioned in Section 3, the diameter of this set is inversely proportional to the weight decay parameter, i.e., $D \sim \frac{1}{\lambda}$. This implies that as $\lambda \rightarrow 0$, we essentially move to the Muon/Lion method without weight decay. However, in that case $D \rightarrow +\infty$, which leads to the following issue: the convergence theorems would no longer apply, and the stochastic Frank-Wolfe interpretation breaks down. On the other hand, there are convergence analyses of Muon based on different interpretations; for example, [1] interprets Muon within a trust-region framework. I understand that weight decay is a well-established tool in deep learning, but typically reducing it to zero should not worsen theoretical convergence; yet without it, the proposed Frank-Wolfe interpretation would not be possible.

---

### References

[1] Understanding Gradient Orthogonalization for Deep Learning via Non-Euclidean Trust-Region Optimization. 2025

---

> ### Author Rebuttal · Authors · 2026-03-31
>
> We appreciate the reviewer's thorough reading of our paper and the thoughtful feedback provided.
>
> - **Constrained formulation**: We emphasize that the primary objective of this work is to advance the theoretical and algorithmic foundations of **Stochastic Frank-Wolfe methods**. Accordingly, the proposed algorithmic variants and their convergence guarantees are formulated at the level of the Stochastic Frank-Wolfe and are not tied to specific instances such as Lion or Muon. These methods instead serve as motivating and practically relevant examples for machine learning setups. Their empirical success, together with the observation that their update rules can be recovered within an SFW formulation, provides a clear motivation for studying this broader class in a unified manner. \
> A key aspect of this connection is that Frank-Wolfe-type methods are inherently designed for **constrained** optimization problems. As a result, both the algorithms and their analysis in terms of the Frank-Wolfe gap do not directly extend to the unconstrained setting. In particular, the limit $\lambda \to 0$ corresponds to an effectively unconstrained problem, in which the Frank-Wolfe method and the associated convergence guarantees in terms of the Frank-Wolfe gap are no longer appropriate.
> As discussed in the paper, there exists  prior work that studies the convergence of Muon without weight decay using different measures of convergence, see e.g., [1], [2], [3], [4], [5], and the reference provided by the reviewer. In addition, we note that the original algorithmic formulation of Lion includes by default the weight decay term. However, the focus of our work is not to provide theoretical guarantees for the specific cases of Lion and Muon, but rather for the algorithmic family of the proposed Stochastic Frank-Wolfe. In any case, we have explicitly acknowledged the limitation of the analysis of Lion and Muon without weight decay from the SFW perspective in the discussion section of our work. \
> From a practical perspective, however, weight decay is not merely a technical artifact. Recent works [6], [7], [8] consistently indicate that Lion and Muon with weight decay outperform their vanilla counterparts. Furthermore, weight decay is recognized as a decisive determinant of optimization stability and convergence [9], [10], with recent evidence [11] suggesting it plays a pivotal role in improving model plasticity during large-scale language model training. Therefore, incorporating weight decay is both aligned with practical use and essential for capturing the regimes in which these methods are most effective.
>
> - **Different norms**: We thank the reviewer for this important question. Extending our algorithmic guarantees to cases where the gradient clipping norm is defined in a general Banach space without any dimension-dependent factor is indeed a compelling direction for future work. Such an extension would involve reformulating our assumptions to accommodate general norms and likely leveraging an appropriate notion of smoothness property, which captures curvature in Banach spaces and replaces the Hilbert space inner-product structure with a more general functional counterpart. A good starting point would be to adapt the toolkit on concentration inequalities for Banach spaces presented for the case of normalized SGD with momentum in [12], which provides high-probability bounds in non-Euclidean geometries. However, a rigorous treatment of our Stochastic Frank-Wolfe algorithms within more general geometries would require further investigation with careful effort.
>
> We believe our clarifications have addressed the reviewer's primary concern regarding the scope of our analysis and the role of weight decay. In light of this, we would appreciate it if the reviewer would consider raising their score.
>
> [1] Li \& Hong (2025), A note on the convergence of
> muon
>
> [2] Shen et al. (2025), On the convergence analysis of muon
>
> [3] An et al. (2025), Adaptive structured gradient optimization
>
> [4] Riabinin et al. (2025), Gluon: Making muon \& scion great again! (bridging theory and practice of lmo-based optimizers for llms)
>
> [5] Pethick et al. (2025), Training deep learning models with norm-constrained lmos.
>
> [6] Liu et al. (2025), Muon is scalable for LLM training
>
> [7] Chen et al. (2024), Lion Secretly Solves Constrained Optimization: As Lyapunov Predicts
>
> [8] Zhao et al. (2025), Deconstructing What Makes a Good Optimizer for Language Models
>
> [9] D’Angelo et al. (2024), Why Do We Need Weight Decay in Modern Deep Learning?
>
> [10] Wang and Aitchison (2024), How to set AdamW’s weight decay as you scale model and dataset size
>
> [11] Han et al. (2026), Weight Decay Improves Language Model Plasticity
>
> [12] Cutkosky and Mehta (2021), High-probability bounds for Non-Convex Stochastic Optimization with Heavy Tails

---

> > ### Author Rebuttal · Reviewer_Fx2h · 2026-04-03
> >
> > Thank you to the authors for the fairly detailed response. I fully agree that the authors are in fact investigating the Stochastic Frank-Wolfe approach in various forms, rather than Muon and Lion in their specific variants. As I see it, this work should be viewed as an SFW framework that, in some sense, makes it possible to connect to currently popular optimizers; to some readers, the logic may appear to be constructed in reverse. Nevertheless, this does not in any way diminish the contribution of the paper, and I agree with the authors’ comments regarding my concerns. Accordingly, I am ready to raise my score to 4.
> >
> > Unfortunately, given the central idea of the work - interpreting Muon/Lion as SFW - there are some minor conceptual issues with the limiting transition (weight decay parameter $= 0$), so I am not ready to give a higher score.

---

> > > ### Author Response · Authors · 2026-04-06
> > >
> > > We are grateful for the reviewer’s positive assessment of our contribution and for recognizing the intended perspective of our work through the Stochastic Frank-Wolfe lens. We sincerely appreciate the reviewer's feedback and thoughtfulness in reviewing our work!

---

### Decision · Program_Chairs · 2026-04-30

**Decision:**

Accept (regular)

**Comment:**

The paper develops new theoretical results for the Stochastic Frank–Wolfe (SFW) algorithm and its variants, including clipping and variance reduction, and establishes a clear connection between SFW and widely used optimizers such as Muon. The authors derive in-expectation convergence rates for the expected Frank–Wolfe gap under standard assumptions of smoothness and bounded stochastic gradient variance. They further extend these results to a momentum-based variance reduction scheme, obtaining improved rates. In addition, the paper provides novel high-probability guarantees under heavy-tailed noise for clipped variants of the method, both with and without variance reduction. The theoretical findings are complemented by small-scale numerical experiments using nanoGPT.

Overall, the paper makes a meaningful and timely contribution. The results are technically sound, clearly presented, and align with the consensus of the reviewers. The concerns raised by Reviewers Chtx, Fx2h, and ndSg have been adequately addressed. Reviewer mKKs raises questions regarding the level of insight and the discussion of technical challenges. While the rebuttal does not explicitly elaborate on technical difficulties, I do not view this as a critical weakness: the results are novel and address an important problem. Moreover, expectations around highlighting technical challenges are inherently subjective, particularly once the main results have been established.

Regarding practical insights, while their inclusion would strengthen the paper, they are not essential for a primarily theoretical contribution. More importantly, the paper engages with several highly relevant topics in modern optimization for machine learning (such as Muon-style methods, gradient clipping, heavy-tailed noise, and high-probability analysis) and is likely to stimulate further research in these directions.

Taking into account the reviews, the authors’ responses, the area chair discussion, and my own assessment, **I recommend acceptance**. I encourage the authors to **incorporate the reviewers’ suggestions in the final version**. In particular, **revising the title to better reflect the main contributions would help avoid potential confusion**.